# TLR2 on blood monocytes senses dengue virus infection and its expression correlates with disease pathogenesis

José A. Aguilar-Briseño [1,6], Vinit Upasani[1,2], Bram M. ter Ellen[1], Jill Moser[3], Mindaugas Pauzuolis[1], Mariana Ruiz-Silva[1], Sothy Heng[4], Denis Laurent[4], Rithy Choeung[5], Philippe Dussart [5], Tineke Cantaert [2✉], Jolanda M. Smit[1] & Izabela A. Rodenhuis-Zybert [1✉]

Vascular permeability and plasma leakage are immune-pathologies of severe dengue virus (DENV) infection, but the mechanisms underlying the exacerbated inflammation during DENV pathogenesis are unclear. Here, we demonstrate that TLR2, together with its co-receptors CD14 and TLR6, is an innate sensor of DENV particles inducing inflammatory cytokine expression and impairing vascular integrity in vitro. Blocking TLR2 prior to DENV infection in vitro abrogates NF-κB activation while CD14 and TLR6 block has a moderate effect. Moreover, TLR2 block prior to DENV infection of peripheral blood mononuclear cells prevents activation of human vascular endothelium, suggesting a potential role of the TLR2-responses in vascular integrity. TLR2 expression on CD14 + + classical monocytes isolated in an acute phase from DENV-infected pediatric patients correlates with severe disease development. Altogether, these data identify a role for TLR2 in DENV infection and provide insights into the complex interaction between the virus and innate receptors that may underlie disease pathogenesis.

[1] Department of Medical Microbiology and Infection Prevention, University of Groningen and University Medical Center Groningen, 9700 RB Groningen, The Netherlands. [2] Immunology Group, Institut Pasteur du Cambodge, International Network of Pasteur Institutes, Phnom Penh 12201, Cambodia. [3] Departments of Critical Care, Pathology & Medical Biology, Medical Biology section, University Medical Center Groningen, University of Groningen, 9700 RB Groningen, The Netherlands. [4] Kantha Bopha Hospital, Phnom Penh 12000, Cambodia. [5] Virology Unit, Institut Pasteur du Cambodge, International Network of Pasteur Institutes, Phnom Penh 12201, Cambodia. [6] Present address: Department of Microbiology and Immunology, University of Iowa, Iowa City, IA 52242, USA. ✉email: tcantaert@pasteur-kh.org; i.a.rodenhuis-zybert@umcg.nl

The four serotypes of dengue virus (DENV1-4) are estimated to cause 390 million infections per year, of which 96 million manifest clinically[1]. Clinical outcomes of DENV infection vary considerably and can be either limited to an acute febrile illness referred to as dengue fever (DF), or progress to a potentially fatal disease (severe dengue) encompassing dengue hemorrhagic fever (DHF) and dengue shock syndrome (DSS)[2]. Severe disease is associated with a transient increase in vascular permeability due to endothelial dysfunction initiated by increased levels of soluble inflammatory mediators, such as IL-1β and TNF-α, released early in the course of infection[3,4]. Factors exacerbating inflammation such as high viral titers and presence of cross-reactive, infection-enhancing antibodies raised from previous infection with another serotype, increase the risk of severe disease[5–7]. The mechanisms governing the initiation of excessive inflammation however, remain poorly understood. Consequently, there is currently no diagnostic marker indicative of severe disease nor specific treatment options available for DENV patients.

Cells of the innate immune system together with epithelial and endothelial cells are responsible for the early activation of inflammatory responses to invading pathogens. Pattern-recognition receptors (PRRs) expressed on these cells detect and respond to a variety of pathogen-associated molecular patterns (PAMPs) as well as to tissue-derived danger associated molecular patterns (DAMPs), released from stressed or dying cells. Upon engagement, PRRs activate intracellular signaling cascades inducing pro-inflammatory responses[8]. Regulation of this response is crucial, since exacerbated release of pro-inflammatory mediators is known to trigger adverse effects including excessive endothelial inflammatory activation, vascular permeability and hemorrhagic manifestations such as those seen in severe dengue patients[9–11].

Toll-like receptor 2 (TLR2) is one of the PRRs expressed on the surface of immune cells. Although generally known as a sensor of bacterial lipoproteins, TLR2 can also sense molecular patterns of viruses[12–18]. The ligand-binding specificity of TLR2 is modulated by its heterodimerization partners: TLR1 or TLR6 and co-stimulatory molecules CD14 and CD36[8,19,20]. Engagement of the TLR2 axis leads to activation of NF-κB pathway, increased gene expression and release of inflammation-driving mediators such as inter alia IL-1β and TNF-α. In human blood, expression and activation of TLR2 has been shown to regulate the function of many cell types including those representing early targets of DENV replication, such as dendritic cells (DC's) and monocytes[21–23]. Chen et al., attributed TLR2-activation following DENV infection to one of the nonstructural viral proteins, nonstructural protein 1 (NS1) that is released from cells replicating the virus[24]. Notably, however, subsequent studies by two independent groups debated Chen's data demonstrating that NS1 protein in fact engages TLR4 instead of TLR2[25–27] and indicated that the conclusions were compromised due to the use of impure and misfolded *Escherichia coli*-derived recombinant NS1[26]. Thus, to date, the mechanism and significance of TLR2-upregulation in the context of DENV pathogenesis remains unknown.

In the current study, we examine TLR2 expression on PBMCs isolated from 54 pediatric patients in the acute phase of DENV infection with different disease outcomes. Furthermore, using an in vitro PBMC infection and human vascular endothelium activation models, we investigate the function of TLR2 during DENV infection. Altogether, our results identify TLR2 as a key regulator of DENV-induced inflammation and a prognostic marker of immunopathology.

## Results

**TLR2 on monocytes of acute patients correlates with disease severity.** Circulating monocytes represent a versatile and dynamic cell population, composed of multiple subsets which differ in phenotype, and function[28,29]. In humans, these discrete monocyte subsets can be distinguished by the expression of CD14 and CD16[28]. CD14++CD16− classical monocytes (CM) make up ~85% of the circulating monocyte pool, whereas the remaining ~15% consist of CD14++CD16+ intermediate (IM) and CD14+ CD16++ non-classical monocytes (NM)[30,31]. Importantly, their frequencies are influenced by inflammatory conditions[32]. Previous studies have shown that DENV infection in vivo increased frequencies of either IM or NM[21,33]. In line with previous studies, our patient cohort showed an overall increase in IM and NM when compared to age-matched healthy controls (Fig. 1a and Supplementary Fig. 1a). In addition, Azeredo et al.[21] demonstrated increased TLR2 expression on blood monocytes in DENV-infected patients when compared to healthy controls yet how TLR2 is distributed over the three monocyte subsets and its potential impact on disease burden remain elusive[21,24]. To investigate this, we isolated PBMCs from DENV-infected patients during the acute phase of infection ($n = 54$) and 15 age-matched healthy donors (HD) and subsequently stained with an anti-human TLR2 antibody or a conjugated isotype-matched antibody as a control (Supplementary Fig. 1b). Patients were classified for disease severity according to the WHO 1997 guidelines as specified in Methods section and their characteristics are listed in Table 1. We proceeded to determine DENV infection in monocyte subsets. To ensure detection of active DENV replication, rather than viral uptake, we used a rabbit polyclonal antibody against DENV non-structural protein 3 (NS3) and subsequently a secondary antibody labeled with FITC. A non-specific rabbit polyclonal antibody along with the same secondary antibody was used as a negative control (Supplementary Fig. 2a, b). The anti-NS3 staining was only performed on a subset of patient samples that had a sufficient cell yield after PBMC thawing to perform a good quality intracellular staining ($n = 15$). In line with the above studies, immunophenotyping of patients' PBMCs showed that both CM and IM had significantly higher expression ($p < 0.0001$, two-tailed Mann–Whitney test) of TLR2 on their surface compared to NM (Fig. 1b). However, when TLR2 expression in DENV+ patients was compared to HD, all monocyte subsets from HDs had significantly higher TLR2 expression ($p < 0.0001$, two-tailed Mann–Whitney test) compared to DENV+ patients (Fig. 1b). As the expression of TLR2 on monocytes during the acute phase of DENV infection could vary according to the days post-fever, we classified the DENV-positive patients according to the number of days post-fever (day 2–5). TLR2 expression on CM, IM and NM was not dependent on the number of days post fever that the samples were obtained (Supplementary Fig. 3). Interestingly, CM from patients who developed DHF/DSS showed significantly higher expression of TLR2 when compared to patients with mild disease (DF) ($p < 0.01$, two-tailed Mann–Whitney test). No differences were observed for IM while for NM, patients with severe dengue had marginally lower percentage of TLR2 compared to DF patients ($p < 0.05$, two-tailed Mann–Whitney test) (Fig. 1c). Similar results were yielded when TLR2 expression was stratified based on the infecting serotype; CM from patients who developed DHF/DSS following DENV1 and DENV2 infections showed significantly higher expression of TLR2 when compared to those who developed DF ($P < 0.05$, two-tailed Mann–Whitney test) (Supplementary Fig. 4a, b, respectively) while no differences were observed for IM and NM (Supplementary Fig. 4). Unfortunately, there were not enough patients to evaluate the correlation between TLR2 expression and disease severity following DENV serotypes 3 and 4. Notably, CM and IM, but not NM, were the primary targets of DENV replication as measured by the percentage of cells positive for DENV NS3 (Fig. 1d). The percentage of CM predominated in patients

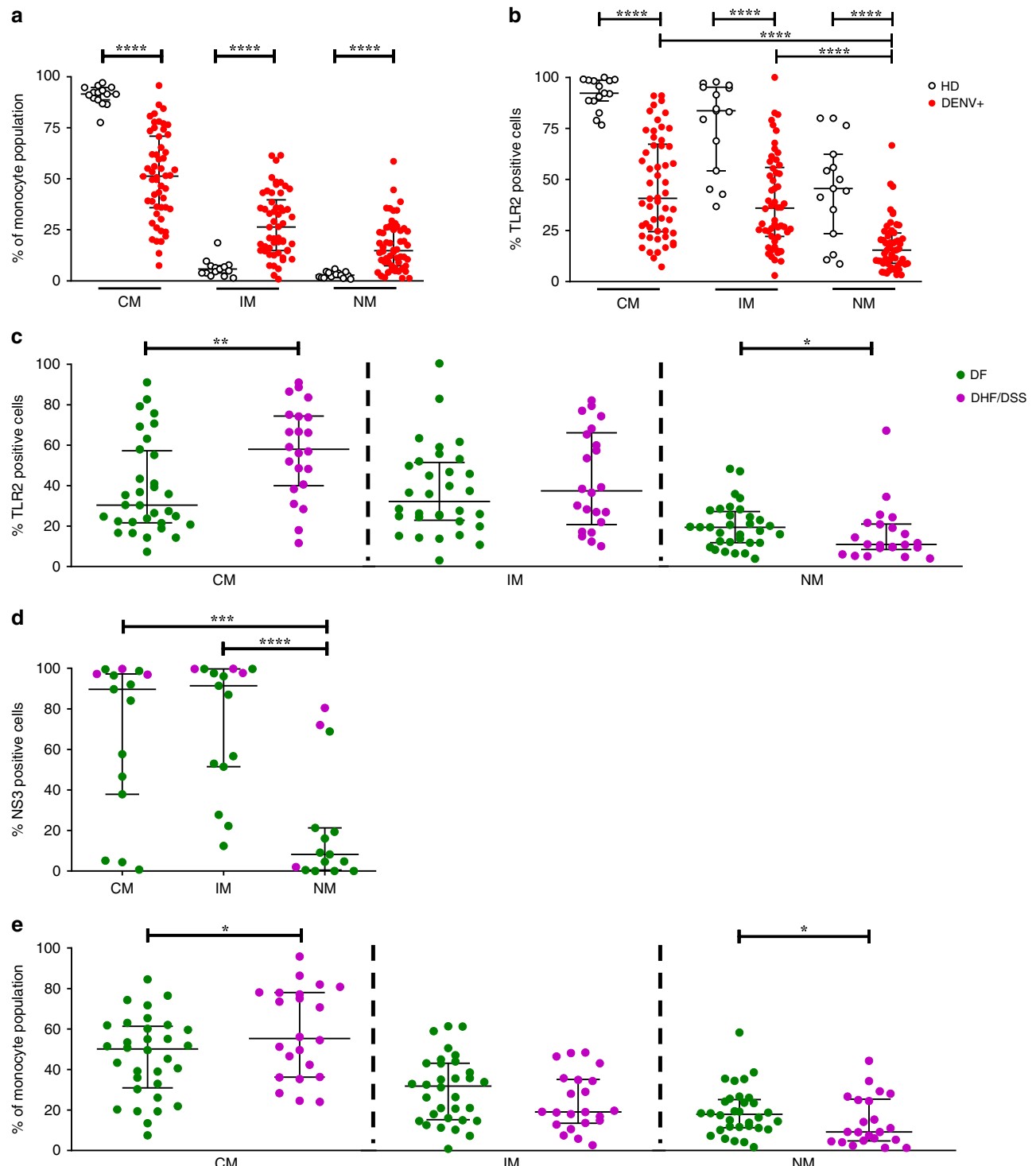

**Fig. 1 Sustained high expression of TLR2 and increased frequency of monocytes correlates with DENV disease severity. a–e** PBMCs were isolated from 15 age-matched healthy donors (HD) and 54 patients undergoing acute DENV infection (DENV+) who developed relatively mild (DF, n = 32) or severe (DHF/DSS, n = 22) disease. **a** Monocyte subsets distribution in healthy and DENV+ patients (two-tailed Mann–Whitney test, ****P < 0.0001). **b** Percentages of cells expressing TLR2 were determined for each monocyte subset (two-tailed Mann–Whitney test, ****P < 0.0001) and **c** stratified by disease severity (two-tailed Mann–Whitney test, *P < 0.05; **P < 0.01). **d** Percentages of NS3+ infected cells in DENV-positive patients (n = 15) stained intracellularly for DENV NS3 (two-tailed Mann–Whitney test, ***P < 0.001; ****P < 0.0001). **e** Monocyte subsets distribution in patients, stratified by disease severity (two-tailed Mann–Whitney test, *P < 0.05). CM classical monocytes, IM intermediate monocytes, NM non-classical monocytes. Bars represent median with interquartile range (IQR). Source data are provided as a Source data file.

**Table 1 Demographic data and clinical parameters of the studied population.**

| Studied population | Total patients ($n = 54$) | DF $N = 32$ | DHF/DSS 22 (DHF −12; DSS −10) | Healthy donors ($n = 15$) |
|---|---|---|---|---|
| Age | $8.14 \pm 4.01$ | $8.29 \pm 4.29$ | $7.29 \pm 3.67$ | $10.08 \pm 4.06$ |
| M/F ratio | 0.8 | 0.9 | 0.7 | 1.5 |
| Weight (kg) | 22.7 | 23.7 | 21.2 | 29.3 |
| Height (cm) | 121.1 | 122.6 | 119.3 | 127.5 |
| Temperature (°C) | 37.6 | 38.0 | 37.1 | NA |
| Hematocrit (%) | 42.8 | 38.0 | 44.0 | |
| Platelets ($\times 10^9$/L) | 93.7 | 116.3 | 65.0 | |
| Day of fever (Mean) | 3.6 | 3.3 | 3.9 | |
| DENV1 | 8 | 5 | 3 | |
| DENV2 | 37 | 21 | 16 | |
| DENV3 | 0 | 0 | 0 | |
| DENV4 | 3 | 3 | 0 | |
| N/A | 6 | 3 | 3 | |
| NS1+ | 32 | 21 | 11 | |
| PCR+ | 48 | 30 | 18 | |
| Viral load (copies/ml) (Median, IQR) | $2.2 \times 10^4$ | $1.2 \times 10^5$ | $7.6 \times 10^3$ | |
| | $(7.4 \times 10^3 - 3.4 \times 10^5)$ | $(1.4 \times 10^4 - 1.2 \times 10^6)$ | $(6.3 \times 10^3 - 3.5 \times 10^4)$ | |
| Secondary infection | 74% | 65% | 90% | |

that developed severe disease ($p < 0.5$, two-tailed Mann–Whitney test) while there were no significant changes in the percentages of IM. In addition, patients that developed DF had a moderately increased percentage of NM ($P < 0.5$, two-tailed Mann–Whitney test) (Fig. 1e). Altogether, these data suggest that sustained high levels of TLR2 on CM during the acute phase of DENV infection are associated with severe disease development.

**DENV2 engages TLR2/6 and CD14 to trigger NF-κB activation.** To investigate if TLR2 plays a role during DENV infection, we first assessed whether TLR2 is able to sense DENV particles. To this end, we used reporter cells HEK-Blue™ hTLR2 cells (InvivoGen) that stably co-overexpress human TLR2/6/1, CD14 and NF-κB/AP1-inducible SEAP (secreted embryonic alkaline phosphatase) genes. Stimulation of TLR2 is monitored by the activation of NF-κB/AP1. Synthetic lipopeptides PAM3CSK4 (PAM3) and PAM2CSK4 (PAM2), potent agonists of TLR2/1/CD14 and TLR2/6, respectively, were used as positive controls in the assays. Interestingly, we observed that DENV2 strain 16681 induces activation of HEK-Blue™ hTLR2 cells (Fig. 2a) but not the parental HEK-Blue™ Null1 cells (Supplementary Fig. 5a). As shown in Fig. 2a, NF-κB activation was not dependent on viral replication as UV-inactivation of the virus (UV-DENV) did not abrogate the sensing, and increased with viral dose (Supplementary Fig. 5b). Moreover, purified DENV virions [pDENV] activated NF-κB demonstrating that molecular patterns present on the surface of the virus particles eg. (pr)M/E proteins rather than a soluble factor are responsible for the activation. DENV-mediated NF-κB activity was significantly inhibited upon blocking the TLR2 receptor ($p < 0.0001$, paired one-tailed $t$ test) and significantly attenuated by blockage of the TLR2 co-receptors: TLR6 and CD14 ($p < 0.001$, one-way ANOVA, Dunnett post hoc test) (Fig. 2b, c). Importantly, PAM3 and our standard DENV preparations did not activate HEK-Blue™ hTLR4 cells (Supplementary Fig. 5c, d), which only responded to LPS and TNF-α treatments. These results confirm the specificity of the HEK-TLR Blue system employed and imply that the soluble form of DENV nonstructural protein 1 (NS1), previously shown to signal through TLR4[25,26], was not a confounding factor in our experiments.

To evaluate the capacity of mosquito and human-derived DENV to engage TLR2, we next compared the ability of DENV-2 16681 produced in mosquito C6/36 cells to activate NF-κB in the

HEK-Blue™ hTLR2 cells with that produced on monocyte-derived DC's and monocyte-derived macrophages (Mφ). The human-derived viruses were obtained by infecting monocyte-derived DC's and monocyte-derived Mφ with C6/36-derived virus for 2 h, after which the surplus of inoculating virus and incubation was continued until 48hpi. Following titrations of DC-and Mφ -derived DENV2 preparations, HEK-Blue™ hTLR2 cells were exposed to increasing numbers of virus particles i.e. multiplicity of genomes (MOGs) to ensure fair comparison. Interestingly, DC-derived DENV2 induced NF-κB activation on hTLR2 cells albeit ~3× lower than the C6/36-derived virus at a similar MOG (Supplementary Fig. 5b), suggesting that the structural differences between human and mosquito-derived viruses modulate the TLR2 recognition. Notably however, Mφ-produced DENV2 did not induce NF-κB activation at any of MOG tested, implying that TLR2 engagement may be sensitive to changes in virion characteristics that are not intrinsic to mosquito or human cells, but are cell-type specific. It is important to mention however that due to the overall lower yield of DENV released by macrophages, MOG of 4000 could not be reached and thus more in-depth studies are needed to test the premise of intra-host cell type-specific factors influencing TLR2 activation by DENV. Altogether, our data revealed that TLR2 has the capacity to sense DENV virions.

While expressed at the plasma membrane, TLR2-induced NF-κB activation is controlled by clathrin-mediated endocytosis (CME), in which CD14 serves as an important upstream regulator[34]. Accordingly, we used not-toxic concentrations of various perturbants of endocytic pathways, to dissect whether CME is required for TLR2 to sense DENV. Specifically, pit-stop (PS), an inhibitor of clathrin-pit formation was used to inhibit CME, $NH_4Cl$ was used to neutralize the pH of intracellular compartments; wortmannin (W), a PI3K inhibitor affecting phagocytosis and macropinocytosis served as a negative control[35]. Chloroquine, chlorpromazine and dynasore were excluded from the experiments due to their high level of cytotoxicity (Supplementary Fig. 6). PS and $NH_4Cl$ significantly reduced but did not abrogate NF-κB activation mediated by PAM3 ($p < 0.05$, one-way ANOVA, Dunnett post hoc test) and DENV ($p < 0.05$, one-way ANOVA, Dunnett post hoc test), whereas no significant effect was seen for PAM2 (Fig. 2d). As expected, wortmannin treatment did not alter TLR2-mediated NF-κB activation by any of the agonists. Thus, activation of TLR2/6/CD14 axis by DENV

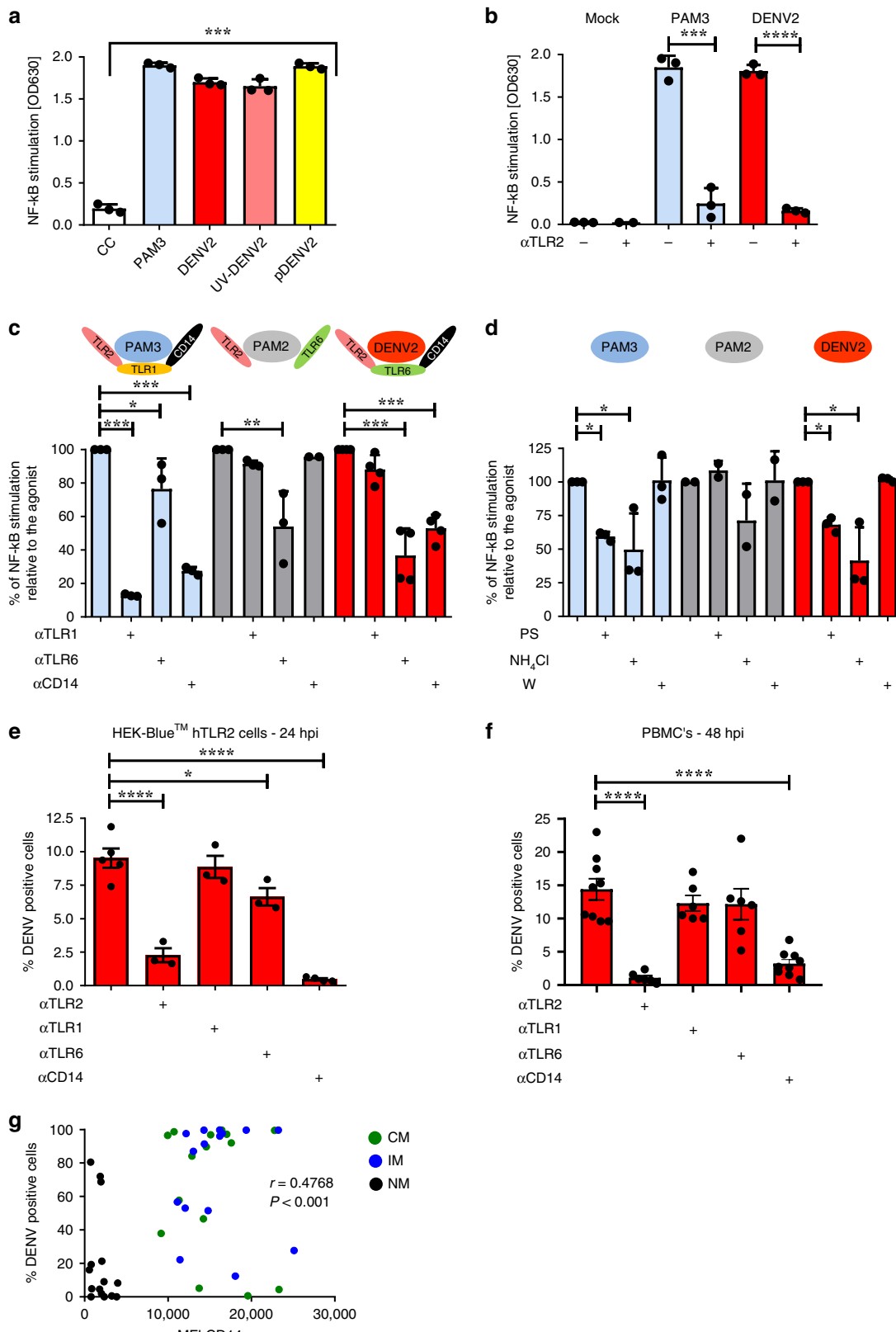

occurs at the plasma membrane and can be potentiated by internalization of the complex via CME.

Interestingly, many viruses, including DENV, hijack CME to gain access to internal compartments of host cells[36,37]. Moreover, CD14 and its unknown co-receptor has been previously proposed to act as an attachment receptor for DENV[38]. We therefore

hypothesized that TLR2/CD14-dependent CME facilitates DENV entry to establish infection. Accordingly, we tested if blockade of TLR2 or its co-receptors had any impact on the percentage of infected cells and/or virus production. Since, due to technical issues, we could not exploit the same NS3 antibody as we used in our patient cohort, we tested the accumulation of E protein and

**Fig. 2 DENV engages TLR2/6 and CD14 to activate NF-κB and to establish infection. a** NF-κB activation in HEK-Blue™ hTLR2 cells (mock-) treated with PAM3CSK4 (PAM, 25 ng/mL), DENV2 (MOG1000), UV-I DENV2 (MOG1000) or purified DENV2 (pDENV2) (MOG1000) for 24 h ($n = 3$ one-way ANOVA, Dunnett post hoc test, ***$P < 0.001$). OD630 values represent induction of NF-κB. **b, c** NF-κB activation in HEK-Blue™ hTLR2 cells pretreated for 2 h with **b** αTLR2 ($n = 3$, one-tailed paired $t$ test, ***$P < 0.001$; ****$P < 0.0001$) or **c** αTLR1, αTLR6 and αCD14 (15 μg/mL) before exposure to PAM3 (25 ng/mL), PAM2CSK4 (PAM2, 1 ng/mL) or DENV2 (MOG1000) for 24 h ($n = 3$, one-way ANOVA, Dunnett post hoc test, *$P < 0.05$; **$P < 0.01$; ***$P < 0.001$). **d** NF-κB activation in HEK-Blue™ hTLR2 pretreated for 1 h with endocytosis inhibitors pitstop (PS, 60 μM), ammonium chloride (NH4Cl, 50 mM) and wortmannin (W, 2 μM) prior to exposure to PAM3 (25 ng/mL), PAM2 (1 ng/mL) or DENV2 (MOG1000) for 24 h ($n = 3$, one-way ANOVA, Dunnett post hoc test, *$P < 0.05$). Percentages of DENV (E)—positive cells were determined by flow cytometry in **e** HEK-Blue™ hTLR2 cells ($n = 5$, one-way ANOVA, Dunnett post hoc test, *$P < 0.05$; ****$P < 0.0001$) or **f** monocytes in the context of PBMCs, in the presence or absence of TLR2 axis blockade after 24 and 48 h, respectively ($n = 3$, three donors and three different DENV2 preparations, one-way ANOVA, Dunnett post hoc test, ****$P < 0.0001$). Bars represent the mean ± SEM. CC (cellular control), PAM2, PAM3, and DENV2 as controls of their respective blocking/treatment conditions. N refers to the number of independent biological experiments unless otherwise specified. **g** Correlation of mean fluorescence intensity (MFI) of CD14 expression on different monocyte subsets and DENV infection (NS3) in our patient's cohort. Statistical differences were determined by Spearman correlation analysis. Source data are provided as a Source data file.

used UV-inactivated DENV to ensure we do not detect incoming virus in our assay (Supplementary Fig. 7a, b, and Supplementary Fig. 8a). As shown in Fig. 2e, blockage of TLR2, CD14 ($P < 0.0001$, one-way ANOVA, Dunnett post hoc test) and to a lesser extent that of TLR6 ($P < 0.05$, one-way ANOVA, Dunnett post hoc test), reduced the number of DENV-Ag-positive HEK-Blue™ hTLR2 cells. Consistent with these results, in PBMCs, specific blocking of TLR2 ($P < 0.0001$, one-way ANOVA, Dunnett post hoc test) and CD14 ($P < 0.0001$, one-way ANOVA, Dunnett post hoc test) but not that of TLR1/6 significantly decreased the frequency of DENV Ag-positive monocytes (Fig. 2f, and Supplementary Fig. 8b for isotype controls). These data might explain why among the three monocyte subsets, the TLR2-positive CD14[++] monocytes (Fig. 1d) expressed the highest level of DENV NS3 during acute DENV infection and why CD14 expression correlates with DENV replication in vivo (Fig. 2g). Counterintuitively, however, despite the significant decrease in the frequency of DENV-Ag-positive cells, TLR2 and CD14 blockage had little to no effect on viral production in both cell models (Supplementary Fig. 9). There was also no clear correlation between the expression of TLR2 (percentage of positive cells and MFIs) and the viral load in our patient's cohort (Supplementary Fig. 10a, b), suggesting that TLR2/CD14-dependent CME does not significantly contribute to progeny virus release and/or other cells such as DCs[39] are the main source of the virus. Altogether, these data indicate that host cells utilize TLR2/6/CD14 mediated NF-κB activation as a quick innate mechanism to sense DENV infection.

**TLR2-mediated sensing of DENV is virus strain rather than the serotype-specific.** We proceeded to investigate whether TLR2-engagement is a PAN-dengue phenomenon. All four DENV serotypes had the ability to activate NF-κB in a TLR2-dependent manner (Table 2) yet, not all strains and/or genotypes of the same serotype had that property. Other mosquito-borne viruses including West Nile Virus (WNV), Zika virus (ZIKV), and chikungunya virus (CHIKV), did not activate NF-κB in this system (Table 2). Interestingly, despite the varying ability of several DENV2 16681 preparations used in this study to activate NF-κB, all of them infected the HEK-Blue™ hTLR2 cells and PBMCs in TLR2/CD14-dependent manner (Fig. 2e, f). Thus, TLR2-mediated NF-κB activation but not TLR2 axis-mediated infection appears to be DENV strain/genotype specific.

**In vitro DENV infection upregulates TLR2 and CD16 on monocytes.** To further substantiate the role of TLR2 as a regulator of inflammatory responses, we isolated PBMCs from healthy, DENV-seronegative, donors and infected them under

**Table 2 TLR2 engagement by DENV is strain specific.**

| Virus | Strain | TLR2/NF-κB activation |
|---|---|---|
| DENV1 | 16007 | +++ |
|  | Hawaii | − |
| DENV2 | 16681 | +++/++[a] |
|  | NGC | +++ |
|  | TSV01 | + |
| DENV3 | 16562 | +++ |
|  | H87 | − |
| DENV4 | 1036 | + |
|  | H241 | − |
| ZIKV | SL0216 | − |
| WNV | NY99 | − |
| CHIKV | LR OPY1 | − |

Source data are provided as a source data file.
++ NF-kB activation is half the triggered by PAM3CSK4, + 10–20% activation of NF-kB in comparison to PAM3CSK4, − Does not trigger NF-kB activation.
[a]Differences between various preparations

TLR2 axis blocking and non-blocking conditions with DENV2 16681 strain at multiplicities of infection (MOI) of 10, as described previously[40]. To gain further insights into the possible repercussions of TLR2-engagement on PBMCs, we used virus preparations that had a differential capacity to activate HEK-Blue™ hTLR2 reporter cells (Table 2). To discriminate between pathways triggered due to sensing and/or by replication, an equal dose of UV-inactivated virus was used as a control in all experiments. Regardless of virus preparation, in vitro DENV infection of monocytes (within PBMCs) increased the mean fluorescent intensity (MFI) of TLR2 (Fig. 3a and Supplementary Fig. 11) and the percentage of TLR2-positive cells (Fig. 3b). In contrast, UV-DENV (Fig. 3a, b) and PAM3CSK4 (Supplementary Fig. 12a, b) did not upregulate TLR2 expression when compared to mock-infected cells. In addition, neither DENV infection nor TLR2 agonists had an effect on the expression of TLR2 on lymphocytes (Supplementary Fig. 12c, d). Notably, the increase in TLR2 expression following in vitro-infection was in contrast to the data collected from our ex vivo samples (Fig. 1b) but in line with previous findings[21]. Importantly, PBMCs isolated from adult healthy and DENV-seronegative donors in the Netherlands expressed similar levels of TLR2 as our pediatric HD in Cambodia. This might suggest that monocyte responses and thereby the regulation of TLR2 expression on the surface of these cells depends on the age, genetic background and/or past DENV infection. Thus, in vitro DENV infection but not ex vivo infection leads to the selective upregulation of TLR2 on monocyte fractions.

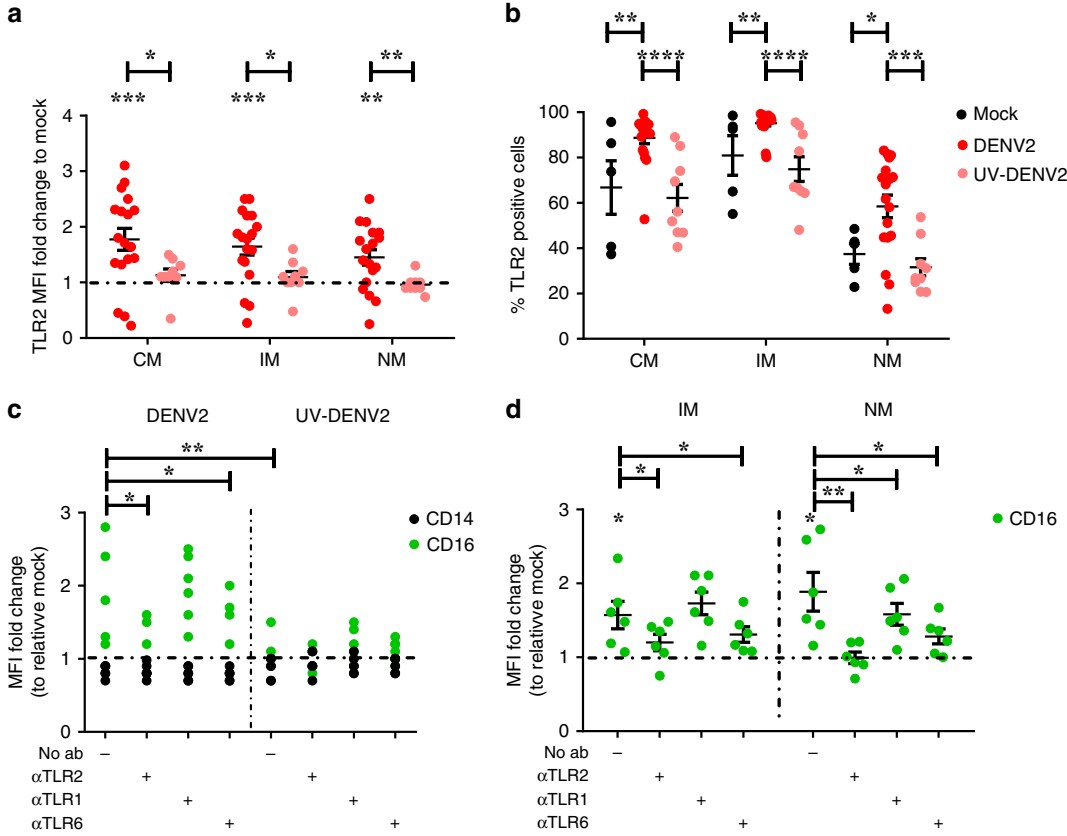

**Fig. 3 Active DENV infection upregulates TLR2 and increases CD16 expression in a TLR2/TLR6 dependent manner.** PBMCs from healthy donors were (mock-) treated with αTLR2, αTLR1 and αTLR6 (5 μg/mL) for 2 h prior to infection with DENV2 at MOI of 10 or its UV-inactivated equivalent (UV-DENV2) for 48 h. **a** MFI of TLR2 expression ($n = 5$, five different donors and up to 4 different DENV2 preparations, paired one-tailed t test, $*P < 0.05$; $**P < 0.01$, $***P < 0.001$) and **b** Percenatge of TLR2-positive cells within each of monocytes subset ($n = 5$, five different donors and up to four different DENV2 preparations, paired one-tailed $t$ test, $*P < 0.05$; $**P < 0.01$, $***P < 0.001$, $****P < 0.0001$). **c** CD14 and CD16 expression were detected for total monocytes in blocking and non-blocking condition ($n = 2$, two different donors and three different viral preparations, paired one-tailed $t$ test, $*P < 0.05$; $**P < 0.01$) and **d** per monocyte subset ($n = 2$, two different donors and three different viral preparations, paired one-tailed $t$ test, $*P < 0.05$; $**P < 0.01$). CM classical monocytes, IM intermediate monocytes and NM: non-classical monocytes. Data represented as fold-changes in MFI relative to the respective mock or as a percentage of positive cells of the respective marker. Bars represent mean ± SEM. Source data are provided as a Source data file.

Immunophenotyping of in vitro-infected cells showed that both DENV and UV-DENV significantly increased the frequencies of CM ($P < 0.0102$ and $P < 0.0022$, respectively, unpaired one-tailed $t$ test) and NM ($P < 0.0088$ and $P < 0.0123$, respectively, unpaired one-tailed $t$ test) while the IM population was decreased ($P < 0.0001$ and $P < 0.0012$, respectively, unpaired one-tailed t test) when compared to mock-treated PBMCs (Supplementary Fig. 12e). This indicates that sensing of virions is enough to trigger a shift in the monocyte subpopulations. This effect was, however, independent of TLR2, since TLR2 block did not prevent the increase in CM numbers (Supplementary Fig. 12e). Similar results were obtained for PAM3 (Supplementary Fig. 12f), suggesting that other receptors are involved in this process. Notably, we found that active DENV infection but not UV-DENV, significantly increased the expression of CD16 in intermediate and non-classical monocytes ($p < 0.0136$ and $p < 0.0099$, respectively, paired one-tailed $t$ test) (Fig. 3c, d). Moreover, this upregulation was in control of TLR2 and TLR6 but not that of TLR1, as blockade of TLR2 and TLR6 significantly reduced ($P < 0.0211$ and $P < 0.0383$, respectively, paired one-tailed $t$ test) the upregulation of CD16 induced by DENV infection (Fig. 3d). Remarkably, in patients, expression of CD16 was negatively associated with the percentage of DENV-infected cells (Supplementary Fig. 13) suggesting that TLR2/6-mediated CD16 upregulation might serve as an antiviral mechanism. This

would explain, at least in part, why sustained levels of TLR2 expression on NM correlated with mild disease (Fig. 1c). There was no difference in the expression of CD14 after DENV infection with or without blocking conditions (Fig. 3c).

**TLR2 controls DENV infection-induced inflammatory responses of PBMC.** Activation of blood cells as a consequence of DENV infection leads to the production of inflammatory cytokines, which in turn activates human endothelial cells and can lead to the loss of their barrier function[3,41–43]. To test whether TLR2 engagement during DENV infection of PBMCs contributed to the vascular responses, we incubated human umbilical vein endothelial cells (HUVEC) with supernatants of infected PBMCs, as described in Fig. 4a. Endothelial cell activation was assessed by quantification of surface and/or mRNA expression of E-selectin, vascular cell adhesion protein 1 (VCAM-1), intercellular adhesion molecule 1 (ICAM-1) and inflammatory mediators including *IL-6, IL-8, MCP-1*, and *CXCL6*. To ensure that the activation of endothelial cells was due to soluble inflammatory mediators excreted by infected PBMCs rather than the presence of the virus itself, HUVEC were incubated with an equal number of GEc as those present in the supernatants of infected PBMCs. Incubation for six hours did not lead to the upregulation of adhesion molecules when compared to LPS (positive control) (Supplementary Fig. 14a, b). Thus, endothelial cell activation observed

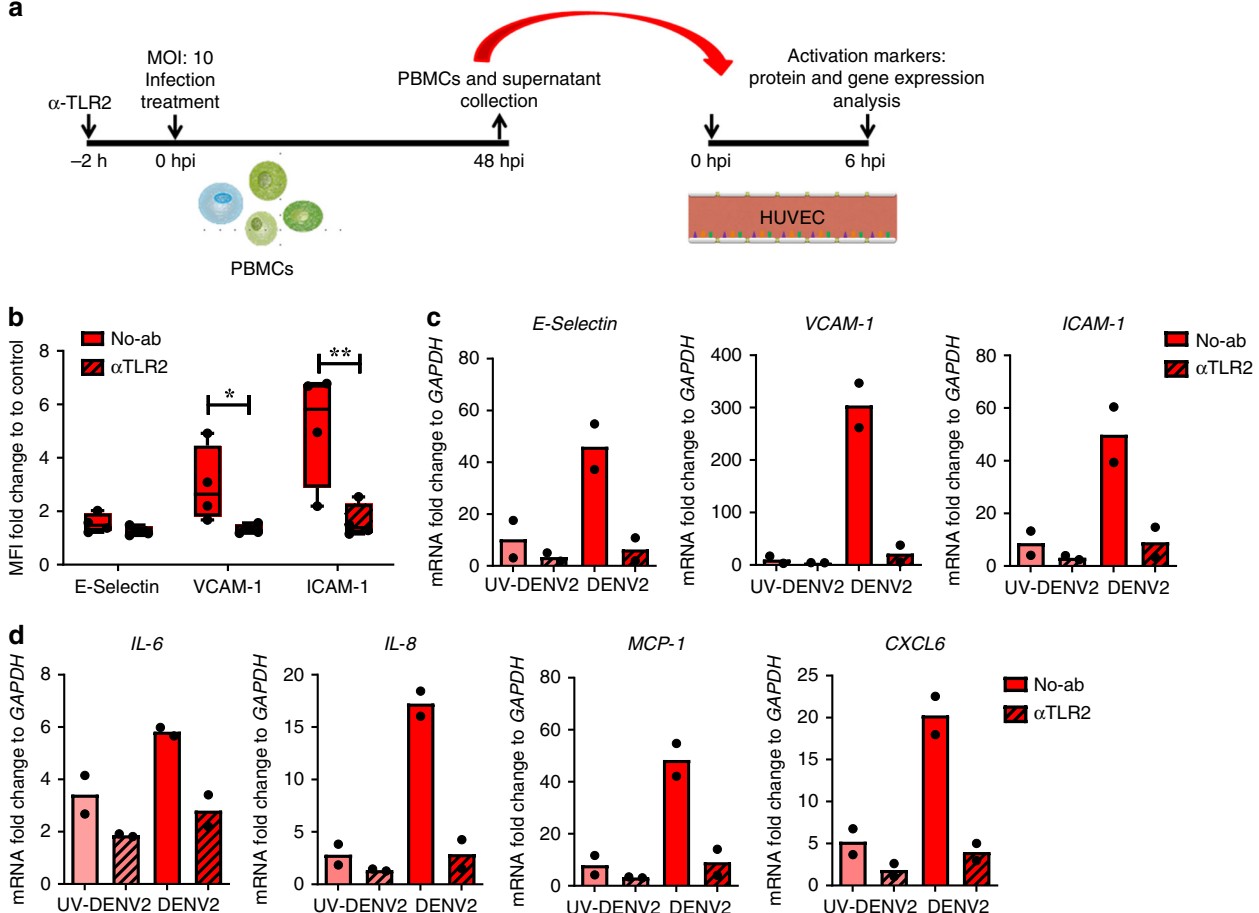

**Fig. 4 Active DENV infection induces inflammatory mediators in a TLR2-dependent manner. a** HUVEC were incubated for 6 h with cell-free supernatants from PBMCs infected in the presence or absence of TLR2 blockade. **b** Boxplots show the fold-changes in surface expression of E-selectin, VCAM-1 and ICAM-1 compared to the respective mock ($n = 4$, paired one-tailed $t$ test, *$P < 0.05$; **$P < 0.01$). The horizontal line represents the median and the whiskers the minimum and maximum values. **c** Fold-changes in gene expression levels of *E-selectin*, *VCAM-1*, and *ICAM-1* relative to the respective mock ($n = 2$). **d** Fold-changes in gene expression levels of *IL-6*, *IL-8*, *MCP-1*, and *CXCL6* relative to the respective mock ($n = 2$). N refers to the number of independent biological experiments in HUVEC. Source data are provided as a Source data file.

with the supernatants of infected PBMCs was due to soluble mediators excreted upon infection in PBMCs (Fig. 4b, c) and not due to the carryover of the virus. Surprisingly, blocking TLR2 on PBMCs prior to exposure to DENV infection abolished inflammatory responses and significantly attenuated endothelial activation as evidenced by decreased expression of adhesion molecules (Fig. 4b), mRNA levels of their respective genes (Fig. 4c) as well as reduced transcription of inflammatory cytokines and chemokines such as *IL-6*, *IL-8*, *MCP-1*, and *CXL6* (Fig. 4d). Supernatants from UV-DENV treated PBMCs led to a relatively mild activation of HUVEC when compared to infectious DENV (Fig. 4c, d; Supplementary Fig. 15a, b), despite potent activation of NF-κB in HEK-Blue™ hTLR2 by UV-DENV (Fig. 2a). Additionally, isotype control antibody block did not have an effect on the vascular responses of PBMCs infected with DENV2 (Supplementary Fig. 15c). Moreover, the TLR2 block had no effect on the soluble inflammatory responses of PBMCs treated with the TLR4/CD14 agonist LPS, indicating that the effect observed for DENV infection is indeed TLR2-specific (Supplementary Fig. 15d).

Increased levels of IL-1β and TNF-alpha have been reported to be present in the plasma of DENV patients and may contribute to increased endothelial permeability[3]. To determine whether endothelial activation during DENV infection was due to the presence of these cytokines, we measured the intracellular

accumulation of IL-1β and TNF-alpha in DENV-infected PBMCs (Supplementary Fig. 16). As expected, positive control PAM3CSK4 induced the intracellular accumulation of IL-1β and TNF-α in the concentration and TLR2-dependent manner (Supplementary Fig. 17a, b). Interestingly, at the concentration tested only the intracellular accumulation of IL-1β was significantly reduced by prior TLR2 blockade (Supplementary Fig. 17a), suggesting that differential pathways down-stream of TLR2- trigger the production of these cytokines[44]. Indeed, TLR2 signaling may also lead to the activation of the inflammasome pathway that contributes to the production of IL-1β[45,46]. Alternatively, signal integration of different PRR can lead to different and non-additive immune responses[47,48]. Importantly, at 12 hpi we did not detect TNF-α accumulation following exposure to DENV2 (Supplementary Fig. 17c) while IL-1β accumulation was evident in the monocyte fraction of three out of four tested donors and depending on the donor was induced by exposure to either infectious (DENV) or non-infectious virus UV-DENV (Supplementary Fig. 17d). None of the treatments induced intracellular accumulation of both cytokines in the lymphocytic fraction of the PBMCs (Supplementary Fig. 17e, f). Therefore, we analyzed inflammatory mediators released from the cells throughout 48 h of infection. Interestingly, the concentrations of IP-10 ($P < 0.01$, one-way ANOVA, Dunnett post hoc test), IFN-α2 ($P < 0.05$, one-way

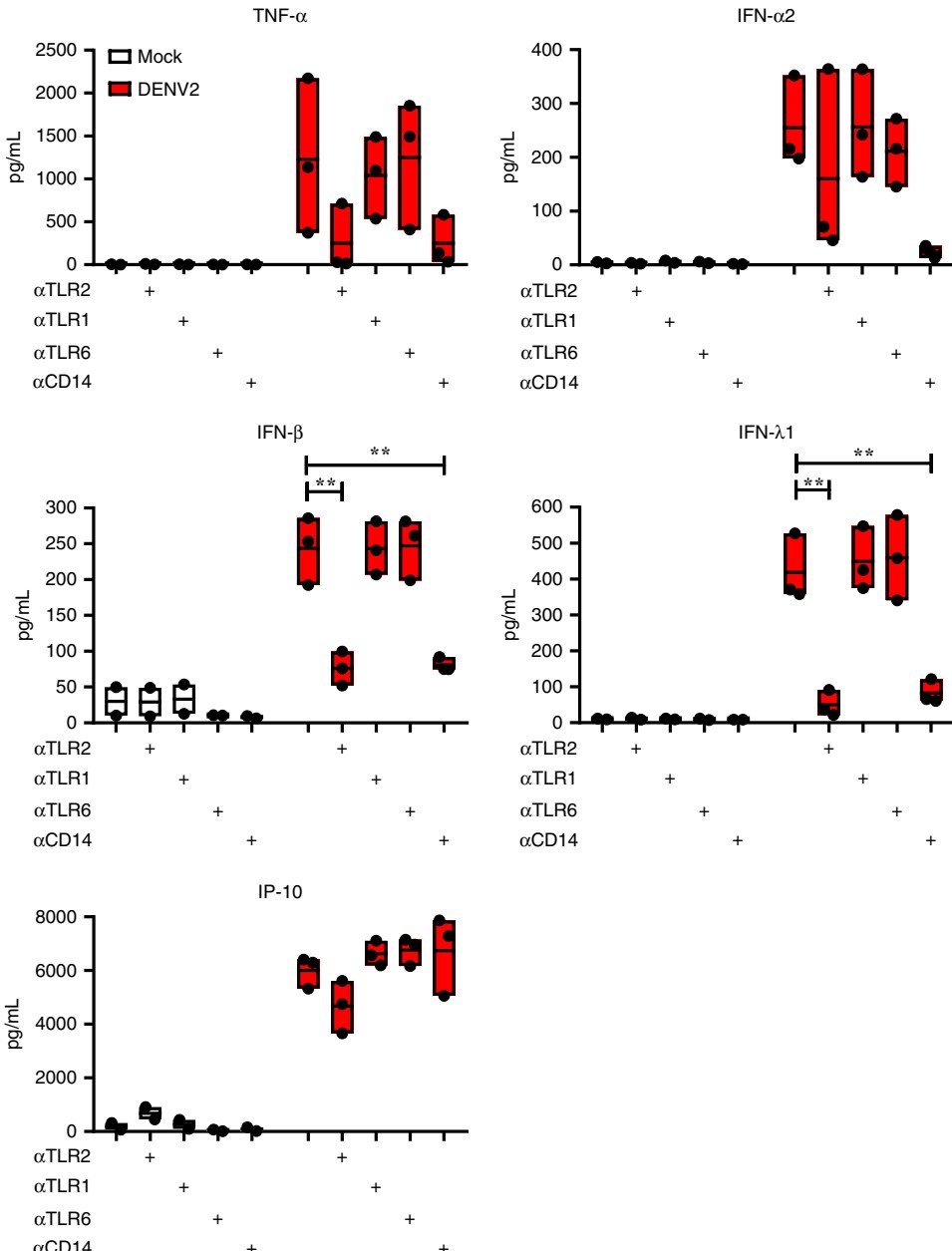

**Fig. 5 TLR2/CD14-dependent cytokines induced by DENV2 infection.** PBMCs from healthy donors were (mock)—treated with αTLR2, αTLR1, αTLR6 (5 μg/mL) and αCD14 (3 μg/mL) for 2 h prior to infection with DENV2 at MOI of 20 for 48 h ($n = 2$, two different donors and 3 different DENV2 preparations, one-way ANOVA, Dunnett post hoc test, **$P < 0.01$). Cytokine production was measured by flow cytometry using LegendPlex. Each boxplot in the graphs shows the production in picograms per milliliter (pg/mL) of the respective cytokine. The horizontal line represents the mean and the bottom and top of the box show the minimum and the maximum values. Source data are provided as a Source data file.

ANOVA, Dunnett post hoc test), IFN-β ($P < 0.05$, one-way ANOVA, Dunnett post hoc test) and IFN-λ1 ($P < 0.01$, one-way ANOVA, Dunnett post hoc test) were significantly higher in the supernatants from DENV2-infected PBMCs in comparison from those of UV-DENV2-stimulated PBMCs (Supplementary Fig. 18), while only a slight increase was observed for TNF-α, IL-6, IL-10 and IFN-γ (Supplementary Fig. 18). Additionally, UV-DENV2 also induced production of TNF-α, IL-1β, IL-6 and IFN-λ2/3 (Supplementary Fig. 18), however not as potently as the replicative virus. Remarkably, blocking TLR2 and CD14 but not TLR1/6 or the use of control antibody (Fig. 5 and Supplementary Fig. 19), significantly decreased the production of IFN-β ($P < 0.01$, one-way ANOVA, Dunnett post hoc test) and IFN-λ1 ($P < 0.01$) induced by active DENV infection, while the production of TNF-

α was marginally reduced in these conditions (Fig. 5). The levels of IFN-α2 were moderately reduced when blocking CD14 but not when blocking other TLR2 (co-) receptors, while only the block of TLR2 was able to reduce the levels of IP-10 (Fig. 5). The TLR2 axis block did not impair the levels of IL-1β, IL-6, IL-10, and IFN-γ induced by DENV infection (Supplementary Fig. 20). Altogether, these results support the DENV replication- dependent and TLR2-mediated production of inflammatory mediators capable to activate endothelial cells in vitro (Fig. 4) and highlight that variety of pathways cross-talking following TLR2 axis-dependent and independent sensing of DENV infection. In addition, it is important to note that the observed variations in cytokine production following (UV-) DENV infections were donor- rather than virus preparation-dependent, further

highlighting the inherent differences in TLR2-mediated sensing of DENV infection between the overexpression system HEK-Blue™ hTLR2 and primary monocytes, which are equipped with multiple PRRs. Altogether, our data show that TLR2 sensing of dengue virus infection induces production of inflammatory mediators, which in turn can activate endothelial cells.

## Discussion

By analyzing TLR2 expression and the frequencies of the different monocyte subsets in a DENV-infected pediatric cohort, we identified a prognostic value of TLR2 expression in disease pathogenesis. We show that in children, DENV infection lead to the overall decrease in the TLR2 expression on the surface of all monocyte subsets when compared to age-matched HD. Notably, the sustained relatively high expression of TLR2 in acute infection on CM was associated with severe disease development while profound reduction in TLR2 expression correlated with mild disease. The functional analyses uncovered the ability of TLR2/6/CD14 to sense DENV infection and to drive infection-mediated inflammatory responses leading to activation of human vascular endothelium. On the other hand, however, TLR2 and CD14 co-receptors increased the infected cell mass thereby revealing a pro-viral role for TLR2. Altogether, our data provided evidence for the critical and dual role of TLR2 in DENV infection and infection-mediated immune responses.

Our data have identified TLR2 as a potent regulator of immune responses following DENV infection. TLR2 together with TLR4 have been shown to play an important role in the induction of inflammation during many infectious diseases[49]. For instance, single-nucleotide polymorphisms occurring in the TLR2/1/6 axis have been reported to influence the course of chlamydiosis, leprosy and hepatitis B and C virus infections[50]. Notably, during the course of DENV infection, TLR2 appears to have a both protective and unfavorable role for the host. Although high frequencies of TLR2-expressing CM correlated with severe disease, the opposite trend, albeit less strong, was noted for non-classical monocytes. The seemingly contrasting roles of TLR2 on CM and NM are likely to be linked to their distinct functions. CM are equipped with various PRRs and scavenger receptors that recognize PAMPS, remove microorganisms, lipids, and dying cells via phagocytosis and are thus, involved in sensing and inducing inflammatory responses to stress-inducing factors. In contrast, NM or patrolling monocytes have a unique role in protecting endothelial integrity by removing damaged cells and debris. Moreover, once activated, NM differentiate into anti-inflammatory macrophages to repair damaged tissues, thereby promoting wound healing and the resolution of inflammation[51–54]. The functions of TLR2 on these monocyte subsets may also be dictated by the differential expression of CD14, CD16, TLR10, CD36[20,55–58] and/or other receptors capable of modulating DENV infection and DENV-infection-mediated responses[59,60].

Changes in monocytes and monocyte subset distributions have been associated with many inflammatory diseases, though their relevance for disease pathogenesis is poorly understood[21,32,33]. DENV infection has been shown to lead to an increase in frequencies of IM (from ~5 in healthy to ~10 % in DENV patients) accompanied by a modest decrease (~77% in healthy to ~63% in frequencies of CM[21,33]. Based on their inflammatory makeup, prevalence of CM and IM in the blood of DENV patients could be linked to exacerbated inflammation and severe disease. Interestingly, even though DENV infection increased the frequencies of IM and NM compared to HD, this increase did not differentiate between the severity groups. In fact, in our patient cohort, only the increased frequency of CM correlated with severe disease development. Notably however, in vitro, we observed a similar shift in monocyte subsets distribution with UV-inactivated DENV as with the infectious virus while only the latter led to TLR2-mediated CD16 upregulation and release of inflammatory mediators. Altogether, this suggests that changes in monocyte distribution cannot be considered as a direct and a sole marker of inflammatory responses.

Differential surface expression of TLR2 on CM of patients with relatively mild disease symptoms (DF) and those who progressed to severe disease (DHF/DSS) suggests a distinct regulation of TLR2 expression in these patients. Based on our data and that of others'[34] TLR2-axis-mediated NF-κB activation is partially dependent on CME. Thus, the reduction of TLR2 expression on the surface of CM during viremia observed in patients with DF might indicate that activation of the DENV infection-mediated TLR2 axis led to TLR2/ligand internalization, and ultimately desensitization of the monocytes to TLR2-engaging PAMPs and DAMPS[61]. In addition, TLR2 complex internalization might also result in more balanced inflammatory and antiviral immune responses, since internalization is required to induce IFN type I producing signaling cascades[18]. At the same time, and considering the ability of TLR2 blockade to limit the number of DENV-Ag-positive monocytes, it is internalization would also impede the ability of the virus to infect these cells. Conversely, a sustained relatively high TLR2 surface expression following DENV infection, as seen on CM of patients that developed severe disease, could be suggestive of a reduced internalization of the TLR2 complex. In this scenario, prolonged sensing of TLR2—engaging PAMPs and DAMPS would lead to mainly pro-inflammatory responses and a relatively higher number of infected monocytes. The exact mechanisms governing TLR2 expression following DENV sensing and/or infection remain to be elucidated and will be the focus of our future studies. Importantly, reduced TLR2 surface expression following infection may also mirror the release of soluble TLR2, the levels of which were shown to be elevated in some infection and inflammatory conditions[62–64]. Moreover, plasma levels of soluble TLR2 were found to be associated with a resolution of inflammation[64].

The engagement of TLR2 was independent of the ability of DENV to replicate and relied on the sensing of the viral particle suggesting the involvement of structural proteins expressed on the virus surface [E, (pr)M] in the engagement of this PRR rather than any of the five non-structural proteins (NS1- NS5). This is in stark contrast to Chen et al., who attributed TLR2-mediated responses following DENV infection to the NS1 protein[24]. Importantly, subsequent studies have provided evidence that NS1 protein engages with TLR4 instead of TLR2[25–27]. Interestingly, regardless of Chen et al. results obtained with NS1 protein, their data from DENV–infected TLR6 KO mice are in fact, in consonance with our present findings and corroborate the potentially unfavorable for the host role of the TLR2 axis in DENV infection in vivo.

DENV produced in mosquito cells triggered stronger TLR2-dependent NF-κB activation than the DC- and Mθ-derived virus. This suggests that the virus transmitted during the blood meal is likely to initiate and more significantly contribute to TLR2-mediated inflammation. The reason for the differences in the capacity to activate TLR2 axis depending on the virus origin is yet unclear however is likely attributable to the host- and/or cell-type specific modifications on the surface of the virus particle. For instance, differential glycosylation patterns of two potential N-linked glycosylation sites on the DENV E protein produced in mosquito and primary human DCs influence their capacity to interact with DENV receptors, DC-SIGN and L-SIGN, and thus may dictate DENV tropism in vivo[65]. Interestingly, E protein is not the only protein on the surface of the virion that can be glycosylated. DENV prM protein contains a single glycosylation site, and although prM is cleaved by furin during viral

maturation, a substantial fraction of uncleaved prM is present on some DENV particles. In fact, DENV exists as a number of different viral forms depending on the degree of maturation. PrM -containing fully immature and partially mature DENV virions are particularly abundant in mosquito and mammalian tumor cell lines-produced viral preparations[66,67]. On the other hand, DENV produced in primary cells[68] and human circulating DENV1 virions[69] seem to show a higher degree of maturity than those produced in cell lines. Considering the reduced ability of human primary cell derived DENV2 to engage TLR2 described in our study, it will thus be important to address how both glycosylation and maturation levels of the virions influence TLR2-mediated responses during infection. Unfortunately, due to the limited amount of sample obtained from our pediatric cohort, we could not assess the effect of the human circulating virus in our in vitro systems. However, we are currently assessing the role of immature dengue particles in our in vitro models.

The results obtained in PBMCs did not entirely mirror the results obtained in HEK-Blue™ TLR2 cells since the UV-irradiated virus preparation did not activate NF-κB to the same extent as the replicative virus. The level of NF-κB/AP1 activation measured in the reporter cell lines is unlikely to reflect the variety of different immunomodulators that can be released down-stream of TLR2 in primary immune cells. The differences in cytokine levels produced between active DENV infection and UV-DENV in PBMCs may reflect the differential expression, as well as the crosstalk of different PRR's in sensing replicative and non-replicative virions, which do not occur, or are not detectable in the NF-kB/AP-1 reported system[48,70]. Indeed, by UV-irradiating the virus, we likely impeded the endosomal sensing of viral ssRNA by TLR7/8, which ultimately might have affected the potentiating effect of TLR2/7/8 cross talk[71]. Moreover, the TLR2/CD14-dependent production of IFN-λ1, which has been described to be released by dendritic cells in a TLR3 dependent manner[72], may suggest the possibility of TLR2/TLR3 cross talk[73]. Indeed, in human DC's the stimulation of TLR2 blocked the induction of cytokines that are controlled by TLR3[47]. Further studies are required to elucidate the cross talk of different PRR's in the course of DENV infection.

Our study provides fundamental insights into the function of TLR2 in the course of DENV infection, however some limitations should also be considered. Nearly 74% of our cohort included patients undergoing secondary infections implying that they had pre-existing cross-reactive antibodies circulating in the blood at the time of sampling. Our in vitro infection model, however, did not address the infection enhancing or neutralizing effects of dengue-specific antibodies on TLR2-mediated immune responses[60]. For instance, the phenomenon of antibody-dependent enhancement (ADE) of infection, postulating that sub neutralizing concentrations of DENV-specific antibodies, can facilitate an additional mode of entry, thereby enhancing DENV infection and the aberrant inflammatory responses seen in DHF/DSS patients. Additionally, the presence of DENV-Ab immunocomplexes is likely to influence TLR2/CD14-mediated responses[60]. Lastly, considering the role of TLR2 and TLR4 in detecting bacterial PAMPS, it will be important to address the effect of common co-morbidities including bacterial and parasitic co-infections or microbial translocation[6,7,74,75], on DENV pathogenesis and prognosis.

In conclusion, our data identified a fundamental role of cell surface-expressed TLR2 as a regulator of inflammatory responses in the course of DENV infection. DENV infection modulates TLR2 expression on monocytes, which depending on their co-stimulatory markers contribute to infection-mediated inflammatory responses that may underlie severe disease development. Consequently, our study disclosed the TLR2 axis as a potential pharmacological target which may mitigate the pathogenesis of severe disease. Finally, sustained high expression of TLR2 in DENV patients that progress

to severe disease indicate a potential prognostic value of TLR2 on blood monocytes.

## Methods

**Ethics statement**. Ethical approval was obtained from the National Ethics Committee of Health Research of Cambodia. Written informed consent was obtained from all participants or the guardians of participants under 16 years of age before inclusion in the study.

**DENV patient recruitment**. Blood samples were obtained from hospitalized children (≥2 years) who presented with dengue-like symptoms at the Kanta Bopha Hospital in Phnom Penh, Cambodia. The time-point for collection of blood samples was within 96 h of appearance of fever in the patient. A second blood sample was collected upon discharge from the hospital. Patients were classified according to the WHO 1997 criteria upon hospital discharge[2]. Plasma leakage was confirmed by at least one of the following manifestations: 1/ a rise in the hematocrit equal to or >20% above average for age, sex and population in the admission sample (reference percentages: http://www.hematocell.fr/index.php/les-cellules-du-sang/15-les-cellules-du-sang-et-de-la-moelle-osseuse/valeurs-normales-de-lhemogramme-selon-lage/129-hemogramme-selon-lage) or 2/A drop in the hematocrit following volume-replacement treatment equal to or >20% of baseline and follow up visit or discharge (between 1 and 3 days after initial sample) or 3/Signs of plasma leakage such as pleural effusion and ascites by ultrasound.

**Laboratory diagnosis**. Serum specimens were tested for presence of DENV using nested qRT-PCR at the Institut Pasteur in Cambodia, the National Reference Center for arboviral diseases in Cambodia[76]. NS1 positivity was determined using rapid diagnostic tests (combo test for NS1 and IgM/IgG detection, SD Bioline Dengue Duo kits from Standard Diagnostics—Kyonggi-do, Korea). Anti-DENV IgM was measured with an in-house IgM-capture ELISA (MAC-ELISA), as previously described[77]. Patients were diagnosed as acute DENV-infected as following: a positive qRT-PCR or NS1 at hospital admission, or seroconversion from anti-DENV IgM negative to anti-DENV IgM positive during the hospital stay. Samples from patients positive for DENV were further tested with hemagglutination inhibition (HI) test to determine primary/secondary DENV infection as per WHO criteria[2].

**Patient PBMC phenotyping**. Human peripheral blood mononuclear cells (PBMCs) were isolated from blood samples of DENV-positive patients and age-matched healthy controls using Ficoll-Paque density gradient centrifugation and frozen in 10% DMSO until analysis. PBMCs were thawed in RPMI supplemented with fetal bovine serum (FBS), washed and counted in phosphate-buffered saline/bovine serum albumin. PBMCs were stained for surface markers using the following antibodies: TLR2 PE (1:25, clone TL2-1, #309707), CD14 APC (1:50, clone 63D3, #467117), CD16 PE-Cy7 (1:250, clone 3G8, #302016), CD19 AF488 (1:50, clone HIB19, #302219), CD3 BV510 (1:50, clone OKT3, #317332), CD4 PerCP/Cy5.5 (1:50, clone OKT4, #317428), CD45RO APC-Cy7 (1:25, clone UCHLI, #304228) and CD56 BV421 (1:25, clone 5.1H11, #362552), all purchased from BioLegend, and analysed on a FACSCanto II (BD). Isotype-matched antibody labelled with PE (1:25, clone MOPC-173, #400211, BioLegend) was used as negative control for comparing TLR2 expression. Due to the inability of the DENV envelope-directed 4G2 antibody (Millipore) to detect virus infection in our patients' PBMCs, we used the recently characterized rabbit polyclonal anti-DENV NS3 antibody. The anti-NS3 staining panel was only performed on a subset of patients as more PBMC are required for a good quality intracellular staining. Sufficient cell yield after PBMC thawing was used as selection criteria to perform the staining. Briefly, PBMCs were fixed and permeabilized using True nuclear transcription factor buffer set (BioLegend) and stained intracellularly using a rabbit anti-DENV NS3 antibody (10 μg/mL, #GTX124252, GeneTex) or rabbit IgG Isotype control (10 μg/mL, #910801, BioLegend) followed by goat anti-rabbit IgG conjugated with AF488 (1:500, #ab150077, Abcam). Data were analyzed using the FlowJo software (BD Biosciences).

**Cells**. PBMCs were isolated by standard density gradient centrifugation procedures with Ficoll-Paque™ Plus (GE Healthcare) from buffy coats obtained with written informed consent from healthy, anonymous volunteers, in line with the declaration on Helsinki (Sanquin Bloodbank, Groningen, the Netherlands). The PBMCs were cryopreserved at −196 °C. *Aedes albopictus* C6/36 cells were maintained in MEM supplemented with 10% FBS, 25 mM HEPES, 7.5% sodium bicarbonate, penicillin (100 U/mL), streptomycin (100 μg/mL), 200 mM glutamine and 100 μM non-essential amino acids at 30 °C. HEK-Blue™ hTLR2, HEK-Blue™ hTLR4 and HEK-Blue™ Null2 cell lines (InvivoGen) were cultured in DMEM supplemented with FBS, penicillin (100 U/mL), streptomycin (100 μg/mL) and maintained according to the manufacturer's instructions. VERO-E6 and VERO-WHO were cultured as the HEK cells with the addition of 10 mM HEPES and 200 mM glutamine. BHK-15 were cultured as the VERO cells with the addition of 100 μM of non-essential amino acids. Primary human umbilical vein endothelial cells (HUVEC) (Lonza, the Netherlands) were cultured in EBM-2 supplemented with EGM-2 endothelial growth SingleQuot kit supplement & growth factors (Lonza, the Netherlands). All

of the cell lines used in our experiments were tested negative for the presence of *Mycoplasma spp.* using a commercial functional method (Lonza, the Netherlands) and/or in-house qPCR assay adapted from Baronti et al.[78].

**Viruses**. DENV1 (strains 16007, Hawaii), DENV2 (strains 16681, NGC, TSVO1), DENV3 (strains H87, 16562) and DENV4 (strains 1036, H241), were produced in the *Aedes albopictus* C6/36 cell line as described before[79]. Briefly, and 80% confluent monolayer of cells was infected at multiplicity of infection (MOI) of 0.1. Depending on the serotype, virus progeny was harvested at 72 or 168 h post infection (hpi). Purification was performed as described by Ayala-Nuñez et al.[35]. WNV strain NY99 was propagated in Vero cells as described before[80]. CHIKV (La Reunion OPY1) was a gift from A. Merits (University of Tartu, Estonia), and was produced from infectious cDNA clones and passaged twice in Vero E6 cells[81]. ZIKV (SL0216) was a gift from M.J. van Hemert (LUMC, the Netherlands); virus was produced in Vero E6 cells and harvested at 72hpi. Viral preparations were analyzed with respect to the infectious titer and the number of genome equivalents copies. The infectivity of DENV and WNV were determined by measuring the number of plaque-forming units (PFU) by plaque assay on BHK-15 cells and the number of genome-equivalent copies (GEc) by quantitative RT-PCR (RT-qPCR), as described previously[82,83]. For CHIKV and ZIKV, the infectious virus titer was determined by standard plaque assay on Vero-WHO and Vero E6 cells, respectively and for CHIKV, reverse transcriptase quantitative PCR (RT-qPCR) was used to determine the number of genome equivalents copies (GEc)[84]. Virus inactivation was performed by 1.5 h incubation of virus aliquots under UVS-28.8 watt Lamp. Inactivation to below level of detection 35 PFU/mL was confirmed using standard plaque assay on BHK-15 cells as described previously[84]. Briefly, for plaque assays, cells were seeded in a 12-well plate. In all, 24 h later, cells were infected with 10-fold serial dilutions of the viral samples. At 2hpi cells were overlaid with 1% seaplaque agarose (Lonza, Swiss) prepared in 2x MEM. Plaques were counted at 44hpi for CHIKV, 72hpi for ZIKV and WNV, and 120hpi for DENVs. GEc were determined via RT-qPCR using a StepOne Real-Time PCR instrument (Applied Biosystems, Foster City, CA, USA). RNA extraction was performed using a QIAmp viral RNA mini kit (QIAGEN, Venlo, The Netherlands). cDNA synthesis from viral RNA was performed using Omniscript (QIAGEN) and the primers 5′-ACAGGCTATGGCA CTGTTACGAT-3′ (forward), 5′-TGCAGCAACACCATCTCATTG-3′ (reverse) for DENV, 5′AGCTCCGCGTCCTTTACCA-3′ (forward), 5′-GCCAAATTGT CCTGGTCTTCCT-3′ (reverse) for CHIKV, and (5′-GTTGGCGGCTGTTTTCT TTC-3′ (forward), 5′-GGGATCTCCCAGAGCAGAATT-3′ (reverse) for WNV. For qPCR the TaqMan probes 5′-FAM-AGTGCTCTCCAAGAACGGGCCTCG-T AMRA-3′, 5′-FAMCACTGTAACTGCCTATGCAAACGGCGAC-TAMRA-3′ and 5′-FAM–AATGGCTTATCACGATGCCCGCC–TAMRA-3′ (Eurogentec, Maastricht, The Netherlands) were used for DENVs, CHIKV and WNV, respectively. DNA amplification was performed for 40 cycles of 15 s at 95 °C and 60 s at 60 °C. The number of GEC was determined using a standard curve (correlation coefficient > 0.995) of a quantified cDNA plasmid containing either the DENV prM and E sequence, the CHIKV E1 sequence or the non-structural genes of WNV NY99. All virus preparations used in this study were tested negative for Mycoplasma *spp.* using a commercial functional method (Lonza, the Netherlands) and/or in-house qPCR assay adapted from Baronti et al.[78].

**In vitro infections**. PBMCs ($1 \times 10^6$ cell/mL) or HEK ($5 \times 10^4$ cells/mL) cells were treated with 5 μg/mL or 15 μg/mL, respectively of anti-hTLR2-IgA (clone B4H2, #maba2-htlr2), pab-hTLR6-IgG (polyclonal, #pab-hstlr6), pab-hTLR1-IgG (polyclonal, #pab-hstlr1), anti-hCD14-IgA (clone D3B8, #maba-hcd14), human IgA2 isotype control (clone T9C6, #maba2-ctrl) or normal rat PAb IgG control blocking antibodies (InvivoGen, #pab-sctr) for 2 h followed by treatment with PAM3CSK4 (600 ng/mL, InvivoGen), LPS (1 μg/mL, InvivoGen), DENV2 (MOI of 10 or its equivalent number of GEc (MOG of 1000) and UV-I DENV2 (MOG 1000 or MOG 3500) for 4, 6, 18, 24, 48, and 72 h). Since the production anti-hCD14IgA (clone B4H2) was discontinued, in experiments assessing the role of CD14 in DENV-infection, anti-hCD14-IgG (clone D3B8, #mabg-hcd14) and its Human IgG1 isotype control were used (#bgal-mab1, both from InvivoGen). For viral production analysis, cells were washed after infection and media replenished (600 μL), 24 h later supernatants were collected and preserved at −80 °C. For endothelial cells activation analysis, cell-free supernatants were collected at 4, 24, 48, and 72 h post infection/treatment and preserved at −80 °C.

**Monocyte subsets, TLR2 expression, and intracellular cytokines measurements**. Surface expression of TLR2, CD14 and CD16 was measured on PBMCs directly after thawing of the cells, after treatments with the blocking antibodies but prior to infection and 24 and 48 h post infection/treatment with the agonists. Briefly, PBMCs were stained with Fixable Viability Dye eFluor 780 (1:500, #65-0865), CD282 (TLR2) PE (1:10, clone TL2.1, #12-9922-41), CD14 eFluor 450 (1:20, clone 613D, #48-0149-41) and CD16 APC (1:20, clone CB16, #17-0168-41). All purchased from eBioscience. Cells were fixed and washed with FACs buffer (PBS 1x, 2% FBS). Intracellular accumulation of TNF-α and IL-1β was measured at 6 h and 18 h post infection/treatment, respectively. Briefly, Brefeldin A (10 μg/mL, BioLegend) was added 6 h prior cytokine detection. PBMCs where then incubated with fixable viability dye eFluor 780. Cells were fixed, permeabilized and

intracellularly stained with TNF-α eFluor 450 (1:10, clone MAb11, #48-7349-41) or IL-1β FITC (1:5, clone B-A15, #BMS127FI), both from eBioscience. Samples were measured on a FACSverse flow cytometer (BD Biosciences). Isotype-matched antibodies labelled with PE (1:10, clone eBM2a), eFluor 450 (1:20, clone P3.6.2.8.1), APC (1:20, clone P3.6.2.8.1) and FITC (1:5, clone P3.6.2.8.1) (eBioscience) were used as controls to compare the expression of each marker. Data were analyzed using the Flowjo software (BD Biosciences).

**NF-κB reporter assay**. Human embryonic kidney reporter cell lines (HEK-Blue™ hTLR2 and HEK-Blue™ Null2 cells; $5 \times 10^4$ cells/well) were infected or treated as described in previous sections in a 96-well flat bottom plate for 24 h at 37 °C, 5% CO2 atmosphere. To test if endocytosis is required for NF-κB stimulation, wortmannin (Sigma Aldrich), pitstop 2 (Sigma Aldrich), dynasore (Sigma Aldrich), chloroquine (Sigma Aldrich), chlorpromazine (Sigma Aldrich) or ammonium chloride (NH4Cl, Merck) were added 1 h before infection or treatment with the agonist. After 24 h post infection or treatment, analysis of NF-κB stimulation was assessed by mixing the supernatants with QUANTI-Blue™ (InvivoGen). Absorbance was measured at 630 nm using Synergy™ HT multi-mode microplate reader (BIOTEK). PAM3CSK4 (25 ng/mL) and PAM2CSK4 (1 ng/mL) were used as positive controls for hTLR2 cells, TNF-α (50 ng/mL, InvivoGen) for positive control in Null2 cells, and LPS (10 ng/mL) as a positive control for hTLR4 cells.

**In vitro infection frequency and titration analyses**. DENV infection was measured by intracellular detection of the envelope E virus protein. Briefly, cells were stained intracellularly using a mouse anti-DENV E antibody (1:800, clone D1-4G2-4-15, Millipore, #MAB10216) followed by rabbit anti-mouse IgG-coupled to AF647 (1:2000, ThermoFisher, #A-21446). Unstained cells, mock-infected cells plus secondary antibody, mock-infected plus detection pair and infected cells plus secondary antibody were included as controls in every experiment. UV-DENV was used as a control in PBMCs. Samples were measured on a FACSverse cytometer (BD Biosciences) and data were analyzed with Flowjo (BD Biosciences). For viral production, supernatants from infected HEK-Blue™ hTLR2 and human PBMC's were analyzed by plaque assay and RT-qPCR as previously described[82,83].

**Endothelial cell activation assay**. HUVEC were incubated with cell-free supernatants from infected and/or pre-treated PBMCs. In addition, direct DENV2 infection, exposure to UV-DENV2, RPMI media and TLR2/4 agonists PAM3CSK4 and LPS, served as controls. After 6 h of stimulation, cells were either lysed and stored at −20 °C for gene expression analysis or used directly for the expression analysis of activation markers using FACS. For FACS, cells were stained with the following antibodies CD62E E-selectin PE (1:100, clone HCD62E, #322606), CD106 VCAM-1 APC (1:100, clone STA, #305810) and CD54 ICAM-1 FITC (1:100, clone HCD54, #322720). Isotype-matched controls labelled with PE (1:100, clone RMG2a-62, #407107), APC (1:100, clone RMG1-1, #406609) and FITC (1:100, clone RMG1-1, #406605) were used as negative controls for comparing E-selectin, VCAM-1 and ICAM-1 expression, respectively. All surface staining antibodies were from BioLegend. Data were collected using a MACSQuant cytometer (Miltenyi Biotec) and analyzed using the Flowjo software (BD Biosciences). For gene expression analysis, total RNA was isolated from HUVEC using a RNeasy Plus Mini Kit (Qiagen, Leusden, The Netherlands), according to the manufacturer's instructions. RNA integrity was checked, cDNA synthesized and RT-qPCR performed using the ViiA 7 system (Applied Biosystems/ThermoFisher Scientific) as previously described[85]. Assay on demand primers were from Applied Biosystems (Nieuwerkerk aan de IJssel, The Netherlands) included *GADPH* (*Glyceraldehyde-3-phosphate dehydrogenase*, Hs99999905_m1), *E-selectin* (Hs00174057_m1), *VCAM-1* (*vascular cell adhesion molecule-1*, Hs00365486_m1), *ICAM-1* (*intracellular adhesion molecule-1*, Hs00164932_m1), *IL-6* (*Interleukin-6*, Hs00174131_m1), *IL-8* (*Interleukin-8*, Hs00174103_m1), *MCP-1* (*Monocyte chemotactic protein-1*, Hs00234140_m1), *CXCL6* (*C-X-C Motif Chemokine Ligand 6*, Hs00605742_g1). Duplicate real-time PCR analyses were performed for each sample, and the obtained threshold cycle (CT) values were averaged. Gene expression was normalized to the expression of housekeeping gene (*GAPDH*) resulting in the ΔCT value. The relative mRNA level was calculated by 2−ΔCT.

**Cytokine and chemokine determination**. Protein levels of IL-1β, TNF-α, IL-6, IL-8, IL-10, IL-12p70, IP-10, GM-CSF, IFN-α2, IFN-β, IFN-γ, IFN-λ1, and IFN-λ2/3 were determined in cell-free supernatants using the human anti-virus response panel (13-plex, LEGENDplex™, BioLegend, #740390). Data were collected using a FACSverse flow cytometer (BD Biosciences) and analyzed using LEGENDplex™ v8.0 (BioLegend).

**Statistical analysis**. Data analysis was performed using Prism 6.01 (Graphpad, USA). For patients' data analysis two-tailed Mann–Whitney test was used to compare the data between groups where a $p < 0.05$ was considered significant. Unless indicated otherwise, data are shown as media ± SD or SEM. One-way Anova followed by Dunnet's post hoc test was used for comparisons vs experimental controls. Paired one-tailed Student's *t*-test, unpaired one-tailed Student's *t*-test or two-tailed Mann–Whitney test were used to determine statistical significance of single experimental results. Correlations were determined by Pearson or Spearman

analysis. In all tests, values of $*p < 0.05$, $**p < 0.01$, $***p < 0.001$, and $****p < 0.0001$ were considered significant.

**Reporting summary**. Further information on research design is available in the Nature Research Reporting Summary linked to this article.

## Data availability

All data generated and analyzed in this study are included in the manuscript and its Supplementary Information files, or are available from the authors upon request. The source data underlying Figs. 1a–e, 2a–g, 3a–d, 4b–d, 5 and Supplementary Figs. 3, 4a, b, 5a–d, 6, 8a, b, 9a–d, 10a, b, 12a–f, 13, 14b, 15a–d, 17a–f, 18–20 are provided as a Source Data file.

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

## Acknowledgements
We thank Alicia Borneman and Heidi Ende-Metselaar, for their excellent technical assistance. We are grateful to Geert Mesander (Flow Cytometry Unit, UMCG) for his assistance and guidance in the flow cytometry assays. J.A.A.B. was supported by CONACYT, Mexico. I.A.R.Z. was supported by NWO-Veni grant and the Research Grant 2019 from the European Society of Clinical Microbiology and Infectious Diseases (ESCMID). T.C. and V.U. were supported by the Institut Pasteur International Network. Funding agencies had no role in the experimental design, decision to publish, or preparation of the paper.

## Author contributions
J.A.A.B., J.M, MRS, R.H, T.C., and I.A.R.Z. conceived and designed the experiments. J.A.A.B., V.U., B.M.tE, M.P, S.H., D.L, R.C., P.D. performed the experiments. T.C., P.D., J.M., V.U., T.C., and J.M.S. reviewed and helped writing the manuscript. J.A.A.B., V.U., P.D., T.C., J.M.S., and I.A.R.Z. interpreted the data and jointly wrote the first draft of the paper. J.A.A.B., V.U., T.C., I.A.R.Z. wrote the subsequent versions of the paper.

## Competing interests
The authors declare no competing interests.
