## [Peer Review File · Nature Communications]

Reviewers' Comments:

Reviewer #1:

Remarks to the Author:

In the manuscript by Aguilar-Briseño et al. the authors investigate the role of the immune receptors TLR2 together with TLR6 and CD14 in the pathogenesis associated with dengue virus (DENV) infections. They propose that TLR2 serves as an immune sensor for DENV particles and mediates inflammation caused during infection, which they found to be induced via NF- κ B activation. The study finds that expression of TLR2 in classical monocytes correlated with severity of severity of disease in a cohort of DENV children in Cambodia. Importantly, validation experiments in monocytes from healthy donors and infected with DENV ex vivo show that blocking TLR2 expression abrogated NF- κ B activation. Moreover, the study also shows that blockage of TLR in PBMCs prior to ex vivo infection with DENV also blocks the activation of vascular endothelial cells HUVEC, suggesting that TLR2 activation by DENV influences development of vascular permeability.

The study is very well done and uses very valuable clinical samples of children infected with DENV showing different degrees of disease severity. The study also shows that all 4 DENV serotypes are capable of inducing activation of monocytes via TLR2, although DENV shows a weaker effect. On the other hand, related flaviviruses, such as ZIKV and WNV and alphaviruses like CHIKV did not show this effect. The study is important and sheds light on possible mechanisms that mediate severe symptoms of DENV infection, such as Dengue Hemorrhagic Fever (DHF) and suggests the use of TLR2 blocking agents to mitigate those severe symptoms in patients.

The study shows that there is a correlation of TLR2 expression in cells infected with DENV and severity of disease. Although this is important and relevant to understand the factors that determine the development of DHF/DSS over DF, there are some remaining issues that should be addressed to strengthen the data and to provide further proof of the involvement of TLR2 in severe dengue disease manifestations.

- Specific comments

The main outstanding question related to this study is the identification of the ligands for TLR2 in the DENV virion. It would be important to know if there is a protein or sugar moiety present in viral proteins present in the virion that is recognized and bound to TLR2. It seems like only viruses grown in mosquito cells are the ones capable of inducing the activation of TLR2 (WNV, ZIKV, CHIKV). This suggests that activation of TLR could be due to different glycosylation of viral proteins in mosquito or mammalian cells. This point should be addressed.

The authors already acknowledge the fact that the study is performed in patients that have pre-existing immunity to DENV and they are not addressing the role of antibody enhancement of disease (ADE). They also perform all the experiments in

PBMCs, which represent the viremic stage of the disease. Have they tested also the effects of DENV infection in the upregulation/activation of TLR2 in other cells that are also susceptible to DENV infection, such as macrophages and dendritic cells? In Table 2, is the weak activation of TLR2 indicated for DENV-4 viruses due to weak binding or different levels of infection observed with that serotype? Also, if titrations of DENV were performed by plaque assays in BHK cells as indicated in the manuscript, it may have underestimated the number of infectious particles in DENV4 preparations, which are normally less capable of inducing plaques in that system.

The labeling of some of the figures is very small and hard to read.

Reviewer #2:

Remarks to the Author:

The authors observed an upregulation of expression of TLR2 in specimen drawn from 48 PCR positive out of 54 dengue virus and 40 out of these also other (unspecified) causative infected infant patients 32 of which were dengue virus protein NS1 positive. Also according to table 1., 32 patients suffered dengue fever, 12 from dengue hemorrhagic fever, and 10 even shock syndrome. Rather than a nucleic acid sensor typically implicated in immunological virus sensing, the expression level of the bacterial product sensor TLR2 on monocytes was altered in a disease severity dependent manner. Like the bona fide TLR2 agonist synthetic lipopeptide, numerous dengue strain virion challenges activated overexpressed and endogenous TLR2 in a fashion that was also CD14 and TLR1, yet not TLR4 and TLR6 dependent. According to statements of the authors, TLR2 upregulation of TLR2 upon dengue virus infection as well as virus protein driven TLR2 and TLR4 activation have been both implicated and debated previously already.

The authors observed increased anti TLR2 antibody and anti virus antigen binding by specific monocyte subpopulations. They also observed activation of HEK cells overexpressing TLR2 by virus #2 preparations, which was blockable not merely with anti TLR2 but also TLR1 and CD14 antibodies. In contrast, TLR4+ HEK cells failed to respond to challenge not merely with #2 virus, but also #1 and -4 variants. Some endocytosis inhibitor treatment impacted on dengue virus and overexpressed TLR2 driven cell activation as well as viability while others did not. Some treatments with antibodies of different specificities also changed properties of different monocyte classes in respect to TLR2 expression. Anti TLR2 treatments of PBMCs also reduced the capacities of their supernatants to induce specific cytokine and chemokine production by HUVEC cells. Some effects were dependent on virus viability whereas others were not. However, not all virus major strains 1, -3 and -4 were analyzed consequently in that an experimental focus was on -2 for a seemingly non-specified reason.

"DENV1 (strains 16007, Hawaii), DENV2 (strains 16681, NGC, TSV01), DENV3 (strains H87, 16562) and DENV4 (strains 1036, H241), were produced in the Aedes albopictus C6/36 cell line as described before (69)." and "CHIKV (La Reunion OPY1) was a gift from A. Merits (University of Tartu, Estonia), and was produced from infectious cDNA clones and passaged twice in Vero E6 cells (71). ZIKV (SL0216) was a gift from M.J. van Hemert (LUMC, the Netherlands); virus was produced in Vero E6 cells .." which largely conceptually excludes microbial such as bacterial product contamination which is an issue important particularly in cases such as novel implications of pattern recognitions. It (e.g. through mycoplasmal infection) must be taken into consideration here because according to table 2 some virus strain preps were devoid of TLR2 activating capacity. Moreover, a novelty of the implication of TLR2 as dengue virus sensor basing on TLR2 upregulation is being put into relation by citation of previous studies also implying TLR4. The current manuscript makes an impression as if it tends to overcome these aspects by blockade and a multitude of expression analyses rather than coherent experimentation towards bringing up further functional evidence. For instance, why some strains do not activate TLR2 and the identity of the mediators of HUVEC activations upon PBMC infections are aspects deserving more rigorous clarification. The authors shall thus revise their manuscript potentially towards a disadvantage of some of somewhat multitude expression analyses and for the advantage of a bringing up of more substantial functional data.

1. Statements such as ".. as dengue or dengue fever" should be specified.
2. Wordings such as "as TLR2 blockade did not restore monocyte frequencies to the mock levels ", "IL-1 β and TNF-alpha has been reported to be present in the plasma of DENV patients", or "This seemingly contrasting role of TLR2 on these monocytes are likely to be linked to distinct phenotype and functions of CM and IM" should be revised for the sake of further increased clarity and informativeness.
3. Abbreviations and special wordings used in figures such as PS and W or DF, DHF, DSS, IM, NM, IQF, DENV GEC, and "stratified by disease severity" should be worded and/or explained in legends. Also "MOG" and legend statements such as ".. hTLR4 cells were (mock)- treated for 24h with purified (p) DENV1 (strain 16007), DENV2 (strain 16681), DENV4 (strain 1036) (MOG3500, MOG10000) or the respective standard (non-purified) preparations (MOG3500), n=2. NF-kB stimulation was assessed .." should be clarified such as by removal of redundant terms such as in respect to "MOG3500" to enable understanding by non expert readers.
4. Controls lacking should be included:
 - Healthy control samples should be analyzed to indicate their respective expressions of antigens analyzed.
 - 1a virus producing cell supernatant and lysate, mock control virus preparation, were the virus producing cells (e.g. mycoplasma) bacteria (e.g. mycoplasma) free?

- 1b activation independent of TLR2 to demonstrate specificity, TNF is mentioned in mat. and meth. yet respective data illustration seem to be missing. Also, "null" (TLR2-) cells should be used in parallel rather than punctually. Anti TLR2 should be applied to TLR4+ HEK cells challenged with LPS to further demonstrate specificity.
 - 1cde control is also needed for CD14 (involvement of which in TLR2 action is not comprehensively uncontroversial) such as TNF, as well as anti TLR1/6 such as by application of the antibodies to TLR4+ HEK cells prior to LPS challenge.
 - Also, is dynasore (Sigma Aldrich), chloroquine (Sigma Aldrich), and chlorpromazine (Sigma Aldrich) just not shown in Fig 2d? According to a common view, TLR2 is expressed on the cell surface for which reason its activation is principally endocytosis independent.
 - The actual isotype of "Isotype-matched antibody labelled with PE (clone MOPC 173) (BioLegend) was used as negative control for comparing TLR2 expression" shall be named and controls be illustrated to indicate that not increased Fc receptor expression is causing increased antibody binding. Also, a diagram indicating often marginal surface TLR2 staining even of monocytes or macrophages should be displayed.
5. Also, other stainings such as that for "NS3" which is being implicated as indicative of virus replication by "Notably, CM and IM, but not NM, were the primary targets of DENV replication (Fig. 1C) and CM predominated in patients that developed severe disease ($p < 0.5$) (Fig. 1D)." should be isotype controlled. Furthermore, the authors should outline why virus antigen positivity not merely represents endocytosis or other cellular uptake of virus rather than replication of it. Application of k.o. mice or cells such as those lacking TLRs and/or CD14 implicated would be highly supportive for the manuscript.
6. The subsequent term "Altogether, these data imply that TLR2 on classical monocytes plays a significant role in disease development." is an overstatement. The data provided by figure 1 rather indicate coincidence, in numerous and not individually indicated cases potentially also with other unspecified infection, of higher TLR2 and virus antigen presence in specified monocytes. Also, why 4G2 was used to indicate infection alternatively seems to have left unexplained. The authors should revise accordingly.
7. CD14, TLR1 and TLR6 expression by HEK cells should be demonstrated in order to justify blockades of both of the two antigens.
8. "All DENV serotypes are sensed by TLR2" seems to collide with "-" (minus) occurring in table 2 for some strains of dengue virus. The authors should revise accordingly.
9. Figure 3 seems to lack subfigure "c" representation in legend. Also, the validity of inter subfigure diagram legends is not specified to degree that would allow for fast understanding. Whether control challenge such as with LPS or other virus used in other experiments throughout the manuscript would have effects resembling those of dengue virus should be illustrated. Not necessarily "must" TLR2 expression

level upregulation correlate with TLR2 recruitment and activity.

10. Besides DENV2, also -1 and -4 should be analyzed to support generality of the conclusions drawn.

11. PBMCs treated with anti TLR2 antibody and infected with virus should be washed after the challenges and prior to transfer of their supernatants to HUVECs to exclude carry over of both of the molecular entities, antibodies and virus to the secondary cell culture. Again, control antibody, such as isotype control or related specificity one application appear as demandable.

12. Figure s8 lacks not merely non-anti TLR2 treated controls, but also plausibility since TLR2 activation towards TNF production with lipopeptide failed to be blockable. The authors should address both aspects by respective data supplementation.

13. Whether TLR2 non-activating dengue strains also increase TLR2 expression levels and whether their pathogenicity differs from that of activating counterparts both qualitatively and quantitatively should be indicated more rigorously by the manuscript.

Reviewer #3:

Remarks to the Author:

Aguilar-Briseño et al. reported Toll-like receptor 2 senses dengue virions and mediates the innate inflammatory response underlying disease pathogenesis. Their results suggested that the role of TLR2 plays a significant role in mediating the immunopathology, and proposed that drugs targeting on TLR2 may have benefit the patients in severe conditions. How does dengue virus induce severe disease in certain subjects remain an enigma despite many decade intensive research efforts? There are many factors derived from innate immune cells been linked to or correlated plasma leakage in severe patients. The current report attempted to provide in vitro evidence with PBMC isolated from acute dengue patients to demonstrate that TLR2 in innate phagocytes cells play significant role in severe dengue.

The major issue with the current study is that the virus utilized was from in vitro and was derived from C6/36 cells. Does NS1 in supernatant in dengue virus infected C3/36 cells? If yes, the authors should check the genetic materials of C3/36 cells to ensure that there is no contamination of mammalian Vero cells 1. Reality is that the virus circulating in acute patients is from infected human beings and the host cells. Interestingly, it has been demonstrated that the viral morphology and property obtained from acute dengue patients are significantly different from that of in vitro virus 2-5. Since the authors had recruited acute dengue patients, the dengue virus should be easily obtained from these specimens and repeat the experiments.

The following points are additional concerns:

1. In Table 1: The individual viral titers in these subjects, presented as PFU/ml of blood, and the day of onset fever should be shown. Parameters collected from dengue patients are easily affected by fever days. In addition, will the viral titers correlate with the claim that TLR2 expressed more in CM and IM, based upon the day of fever?

2. Figure 1C, it is insufficient to claim that NS3+ is a surrogate marker for viral replication or infection. The figure should be shown according to day of fever. It is likely that these NS3+ cells are a result of phagocytic effect. To claim the infection, the authors should sort out these cells and perform the co-culture to recover the infectious dengue virus.

3. Time-dependent is critical parameter in dengue, especially for those subjects with plasma leakage. All figures associated with dengue patients should be shown kinetically.

1. Anderson, R., King, A.D. & Innis, B.L. Correlation of E protein binding with cell susceptibility to dengue 4 virus infection. *J Gen Virol* 73 (Pt 8), 2155-2159 (1992).

2. Wang, S., He, R. & Anderson, R. PrM- and cell-binding domains of the dengue virus E protein. *J Virol* 73, 2547-2551 (1999).

3. Chaichana, P., et al. Low levels of antibody-dependent enhancement in vitro using viruses and plasma from dengue patients. *PLoS One* 9, e92173 (2014).

4. Sirivichayakul, C., Sabchareon, A., Limkittikul, K. & Yoksan, S. Plaque reduction neutralization antibody test does not accurately predict protection against dengue infection in Ratchaburi cohort, Thailand. *Virology* 11, 48 (2014).

5. Hsu, A.Y., et al. Infectious dengue vesicles derived from CD61+ cells in acute patient plasma exhibited a diaphanous appearance. *Scientific reports* 5, 17990 (2015).

6. King, A.D., et al. B cells are the principal circulating mononuclear cells infected by dengue virus. *Southeast Asian J Trop Med Public Health* 30, 718-728 (1999).

Reviewers' comments:

Reviewer #1 (Remarks to the Author):

In the manuscript by Aguilar-Briseño et al. the authors investigate the role of the immune receptors TLR2 together with TLR6 and CD14 in the pathogenesis associated with dengue virus (DENV) infections. They propose that TLR2 serves as an immune sensor for DENV particles and mediates inflammation caused during infection, which they found to be induced via NF- κ B activation. The study finds that expression of TLR2 in classical monocytes correlated with severity of disease in a cohort of DENV children in Cambodia. Importantly, validation experiments in monocytes from healthy donors and infected with DENV *ex vivo* show that blocking TLR2 expression abrogated NF- κ B activation. Moreover, the study also shows that blockage of TLR in PBMCs prior to *ex vivo* infection with DENV also blocks the activation of vascular endothelial cells HUVEC, suggesting that TLR2 activation by DENV influences development of vascular permeability.

The study is very well done and uses very valuable clinical samples of children infected with DENV showing different degrees of disease severity. The study also shows that all 4 DENV serotypes are capable of inducing activation of monocytes via TLR2, although DENV shows a weaker effect. On the other hand, related flaviviruses, such as ZIKV and WNV and alphaviruses like CHIKV did not show this effect. The study is important and sheds light on possible mechanisms that mediate severe symptoms of DENV infection, such as Dengue Hemorrhagic Fever (DHF) and suggests the use of TLR2 blocking agents to mitigate those severe symptoms in patients.

The study shows that there is a correlation of TLR2 expression in cells infected with DENV and severity of disease. Although this is important and relevant to understand the factors that determine the development of DHF/DSS over DF, there are some remaining issues that should be addressed to strengthen the data and to provide further proof of the involvement of TLR2 in severe dengue disease manifestations.

- Specific comments

The main outstanding question related to this study is the identification of the ligands for TLR2 in the DENV virion. It would be important to know if there is a protein or sugar moiety present in viral proteins present in the virion that is recognized and bound to TLR2. It seems like only viruses grown in mosquito cells are the ones capable of inducing the activation of TLR2 (WNV, ZIKV, CHIKV). This suggests that activation of TLR could be due to different glycosylation of viral proteins in mosquito or mammalian cells. This point should be addressed.

The authors already acknowledge the fact that the study is performed in patients that have pre-existing immunity to DENV and they are not addressing the role of antibody enhancement of disease (ADE). They also perform all the experiments in PBMCs, which represent the viremic stage of the disease. Have they tested also the effects of DENV infection in the upregulation/activation of TLR2 in other cells that are also susceptible to DENV infection, such as macrophages and dendritic cells?

In Table 2, is the weak activation of TLR2 indicated for DENV-4 viruses due to weak binding or different levels of infection observed with that serotype? Also, if titrations of DENV were performed by plaque

assays in BHK cells as indicated in the manuscript, it may have underestimated the number of infectious particles in DENV4 preparations, which are normally less capable of inducing plaques in that system.

The labeling of some of the figures is very small and hard to read.

Reviewer #2 (Remarks to the Author):

The authors observed an upregulation of expression of TLR2 in specimen drawn from 48 PCR positive out of 54 dengue virus and 40 out of these also other (unspecified) causative infected infant patients 32 of which were dengue virus protein NS1 positive. Also according to table 1., 32 patients suffered dengue fever, 12 from dengue hemorrhagic fever, and 10 even shock syndrome. Rather than a nucleic acid sensor typically implicated in immunological virus sensing, the expression level of the bacterial product sensor TLR2 on monocytes was altered in a disease severity dependent manner. Like the bona fide TLR2 agonist synthetic lipopeptide, numerous dengue strain virion challenges activated overexpressed and endogenous TLR2 in a fashion that was also CD14 and TLR1, yet not TLR4 and TLR6 dependent. According to statements of the authors, TLR2 upregulation of TLR2 upon dengue virus infection as well as virus protein driven TLR2 and TLR4 activation have been both implicated and debated previously already.

The authors observed increased anti TLR2 antibody and anti virus antigen binding by specific monocyte subpopulations. They also observed activation of HEK cells overexpressing TLR2 by virus #2 preparations, which was blockable not merely with anti TLR2 but also TLR1 and CD14 antibodies. In contrast, TLR4+ HEK cells failed to respond to challenge not merely with #2 virus, but also #1 and -4 variants. Some endocytosis inhibitor treatment impacted on dengue virus and overexpressed TLR2 driven cell activation as well as viability while others did not. Some treatments with antibodies of different specificities also changed properties of different monocyte classes in respect to TLR2 expression. Anti TLR2 treatments of PBMCs also reduced the capacities of their supernatants to induce specific cytokine and chemokine production by HUVEC cells. Some effects were dependent on virus viability whereas others were not. However, not all virus major strains 1, -3 and -4 were analyzed consequently in that an experimental focus was on -2 for a seemingly non-specified reason.

“DENV1 (strains 16007, Hawaii), DENV2 (strains 16681, NGC, TSV01), DENV3 (strains H87, 16562) and DENV4 (strains 1036, H241), were produced in the *Aedes albopictus* C6/36 cell line as described before (69).” and “CHIKV (La Reunion OPY1) was a gift from A. Merits (University of Tartu, Estonia), and was produced from infectious cDNA clones and passaged twice in Vero E6 cells (71). ZIKV (SL0216) was a gift from M.J. van Hemert (LUMC, the Netherlands); virus was produced in Vero E6 cells ..” which largely conceptually excludes microbial such as bacterial product contamination which is an issue important particularly in cases such as novel implications of pattern recognitions. It (e.g. through mycoplasmal infection) must be taken into consideration here because according to table 2 some virus strain preps were devoid of TLR2 activating capacity. Moreover, a novelty of the implication of TLR2 as dengue virus sensor basing on TLR2 upregulation is being put into relation by

citation of previous studies also implying TLR4. The current manuscript makes an impression as if it tends to overcome these aspects by blockade and a multitude of expression analyses rather than coherent experimentation towards bringing up further functional evidence. For instance, why some strains do not activate TLR2 and the identity of the mediators of HUVEC activations upon PBMC infections are aspects deserving more rigorous clarification. The authors shall thus revise their manuscript potentially towards a disadvantage of some of somewhat multitude expression analyses and for the advantage of a bringing up of more substantial functional data.

1. Statements such as “.. as dengue or dengue fever” should be specified.

2. Wordings such as “as TLR2 blockade did not restore monocyte frequencies to the mock levels “, “IL-1 β and TNF-alpha has been reported to be present in the plasma of DENV patients”, or “This seemingly contrasting role of TLR2 on these monocytes are likely to be linked to distinct phenotype and functions of CM and IM” should be revised for the sake of further increased clarity and informativeness.

3. Abbreviations and special wordings used in figures such as PS and W or DF, DHF, DSS, IM, NM, IQF, DENV GEc, and “stratified by disease severity” should be worded and/or explained in legends. Also “MOG” and legend statements such as “.. hTLR4 cells were (mock)- treated for 24h with purified (p) DENV1 (strain 16007), DENV2 (strain 16681), DENV4 (strain 1036) (MOG3500, MOG10000) or the respective standard (non-purified) preparations (MOG3500), n=2. NF-kB stimulation was assessed ..” should be clarified such as by removal of redundant terms such as in respect to “MOG3500” to enable understanding by non expert readers.

4. Controls lacking should be included:

- Healthy control samples should be analyzed to indicate their respective expressions of antigens analyzed.

- 1a virus producing cell supernatant and lysate, mock control virus preparation, were the virus producing cells (e.g. mycoplasma) bacteria (e.g. mycoplasma) free?

- 1b activation independent of TLR2 to demonstrate specificity, TNF is mentioned in mat. and meth. yet respective data illustration seem to be missing. Also, “null” (TLR2-) cells should be used in parallel rather than punctually. Anti TLR2 should be applied to TLR4+ HEK cells challenged with LPS to further demonstrate specificity.

- 1cde control is also needed for CD14 (involvement of which in TLR2 action is not comprehensively uncontroversial) such as TNF, as well as anti TLR1/6 such as by application of the antibodies to TLR4+ HEK cells prior to LPS challenge.

- Also, is dynasore (Sigma Aldrich), chloroquine (Sigma Aldrich), and chlorpromazine (Sigma Aldrich) just not shown in Fig 2d? According to a common view, TLR2 is expressed on the cell surface for which reason its activation is principally endocytosis independent.

- The actual isotype of “Isotype-matched antibody labelled with PE (clone MOPC 173) (BioLegend) was used as negative control for comparing TLR2 expression” shall be named and controls be illustrated to indicate that not increased Fc receptor expression is causing increased antibody binding. Also, a diagram indicating often marginal surface TLR2 staining even of monocytes or macrophages should be displayed.

5. Also, other stainings such as that for “NS3” which is being implicated as indicative of virus replication by “Notably, CM and IM, but not NM, were the primary targets of DENV replication (Fig. 1C) and CM predominated in patients that developed severe disease ($p < 0.5$) (Fig. 1D).” should be isotype controlled. Furthermore, the authors should outline why virus antigen positivity not merely represents endocytosis or other cellular uptake of virus rather than replication of it. Application of k.o. mice or cells such as those lacking TLRs and/or CD14 implicated would be highly supportive for the manuscript.
6. The subsequent term “Altogether, these data imply that TLR2 on classical monocytes plays a significant role in disease development.” is an overstatement. The data provided by figure 1 rather indicate coincidence, in numerous and not individually indicated cases potentially also with other unspecified infection, of higher TLR2 and virus antigen presence in specified monocytes. Also, why 4G2 was used to indicate infection alternatively seems to have left unexplained. The authors should revise accordingly.
7. CD14, TLR1 and TLR6 expression by HEK cells should be demonstrated in order to justify blockades of both of the two antigens.
8. “All DENV serotypes are sensed by TLR2” seems to collide with “-“ (minus) occurring in table 2 for some strains of dengue virus. The authors should revise accordingly.
9. Figure 3 seems to lack subfigure “c” representation in legend. Also, the validity of inter subfigure diagram legends is not specified to degree that would allow for fast understanding. Whether control challenge such as with LPS or other virus used in other experiments throughout the manuscript would have effects resembling those of dengue virus should be illustrated. Not necessarily “must” TLR2 expression level upregulation correlate with TLR2 recruitment and activity.
10. Besides DENV2, also -1 and -4 should be analyzed to support generality of the conclusions drawn.
11. PBMCs treated with anti TLR2 antibody and infected with virus should be washed after the challenges and prior to transfer of their supernatants to HUVECs to exclude carry over of both of the molecular entities, antibodies and virus to the secondary cell culture. Again, control antibody, such as isotype control or related specificity one application appear as demandable.
12. Figure s8 lacks not merely non-anti TLR2 treated controls, but also plausibility since TLR2 activation towards TNF production with lipopeptide failed to be blockable. The authors should address both aspects by respective data supplementation.
13. Whether TLR2 non-activating dengue strains also increase TLR2 expression levels and whether their pathogenicity differs from that of activating counterparts both qualitatively and quantitatively should be indicated more rigorously by the manuscript.

Reviewer #3 (Remarks to the Author):

Aguilar-Briseño et al. reported Toll-like receptor 2 senses dengue virions and mediates the innate inflammatory response underlying disease pathogenesis. Their results suggested that the role of TLR2 plays a significant role in mediating the immunopathology, and proposed that drugs targeting on TLR2 may have benefit the patients in severe conditions. How does dengue virus induce severe disease in certain

subjects remain an enigma despite many decade intensive research efforts? There are many factors derived from innate immune cells been linked to or correlated plasma leakage in server patients. The current report attempted to provide in vitro evidence with PBMC isolated from acute dengue patients to demonstrate that TLR2 in innate phagocytes cells play significant role in severe dengue.

The major issue with the current study is that the virus utilized was from in vitro and was derived from C6/36 cells. Does NS1 in supernatant in dengue virus infected C3/36 cells? If yes, the authors should check the genetic materials of C3/36 cells to ensure that there is no contamination of mammalian Vero cells 1. Reality is that the virus circulating in acute patients is from infected human beings and the host cells. Interestingly, it has been demonstrated that the viral morphology and property obtained from acute dengue patients are significantly different from that of in vitro virus 2-5. Since the authors had recruited acute dengue patients, the dengue virus should be easily obtained from these specimens and repeat the experiments.

The following points are additional concerns:

1. In Table 1: The individual viral titers in these subjects, presented as PFU/ml of blood, and the day of onset fever should be shown. Parameters collected from dengue patients are easily affected by fever days. In addition, will the viral titers correlate with the claim that TLR2 expressed more in CM and IM, based upon the day of fever?
2. Figure 1C, it is insufficient to claim that NS3+ is a surrogate marker for viral replication or infection. The figure should be shown according to day of fever. It is likely that these NS3+ cells are a result of phagocytic effect 6. To claim the infection, the authors should sort out these cells and perform the co-culture to recover the infectious dengue virus.
3. Time-dependent is critical parameter in dengue, especially for those subjects with plasma leakage. All figures associated with dengue patients should be shown kinetically.

1. Anderson, R., King, A.D. & Innis, B.L. Correlation of E protein binding with cell susceptibility to dengue 4 virus infection. *J Gen Virol* 73 (Pt 8), 2155-2159 (1992).
2. Wang, S., He, R. & Anderson, R. PrM- and cell-binding domains of the dengue virus E protein. *J Virol* 73, 2547-2551 (1999).
3. Chaichana, P., et al. Low levels of antibody-dependent enhancement in vitro using viruses and plasma from dengue patients. *PLoS One* 9, e92173 (2014).
4. Sirivichayakul, C., Sabchareon, A., Limkittikul, K. & Yoksan, S. Plaque reduction neutralization antibody test does not accurately predict protection against dengue infection in Ratchaburi cohort, Thailand. *Virol J* 11, 48 (2014).
5. Hsu, A.Y., et al. Infectious dengue vesicles derived from CD61+ cells in acute patient plasma exhibited a diaphanous appearance. *Scientific reports* 5, 17990 (2015).

6. King, A.D., et al. B cells are the principal circulating mononuclear cells infected by dengue virus. *Southeast Asian J Trop Med Public Health* 30, 718-728 (1999).

Reviewer #1

Overall summary

1.1 In the manuscript by Aguilar-Briseño et al. the authors investigate the role of the immune receptors TLR2 together with TLR6 and CD14 in the pathogenesis associated with dengue virus (DENV) infections. They propose that TLR2 serves as an immune sensor for DENV particles and mediates inflammation caused during infection, which they found to be induced via NF- κ B activation. The study finds that expression of TLR2 in classical monocytes correlated with severity of disease in a cohort of DENV children in Cambodia. Importantly, validation experiments in monocytes from healthy donors and infected with DENV *ex vivo* show that blocking TLR2 expression abrogated NF- κ B activation. Moreover, the study also shows that blockage of TLR in PBMCs prior to *ex vivo* infection with DENV also blocks the activation of vascular endothelial cells HUVEC, suggesting that TLR2 activation by DENV influences development of vascular permeability.

The study is very well done and uses very valuable clinical samples of children infected with DENV showing different degrees of disease severity. The study also shows that all 4 DENV serotypes are capable of inducing activation of monocytes via TLR2, although DENV shows a weaker effect. On the other hand, related flaviviruses, such as ZIKV and WNV and alphaviruses like CHIKV did not show this effect. The study is important and sheds light on possible mechanisms that mediate severe symptoms of DENV infection, such as Dengue Hemorrhagic Fever (DHF) and suggests the use of TLR2 blocking agents to mitigate those severe symptoms in patients.

The study shows that there is a correlation of TLR2 expression in cells infected with DENV and severity of disease. Although this is important and relevant to understand the factors that determine the development of DHF/DSS over DF, there are some remaining issues that should be addressed to strengthen the data and to provide further proof of the involvement of TLR2 in severe dengue disease manifestations.

Response:

We appreciate the reviewer's recognition of the quality and comprehensiveness of the study. Considering the seemingly dual role of TLR2 in disease pathogenesis, its role as a dengue sensor as well as a contributor to immunopathology, it is indeed important to understand if differences between glycosylation patterns in mosquito and human cells underlie the ability of TLR2 to sense the mosquito-derived DENV. Therefore, in light of the editor and reviewers' comments, we have analyzed the ability of DENV derived from human target cells, such as dendritic cells and macrophages, to engage TLR2 axis and included these data in the revised manuscript. Please refer to comment **1.2** for a more detailed description of the additional analyses. We agree with the reviewer that the role of TLR2 on these cells might be important in course of DENV infection and is worth studying. Importantly, in our patient cohort we did not observe a significant TLR2 expression on cells others than monocytes. However, different tissue macrophages are known to express TLR2, which in the presence of different co-

receptors might convey various immune responses. We have addressed this suggestion of the reviewer in the comment **1.3**. Furthermore, we provide details on our experimental set up in a sensitive HEK-Blue system where we have addressed the ability of TLR2 to sense all DENV serotypes based on the number of genome-containing particles rather the infectious particles (please refer to comment **1.4** for a detailed clarification).

Specific comments

1.2 The main outstanding question related to this study is the identification of the ligands for TLR2 in the DENV virion. It would be important to know if there is a protein or sugar moiety present in viral proteins present in the virion that is recognized and bound to TLR2. It seems like only viruses grown in mosquito cells are the ones capable of inducing the activation of TLR2(WNV, ZIKV CHIKV). This suggests that activation of TLR could be due to different glycosylation of viral proteins in mosquito or mammalian cells. This point should be addressed.

Response:

We thank the reviewer for this relevant comment. We fully agree that the difference in glycosylation of viral glycoproteins between the virus produced in mammalian or mosquito's cells can have an impact on the activation of toll-like receptor 2 by these viruses. Therefore, to address this, we analyzed the ability of DENV-2 16681 produced in monocyte (Mo)-derived dendritic cells (DC's) and Mo-derived macrophages (M ϕ) to activate NF- κ B in the HEK-Blue hTLR2 cells. The viruses were obtained by infecting monocyte-derived DC's and monocyte-derived M ϕ with the mosquito C6/36 cells -derived virus for 2 hours, after which the cells were washed to remove the surplus of inoculating virus and incubation was continued until 48hpi. Following titrations of DC's-and M ϕ -derived DENV2 preparations, HEK-Blue™ hTLR2 cells were exposed to increasing number of virus particles [based on multiplicity of genomes (MOGs)] to ensure fair comparison; the C6/36-derived DENV served as a reference. Interestingly, as shown in the figure below, DC's-derived DENV2 induced NF- κ B activation on hTLR2 cells albeit approximately 3x lower than the C6/36-derived virus at a similar MOG, suggesting that differences in glycosylation pattern between human and mosquito-derived viruses indeed modulate the TLR2-engagement. Notably however, M ϕ -produced DENV2 did not induce NF- κ B activation at any of MOG tested, which implies that virus recognition by TLR2 is not solely modulated by differences between mosquito and mammalian cells but potentially also other intra host, cell-type/tissue-specific factors eg. cell-type specific glycosylation or efficiency of progeny virus maturation (Dejnirattisai et al., 2010; Parameswaran et al., 2017). It is important to mention however that due to the overall lower yield of DENV released by macrophages, MOG of 4000 could not be reached and thus more in-depth studies are needed to test the premise of intra-host cell type-specific factors influencing TLR2 activation by DENV. Noteworthy in that context are also the data summarized in Table 2 of the manuscript, showing the different capacity of various strains of DENV1-4 to activate TLR2 axis despite being produced in the mosquito C6/36 cells. All these

observations suggest that the activation of TLR2 by DENV particles is influenced by strain and cell type rather than solely by differences in glycosylation's patterns between the mosquito and human cells.

Panel B Fig. S1. Activation of HEK-Blue hTLR2 exposed to DENV2 derived from mosquito and human cells. The cells were exposed to increasing multiplicities of genome equivalent copies (MOG). C6/36 cells (MOG100, n=1; MOG500, n=1; MOG1000, n=3; MOG 3500, n=3), monocyte-derived dendritic cells (MOG250, n=1; MOG4000, n=2) and monocyte-derived macrophages for 24h (MOG250, n=1; MOG4000, n=2). NF-κB stimulation was assessed by QUANTI-Blue™, OD values show the induction of NF-κB. Data represent the mean ± SD. P values were obtained by paired one-tailed t test (*P<0.05, **** P<0.001)

Dejnirattisai, W., *et al.* Cross-Reacting Antibodies Enhance Dengue Virus Infection in Humans. *Science*. 328 (5979): 745-748 (2010)

Parameswaran, P., *et al.* Intrahost Selection Pressures Drive Rapid Dengue Virus Microevolution in Acute Human Infections. *Cell Host & Microbe*. 22: 400-410 (2017)

We have included the above figure as panel B in the Fig. S3 (page 10) and the corresponding text (here below in blue) in the results section on page 7.

Results: Considering the differential glycosylation patterns of the DENV surface proteins during replication in primary human and mosquito cells(33), we next compared the ability of DENV-2 16681 produced in mosquito C6/36 cells to activate NF-κB in the HEK-Blue hTLR2 cells with that produced on monocyte-derived dendritic cells (DC's) and Mo-derived macrophages (Mφ). The viruses were obtained by infecting monocyte-derived DC's and monocyte-derived Mφ with C6/36-derived virus for 2 hours, after which the cells were washed to remove the surplus of inoculating virus and incubation was continued until 48hpi. Following titrations of DC-and Mφ -derived DENV2 preparations, HEK-Blue™ hTLR2 cells were exposed to increasing numbers of virus particles (MOGs) to ensure fair comparison.

Interestingly, DC-derived DENV2 induced NF- κ B activation on hTLR2 cells albeit approximately 3x lower than the C6/36-derived virus at a similar MOG (Fig. S1B), suggesting that differential glycosylation of the virus envelope proteins in human and mosquito cells modulates the TLR2 recognition. Notably however, M ϕ -produced DENV2 did not induce NF- κ B activation at any of MOG tested, which suggest that TLR2 engagement is sensitive to changes in virion characteristics beyond the differences in glycosylation patterns between mosquito and human cells. It is important to mention however that due to the overall lower yield of DENV released by macrophages, MOG of 4000 could not be reached and thus more in-depth studies are needed to test the premise of intra-host cell type-specific factors influencing TLR2 activation by DENV.

1.3 The authors already acknowledge the fact that the study is performed in patients that have pre-existing immunity to DENV and they are not addressing the role of antibody enhancement of disease (ADE). They also perform all the experiments in PBMCs, which represent the viremic stage of the disease. Have they tested also the effects of DENV infection in the upregulation/activation of TLR2 in other cells that are also susceptible to DENV infection, such as macrophages and dendritic cells?

Response:

We agree with the reviewer that it will be important to evaluate the effect of DENV infection on TLR2 expression and its activity in other relevant target cells. In fact, as mentioned in the introduction on page 2, line 69, Torres *et al*, 2013 have observed an increased TLR2 expression on plasmacytoid dendritic cells but not myeloid dendritic cells of dengue virus infected patients. Importantly however, in our pediatric cohort, we did not observe a significant expression of TLR2 on cells other than monocytes and thus we focused on this cell type. Considering the inherent differences (age, ethnicity) between the cohorts analyzed in our study and that of Torres *et al*, we propose that new studies should address their relevance in a side-by-side scrutiny of demographically different cohorts.

Furthermore, we are currently analyzing the effect of DENV infection in the upregulation/activation of TLR2 in human Mo-derived M ϕ , however these data are a part of a different study and thus considered beyond the scope of this manuscript.

Torres, S., *et al*. Differential expression of toll-like receptors in dendritic cells of patients with dengue during early and late acute phases of the disease. *PLoS Pathog*. 11, e1005053 (2013)

1.4 In Table 2, is the weak activation of TLR2 indicated for DENV-4 viruses due to weak binding or different levels of infection observed with that serotype? Also, if titrations of DENV were performed by plaque assays in BHK cells as indicated in the manuscript, it may have underestimated the number of infectious particles in DENV4 preparations, which are normally less capable of inducing plaques in that system.

Response

We understand the concern of the reviewer and are pleased to address the point. As observed in Figure 2 (panel A), the ability of DENV to activate NF- κ B in HEK-Blue™ hTLR2 cells is independent of virus replication as UV-inactivated DENV particles were able to activate NF- κ B to the same extent of the replication competent DENV. Therefore, to ensure fair comparison, for all further experiments, including those in HEK cells, infections were performed based on the total number of DENV particles (equivalent copies/genome containing particles for purified preparations, MOGs) rather than infectious particles (multiplicity of infection, MOI).

1.5 The labeling of some of the figures is very small and hard to read.

Response

We apologize for the inconvenience. We have revised the font to increase the legibility of the figures by changing the font to Arial.

Reviewer #2

Overall summary

The authors observed an upregulation of expression of TLR2 in specimen drawn from 48 PCR positive out of 54 dengue virus and 40 out of these also other (unspecified) causative infected infant patients 32 of which were dengue virus protein NS1 positive. Also according to table 1., 32 patients suffered dengue fever, 12 from dengue hemorrhagic fever, and 10 even shock syndrome. Rather than a nucleic acid sensor typically implicated in immunological virus sensing, the expression level of the bacterial product sensor TLR2 on monocytes was altered in a disease severity dependent manner. Like the bona fide TLR2 agonist synthetic lipopeptide, numerous dengue strain virion challenges activated overexpressed and endogenous TLR2 in a fashion that was also CD14 and TLR1, yet not TLR4 and TLR6 dependent. According to statements of the authors, TLR2 upregulation of TLR2 upon dengue virus infection as well as virus protein driven TLR2 and TLR4 activation have been both implicated and debated previously already.

The authors observed increased anti TLR2 antibody and anti virus antigen binding by specific monocyte subpopulations. They also observed activation of HEK cells overexpressing TLR2 by virus #2 preparations, which was blockable not merely with anti TLR2 but also TLR1 and CD14 antibodies. In contrast, TLR4+ HEK cells failed to respond to challenge not merely with #2 virus, but also #1 and -4 variants. Some endocytosis inhibitor treatment impacted on dengue virus and overexpressed TLR2 driven cell activation as well as viability while others did not. Some treatments with antibodies of different specificities also changed properties of different monocyte classes in respect to TLR2 expression. Anti TLR2 treatments of PBMCs also reduced the capacities of their supernatants to induce specific cytokine and chemokine production by HUVEC cells. Some effects were dependent on virus viability whereas others were not. However, not all virus major strains 1, -3 and -4 were analyzed consequently in that an experimental focus was on -2 for a seemingly non-specified reason. DENV1 (strains 16007, Hawaii), DENV2 (strains 16681, NGC, TSV01), DENV3 (strains H87, 16562) and DENV4 (strains 1036, H241), were produced in the *Aedes albopictus* C6/36 cell line as described before (69).” and “CHIKV (La Reunion OPY1) was a gift from A. Merits (University of Tartu, Estonia), and was produced from infectious cDNA clones and passaged twice in Vero E6 cells (71). ZIKV (SL0216) was a gift from M.J. van Hemert (LUMC, the Netherlands); virus was produced in Vero E6 cells ..” which largely conceptually excludes microbial such as bacterial product contamination which is an issue important particularly in cases such as novel implications of pattern recognitions. It (e.g. through mycoplasmal infection) must be taken into consideration here because according to table 2 some virus strain preps were devoid of TLR2 activating capacity.

Moreover, a novelty of the implication of TLR2 as dengue virus sensor basing on TLR2 upregulation is being put into relation by citation of previous studies also implying TLR4.

The current manuscript makes an impression as if it tends to overcome these aspects by blockade and a multitude of expression analyses rather than coherent experimentation towards bringing up further functional evidence. For instance, why some strains do not activate TLR2 and the identity of the mediators of HUVEC activations upon PBMC infections are aspects deserving more rigorous clarification. The authors shall thus revise their manuscript potentially towards a disadvantage of some of somewhat multitude expression analyses and for the advantage of a bringing up of more substantial functional data.

Response

We would like to thank the reviewer for the valuable feedback and are glad to address all their comments and concerns.

All patients included in the study were diagnosed as undergoing acute DENV infection by either RT-qPCR, seroconversion of DENV IgM antibody or rapid test for detection of DENV NS1 protein as per WHO 1997 and 2009 criteria. Table 1 has been updated (page 4) and contains the information regarding the diagnostic tests performed and disease classification for the patients included in the study. The novel observations made through the structural analysis of patients' data combined with the functional analyses *in vitro* represent the main strength of our study. Moreover, in light of the pertinent literature, showing that NS1 signals through TLR4 and not TLR2, as suggested by Chen et al in 2015, our data sheds new light on the contribution of TLR2 to DENV pathogenesis. Dengue virus serotype 2 is widely used in the field to study DENV pathogenesis due to the fact that it is usually associated with severe disease as well as *in vivo* murine models (Vaughn *et al.*, 2000, Shresta *et al.*, 2006, Clyde *et al.*, 2006, Thomas *et al.*, 2008, Hamlin *et al.*, 2017). Importantly, this serotype was also the most prevalent etiologic agent in our cohort (Table 1). Therefore, we have used a WHO reference strain 16681 isolated from a patient with dengue hemorrhagic fever in Thailand (Russell, *et al.*, 1967, Diamond *et al.*, 2000). Importantly, we produced the virus from infectious clone to minimize introduction of lab adaptation-related mutations in the strain. We would like to clarify for the correct interpretation of the body text of this manuscript that the engagement of TLR2 by DENV requires TLR2/6 and CD14 but not TLR1 as stated by the reviewer.

Furthermore, we agree with the reviewer that bacterial contamination must be taken into consideration in our study, especially when it comes to novel implications of pattern recognition receptors. For this reason, as stated in material and methods section (page 25, lines 755-758 and page 26, lines 779-781) all cells and virus preparations were free of *Mycoplasma spp.* as tested using a commercial functional method (Lonza) and/or in-house qPCR assay adapted from Baronti *et al.*, 2013.

Altogether, we do not share the reviewer's feeling that our study does not provide coherent experimental analyses. Instead, and as highlighted by the reviewer one, due to combined

analyses of highly valuable patient samples with functional studies in several *in-vitro* models, we consider it to afford novel and functional insights into early event of DENV pathogenesis. Importantly, we do realize our study does not answer all remaining questions and therefore highlighted many of them in the discussion of the manuscript.

Since many of the reviewer's general and specific comments largely overlap, to increase the clarity we have addressed these comments in greater detail in the section below.

Russell, PK., *et al.* Antibody response in dengue hemorrhagic fever. *Jpn J Med Sci Biol.* 20: 103-108 (1967)

Diamond, MS., *et al.* Infection of human cells by dengue virus is modulated by different cell types and viral strains. *J Virol.* 74(17): 7814-7823 (2000)

Vaughn, DW., *et al.* Dengue viremia titer, antibody response pattern, and virus serotype correlate with disease severity. *J Infect Dis.* 181, 2-9 (2000)

Shresta, S., *et al.* Murine model for dengue virus-induced lethal disease with increased vascular permeability. *J Virol.* 80(20): 10208-10217 (2006)

Clyde, K., *et al.* Recent Advances in Deciphering Viral and Host Determinants of Dengue Virus Replication and Pathogenesis. *J Virol.* 80(23):11418-11431 (2006)

Thomas, L., *et al.* Influence of the dengue serotype, previous dengue infection, and plasma viral load on clinical presentation and outcome during a dengue-2 and dengue-4 co-epidemic. *Am J Trop Med Hyg.* 78(6):990-998 (2018)

Hamlin, Re., *et al.* High-dimensional CyTOF analysis of dengue virus-infected human DCs reveals distinct viral signatures. *JCI insight.* 2(13): e92424 (2017)

Baronti, C., *et al.* Mycoplasma removal: Simple curative methods for viral supernatants, *J. Virol. Methods* 187, 234–237 (2013).

Specific comments

1. Statements such as “.. as dengue or dengue fever” should be specified.

Response

We have revised the text accordingly (page 2)

Introduction: Clinical outcomes of DENV infection vary considerably and can be either limited to an acute febrile illness referred to as dengue fever (DF), or progress to a potentially fatal disease (severe dengue) also referred to as dengue hemorrhagic fever (DHF) and dengue shock syndrome (DSS) (2).

Response

We have tried to increase readability of these statements. (pages 14, 17 and 21 respectively)

Results: This effect was, however, independent of TLR2, since TLR2 block did not prevent the increase in CM numbers (Fig. S7E).

Increased levels of IL-1 β and TNF-alpha have been reported to be present in the plasma of DENV patients.

Discussion: The seemingly contrasting role of TLR2 on CM and NM are likely to be linked to their overall distinct function.

3. Abbreviations and special wordings used in figures such as PS and W or DF, DHF, DSS, IM, NM, IQF, DENV GEc, and “stratified by disease severity” should be worded and/or explained in legends. Also “MOG” and legend statements such as “.. hTLR4 cells were (mock)- treated for 24h with purified (p) mDENV1 (strain 16007), DENV2 (strain 16681), DENV4 (strain 1036) (MOG3500, MOG10000) or the respective standard (non-purified) preparations (MOG3500), n=2. NF-kB stimulation was assessed ..” should be clarified such as by removal of redundant terms such as in respect to “MOG3500” to enable understanding by non expert readers.

Response

We have adjusted the text to minimize the redundancy (page 10). The abbreviations and special wording listed by the reviewer are explained in the manuscript text and/or figure/table legends.

Results: HEK-Blue™ hTLR4 cells were treated for 24h with LPS (10 ng/mL) TNF- α (50 ng/mL), PAM3CSK4 (50 ng/mL), MOG3500 and MOG10000 of the purified preparations pDENV1 (strain 16007), pDENV2 (strain 16681), pDENV4 (strain 1036) and with MOG3500 of their respective non-purified preparation, n=2, for 24h. (Fig. S3)

4. Controls lacking should be included:

- Healthy control samples should be analyzed to indicate their respective expressions of antigens analyzed.

Response

We agree with the reviewer, yet due to the well-understandable difficulty in obtaining blood samples from healthy children, we were not able to collect sufficient number of age-matched healthy controls before the submission of the original manuscript. Fortunately, we have recently collected PBMCs from 15 age-matched healthy donors (HD), for which we were able

to analyze monocyte subsets distribution and TLR2 expression. The data have been added to the manuscript (updated Fig. 1. panel A and B, page 5). In line with previous studies, our patient cohort (DENV+) showed an overall increase in IM and NM when compared to age-matched healthy donors (Fig. 1A). Moreover, immunophenotyping of patients' PBMCs showed that both CM and IM had significantly higher expression ($p < 0.0001$) of TLR2 on their surface compared to NM (Fig. 1B). However, when TLR2 expression in DENV+ patients was compared to HD, all monocyte subsets from HDs had significantly higher expression ($p < 0.0001$) of TLR2 on their surface compared to DENV+ patients (Fig. 1B).

Fig. 1. Sustained high expression of TLR2 and increased frequency of monocytes correlates with DENV disease severity. (A-E) PBMCs were isolated from 15 age-matched healthy donors (HD) and 54 patients undergoing acute DENV infection (DENV+) who developed relatively mild (DF, n=32) or severe (DHF/DSS, n=22) disease. (A) Monocyte subsets distribution in healthy and DENV+ patients. (B) Percentages of cells expressing TLR2 were determined for each monocyte subset and (C) stratified by disease severity. (D) Percentages of NS3+ infected cells in DENV positive patients (n=15) stained intracellularly for DENV NS3. (E) Monocyte subsets distribution in patients, stratified by disease severity. CM= classical monocytes, IM= intermediate monocytes, NM: non-classical monocytes. Bars represent median with IQR. P values were obtained by Mann-Whitney test (*P< 0.05; **P<0.01; ***P<0.001; ****P<0.0001)

Results:

Their distribution is influenced by inflammatory conditions. Previous studies have shown that DENV infection *in vivo* increased frequencies of either IM or NM (21, 32). In line with previous studies, our patient cohort showed an overall increase in IM and NM when compared to age-matched healthy controls (Fig. 1A). In addition, Azeredo *et al*(21) demonstrated increased TLR2 expression on blood monocytes in DENV-infected patients when compared to healthy controls yet how TLR2 is distributed over the three monocyte subsets and its potential impact on disease burden remain elusive (21, 24). To investigate this, we isolated PBMCs from DENV-infected patients during the acute phase of infection (n=54) and 15 age-matched healthy donors (HD) and subsequently stained with an anti-human TLR2 antibody or a conjugated isotype-matched antibody as a control (Fig. S1A).

In line with the above studies, immunophenotyping of patients' PBMCs showed that both CM and IM had significantly higher expression ($p<0.0001$) of TLR2 on their surface compared to NM (Fig. 1B). However, when TLR2 expression in DENV+ patients was compared to HD, all monocyte subsets from HDs had significantly higher TLR2 expression ($p<0.0001$) compared to DENV+ patients (Fig. 1B).

The percentage of CM predominated in patients that developed severe disease ($p<0.5$) but while there were no significant changes in the percentages of IM, Patients that developed DF were predominantly found in the NM percentage ($P<0.5$) (Fig. 1E). Altogether, these data suggest that sustained high levels of TLR2 play a role in disease development during the acute phase of DENV infection.

Regardless of virus preparation, *in-vitro* DENV infection of PBMCs increased the mean fluorescent intensity (MFI) of TLR2 (Fig. 3A) and the percentage of TLR2-positive cells (Fig. 3B). Notably this was in contrast to the data collected from our *ex-vivo* samples (Fig. 1A) but in line with previous findings (21). Importantly, PBMCs isolated from adult healthy and DENV-seronegative donors in the Netherlands expressed similar levels of TLR2 as our pediatric HD in Cambodia. Hence, this might suggest that monocyte responses and thereby TLR2 expression on the surface of these cells depends on the age, genetic background and/or past DENV infection.

Discussion:

We show that in children, DENV infection led to decreased surface expression of TLR2 on all monocyte subsets when compared to age-matched HD. Notably, the sustained high expression of TLR2 in acute infection on CM was associated with severe disease development.

Interestingly, even though DENV infection increased the frequencies of IM and NM compared to HD, this increase did not differentiate between the severity groups. In fact, in our patients' cohort, only the increased frequency of CM correlated with severe disease development.

Material and methods:

Human peripheral blood mononuclear cells (PBMCs) were isolated from blood samples of DENV positive patients and age-matched healthy controls using Ficoll-Paque density gradient centrifugation and frozen in 10% DMSO until analysis.

- **1a** virus producing cell supernatant and lysate, mock control virus preparation, were the virus producing cells (e.g. mycoplasma) bacteria (e.g. mycoplasma) free?

Response

As mentioned in the response to general comment, all virus preparations and cell lines used in our experiments tested negative for *Mycoplasma spp.* using a commercial functional method (Lonza) and/or in-house qPCR assay adapted from Baronti *et al.*, 2013 (page 25-26 Lines 755-758 and 779-781).

Baronti, C., *et al.* Mycoplasma removal: Simple curative methods for viral supernatants, J. Virol. Methods 187, 234–237 (2013).

- **1b** activation independent of TLR2 to demonstrate specificity, TNF is mentioned in mat. and meth. yet respective data illustration seem to be missing. Also, “null” (TLR2-) cells should be used in parallel rather than punctually.

Response

We share the reviewer's opinion about the necessity to demonstrate TLR2 specificity. We indeed mention TNF- α in Material and Methods as a positive control in HEK-Blue Null2 cells (page 27, line 820) and show the respective data as a part of the Fig. S3, panel A. We also understand the concern of the reviewer about using the data of the Null2 cells in parallel rather than punctually. Importantly, the experiments were performed as much as possible in parallel, however; as these data do not represent the main scope of the Fig. 2 and are merely

another way (besides TLR2 axis blockage) to corroborate the specific involvement of TLR2, we have included as a supplementary data.

-1c Anti TLR2 should be applied to TLR4+ HEK cells challenged with LPS to further demonstrate specificity. - 1c de control is also needed for CD14 (involvement of which in TLR2 action is not comprehensively uncontroversial) such as TNF, as well as anti TLR1/6 such as by application of the antibodies to TLR4+ HEK cells prior to LPS challenge.

Response

We agree with the reviewer in the importance of agonist specificity when using agonist in the HEK-Blue™ cells. Importantly, in our experiments using HEK-Blue hTLR4 cells we used PAM3CSK4 (TLR2/TLR1/CD14 agonist) and TNF- α to prove specificity of this cell line. The figure below shows that only TNF- α and LPS treatments were able to activate HEK-Blue™ hTLR4 cells, which is in agreement with the manufacturer (InvivoGen). Moreover, hTLR4 cells treated with the TLR2/TLR1/CD14 agonist PAM3CSK4 failed to induce NF- κ B activation, thereby confirming the specificity of this cell line to LPS and TNF- α . This figure has been included as panel C of the Fig. S3 and discussed in the revised manuscript in page 7.

Panel C Fig. S3: HEK-Blue™ hTLR4 cells were treated with LPS (10 ng/mL), TNF- α (100 ng/mL) and PAM3CSK4 (25 ng/mL) for 24h, n=3. NF- κ B stimulation was assessed by QUANTI-Blue™, OD values show the induction of NF- κ B. Data represent the mean \pm SD. P values were obtained by paired one-tailed t test (** P<0.01)

Results: Importantly, PAM3 and our standard DENV preparations did not activate HEK-Blue™ hTLR4 cells (Fig. S3C and D), which only responded to LPS and TNF- α treatments. These results confirm the specificity of the HEK-TLR Blue system employed and imply that the soluble form of DENV nonstructural protein 1 (NS1), previously shown to signal through TLR4 (25, 26), was not a confounding factor in our experiments.

-1d Also, is dynasore (Sigma Aldrich), chloroquine (Sigma Aldrich), and chlorpromazine (Sigma Aldrich) just not shown in Fig 2d? According to a common view, TLR2 is expressed on the cell surface for which reason its activation is principally endocytosis independent.

Response

We thank the reviewer for noticing the lack of these results. Indeed, we have tried chlorpromazine, chloroquine and dynasore in the HEK-Blue™ hTLR2 cells (Fig. S4). However, we could not use them in our experiments due to their high level of cytotoxicity observed in these cells. We have changed the text accordingly (page 7, lines 215-216).

Results: Chloroquine, chlorpromazine and dynasore were excluded from the experiments due to their high level of cytotoxicity (Fig. S4).

Indeed, TLR2 was originally thought to be expressed and signal only at the plasma membrane. Importantly however, since then, many groups have shown that depending on the nature of its ligands, TLR2 can be internalized via clathrin-dependent endocytosis and mediate NF- κ B activation from endosomal compartments (Shamsul *et al.*, 2009, Barbalat *et al.*, 2009, Brandt *et al.*, 2013, Stack *et al.*, 2014).

Shamsul, HM., *et al.* The Toll-like receptor 2 (TLR2) ligand FSL-1 is internalized via the clathrin-dependent endocytic pathway triggered by CD14 and CD36 but not by TLR2. *Immunology*. 130: 262-272 (2009).

Barbalat, R., *et al.* Toll-like receptor 2 on inflammatory monocytes induces type I interferon in response to viral but not bacterial ligands. *Nat Immunol*. 10(11): 1200-1207 (2009)

Brandt, KJ., *et al.* TLR2 Ligands Induce NF- κ B Activation from Endosomal Compartments of Human Monocytes. *PLoS ONE*. 8(12): e80743 (2013)

Stack, J., *et al.* TRAM is required for TLR2 endosomal signaling to type I IFN induction. *J Immunol*. 193(12): 6090-6102 (2014)

-1e The actual isotype of “Isotype-matched antibody labelled with PE (clone MOPC 173) (BioLegend) was used as negative control for comparing TLR2 expression” shall be named and controls be illustrated to indicate that not increased Fc receptor expression is causing increased antibody binding. Also, a diagram indicating often marginal surface TLR2 staining even of monocytes or macrophages should be displayed.

Response

PBMCs from dengue patients have been stained with an isotype-matched IgG2a antibody labelled with PE for comparing TLR2 expression. A histogram showing the staining of

monocytes with TLR2 and isotype control antibodies has been included in the manuscript (Fig. S1 panel A) in page 6 and discussed in referenced in page 3, lines 98-99.

Panel A Fig. S1: Expression of TLR2 in patients PBMCs. PBMCs were isolated from a patient undergoing acute DENV infection. To measure the percentage of cells expressing TLR2, cells were stained with an anti-TLR2 antibody conjugated with PE. An isotype and concentration matched antibody labelled with PE was used as negative control for measuring TLR2 expression. The figure above represents a histogram comparing the anti-TLR2 antibody (red line) and the corresponding isotype control (grey line).

Results: To investigate this, we isolated PBMCs from DENV-infected patients during the acute phase of infection (n=54) and 15 age-matched healthy donors (HD) and subsequently stained with an anti-human TLR2 antibody or a conjugated isotype-matched antibody as a control (Fig. S1A).

5. Also, other stainings such as that for “NS3” which is being implicated as indicative of virus replication by “Notably, CM and IM, but not NM, were the primary targets of DENV replication (Fig. 1C) and CM predominated in patients that developed severe disease ($p<0.5$) (Fig. 1D).” should be isotype controlled.

Response

We agree with the reviewer that an addition of an isotype control is useful. PBMCs from DENV infected patients were stained with a rabbit polyclonal anti-DENV NS3 antibody or rabbit IgG along with a goat anti-rabbit antibody conjugated with FITC. A histogram showing the staining of monocytes with NS3 and isotype control antibodies has been included in the manuscript (Fig. S1 panel B) in page 6 and discussed in referenced in page 3, lines 103 -104.

Panel B Fig. S1: Validation of rabbit anti-NS3 antibody in patients' PBMCs. PBMCs were isolated from a patient undergoing acute DENV infection. To confirm the presence of DENV, cells were stained with a rabbit anti-NS3 antibody followed by goat anti-rabbit IgG conjugated with FITC. A non-specific polyclonal rabbit antibody was used as a negative control for staining of DENV infection. The figure above represents a histogram comparing the anti-NS3 antibody (red line) and the isotype control (grey line).

Results: We proceeded to determine DENV infection in monocyte subsets. To ensure detection of active DENV replication, rather than viral uptake, we used a rabbit polyclonal antibody against DENV non-structural protein 3 (NS3) and subsequently a secondary antibody labeled with FITC. A non-specific rabbit polyclonal antibody along with the same secondary antibody was used as a negative control (Fig. S1B).

5. Furthermore, the authors should outline why virus antigen positivity not merely represents endocytosis or other cellular uptake of virus rather than replication of it. Application of k.o. mice or cells such as those lacking TLRs and/or CD14 implicated would be highly supportive for the manuscript.

Response

We would like clarify that nonstructural protein 3 (NS3) is one of the seven nonstructural proteins and as such is not expressed in the flavivirus virions. The expression of NS3 which has a helicase and triphosphates activity (Chambers *et al.*, 1990, Wengler *et al.*, 1993, Warrenner *et al.*, 1993; Westaway *et al.*, 1998. Yon *et al.*, 2005) is thus commonly used as an indication of viral replication rather than endocytosis or any other cellular uptake of the flaviviruses (Aguirre *et al.*, 2017, Michlmayr *et al.*, 2017).

With regard or the murine models, Sun *et al.*, 2016 have reported species-specific preferences in signaling protein usage between mice and humans, especially in the members of the interleukin-1 receptor- associated kinase (IRAK) family, which are importantly involved in TLR pathways. Considering these intrinsic differences between human and murine cells and the inadequacy of wild type murine model in mirroring natural DENV infection, we have deliberately chosen to study infection and immune responses in three

different human cells types (PBMCs, HUVEC and HEKs). We have not only assessed the effect of TLR2 axis blockage on the infection and immune responses in PBMCs but also assessed these in TLR2/co-receptors expressing HEK-Blue cells in comparison to the parental HEK-Blue™ NULL2 cells which lack these receptors. Moreover, in line with the results obtained *in-vitro*, patients' data show no clear correlation between TLR2 expression, virus titers and disease severity (Fig. S6). Altogether, these observations indicate that during DENV infection, TLR2 contributes to the initiation of inflammation and viral attachment and entry to CD14-positive cells rather than the virus yield.

Chambers, T.J., *et al.* Flavivirus genome organization, expression and replication. *Annu. Rev. Microbiol.* 44, 649-688 (1990)

Wengler, G. The NS3 non-structural protein of flavivirus contains an RNA triphosphatase activity. *Virology* 197, 265-273 (1993)

Warrener, P., *et al.* RNA-stimulated NTPase activity associated with yellow fever virus NS3 protein in bacteria. *J Virol.* 67, 989-996 (1993)

Westaway, E., *et al.* Ultrastructure of Kunjin virus-infected cells: colocalization of NS1 and NS3 with double-stranded RNA, and of NS2B with NS3, in virus-induced membrane structures *J. Virol.* 71, 6650-6661(1997)

Yon, C., *et al.* Modulation of the nucleoside triphosphatase/RNA helicase and 5'-RNA triphosphatase activities of Dengue virus type 2 nonstructural protein 3 (NS3) by interaction with NS5, the RNA-dependent RNA polymerase. *J Biol Chem.* 280(29):27412-9 (2005)

Sun, J., *et al.* Comprehensive RNAi-based screening of human and mouse TLR pathways identifies species-specific preferences in signaling protein use. *Sci Signal.* 9, ra3 (2016)

Aguirre, S., *et al.* Dengue virus NS2B protein targets cGAS for degradation and prevents mitochondrial DNA sensing during infection. *Nat Microbiol.* 2. 17037 (2017)

Michlmayr, D., *et al.* CD14+CD16+ monocytes are the main target of Zika virus infection in peripheral blood mononuclear cells in a paediatric study in Nicaragua. *Nat Microbiol.* 2: 1462-1470 (2017)

6. The subsequent term “Altogether, these data imply that TLR2 on classical monocytes plays a significant role in disease development.” is an overstatement. The data provided by figure 1 rather indicate coincidence, in numerous and not individually indicated cases potentially also with other unspecified infection, of higher TLR2 and virus antigen presence in specified monocytes. Also, why 4G2 was used to indicate infection alternatively seems to have left unexplained. The authors should revise accordingly.

Response

We have revised the text accordingly (page 4).

Results: Altogether, these data suggest that sustained high levels of TLR2 play a role in disease development during the acute phase of DENV infection.

With regard of the second point, we do not understand the rationale of the reviewer comment. Considering the high number of patients and in the revised version also healthy controls, the ability of DENV to infect CM, the coincidental association are very unlikely.

The 4G2 antibody detecting virus envelope protein is the most frequently used and one of the best characterized antibody to assess flavivirus infections in our *in-vitro* cell systems (Flipse et al., 2016, Michlmayr et al., 2017, Hamlin, *et al.*, 2017, Diosa-Toro *et al.*, 2019). We have sought to use the 4G2 antibody throughout the whole study, however due to its inability to detect virus infection in patients' PBMCs we used the recently characterized rabbit polyclonal anti-DENV NS3 antibody. We have mentioned this in the material and methods section (pages 24-25).

Material and methods: Due to the inability of the DENV envelope-directed 4G2 antibody (Millipore) to detect virus infection in our patients' PBMCs, we used the recently characterized rabbit polyclonal anti-DENV NS3 antibody. Briefly, PBMCs were fixed and permeabilized using True nuclear transcription factor buffer set (BioLegend) and stained intracellularly using a rabbit anti-DENV NS3 antibody (GeneTex) or rabbit IgG Isotype control (BioLegend) followed by goat anti-rabbit IgG conjugated with FITC (Abcam). Data was analysed using the FlowJo software (BD Biosciences).

Flipse, J., *et al.* Antibody-dependent enhancement of dengue virus infection in human macrophages; balancing higher fusion against antiviral responses. *Scientific Report*. 6: 29201 (2016).

Michlmayr, D., *et al.* CD14+CD16+ monocytes are the main target of Zika virus infection in peripheral blood mononuclear cells in a paediatric study in Nicaragua. *Nat Microbiol*. 2: 1462-1470 (2017)

Hamlin, RE., *et al.* High-dimensional CyTOF analysis of dengue virus-infected human DCs reveals distinct viral signatures. *JCL Insight*. 2(13): e92424 (2017)

Diosa-Toro, M., *et al.* Tomatidine, a novel antiviral compound towards dengue virus. *Antiviral Res*. 161: 90-99 (2019).

7. CD14, TLR1 and TLR6 expression by HEK cells should be demonstrated in order to justify blockades of both of the two antigens.

Response

We would like to clarify that HEK-Blue™ hTLR2 cells were obtained from InvivoGen, which guarantees the cells expressed CD14. The company also verified that the HEK 293 cells used for the ectopic transfection of TLR2 and CD14, endogenously express TLR1 and TLR6. Importantly, we ensure the specificity and sensitivity of the cell line by usage of specific agonists in every experiments. As observed in Figure 2 panel C, blocking of TLR1 and CD14 but not TLR6 decreased NF-κB activation by the TLR2/1/CD14 agonist PAM3CSK6 while blocking of TLR6, but not TLR1 and CD14, partially decreased the NF-κB activation by the

TLR2/6 agonist PAM2CSK4. Our results are thus in line with information provided by InvivoGen.

8. “All DENV serotypes are sensed by TLR2” seems to collide with “-“ (minus) occurring in table 2 for some strains of dengue virus. The authors should revise accordingly.

Response

We have revised the text accordingly (page 12).

Results: *TLR2-mediated sensing of DENV depends on virus strain rather than the serotype*

9. Figure 3 seems to lack subfigure “c” representation in legend. Also, the validity of inter subfigure diagram legends is not specified to degree that would allow for fast understanding. Whether control challenge such as with LPS or other virus used in other experiments throughout the manuscript would have effects resembling those of dengue virus should be illustrated. Not necessarily “must” TLR2 expression level upregulation correlate with TLR2 recruitment and activity.

Response

Subfigure “c” representation in the legend is now highlighted.

We agree with the reviewer that it would be very interesting to compare the effect observed with DENV with LPS or any of the other viruses tested in our manuscript. Importantly however, considering the magnitude and complexity of differences between different agonists and viruses and the fact that our aim was to elucidate the role of TLR2 in shaping DENV-mediated immune responses, we consider these experiments beyond the scope of this manuscript.

Regarding correlation of TLR2 expression with its activation, we would like to clarify that changes in the expression of TLRs in course of inflammation or a disease are generally considered as a surrogate of changes in immune activation. For instance, upregulation of TLR2 expression have been associated with its function in hemorrhagic-primed lung inflammation (Fan *et al.*, 2006) and gastric tumorigenesis (Tye *et al.*, 2012). Importantly, downregulation of these receptors on cells expressing them in high levels in the steady state, either through their internalization (Lu *et al.*, 2018) or shedding (van Montfoort *et al.*, 2016, Holst *et al.*, 2017) has also been implicated in the initiation of inflammation in course of HBV infection (van Montfoort *et al.*, 2016) and sepsis (Holst *et al.*, 2017). Thus, we stand by our statement that differential expression of TLR2 in monocytes in course of DENV infection is indicative of its function and activation.

Fan, J., *et al.* Hemorrhagic shock-activated neutrophils augment TLR4 signaling-induced TLR2 upregulation in alveolar macrophages: role in hemorrhage-primed lung inflammation. *Am J Physiol Lung Cell Mol Physiol.* 290:L738-L746 (2006)

Tye, H., *et al.* STAT3-driven upregulation of TLR2 promotes gastric tumorigenesis independent of tumor inflammation. *Cancer cell.* 22(4): 466-478 (2012)

van Montfoort, N., *et al.* Hepatitis B Virus Surface Antigen Activates Myeloid Dendritic Cells via a Soluble CD14-Dependent Mechanism. *J virol.* 90(14): 6187-6199 (2016)

Holst, B., *et al.* Toll-like receptor 2 is a biomarker for sepsis in critically ill patients with multi-organ failure within 12 h of ICU admission. *Intensive Care Med.* Exp. 5, 2-5 (2017)

Lu, MY., *et al.* The Phosphorylation of CCR6 on Distinct Ser/Thr Residues in the Carboxyl Terminus Differentially Regulates Biological Function. *Front Immunol.* 9: 405 (2018)

10. Besides DENV2, also -1 and -4 should be analyzed to support generality of the conclusions drawn.

Response

We understand the interest of the reviewer to compare the responses mediated by all serotypes. Importantly however, considering the differences in the ability to engage TLR2 activation between various strains within each of DENV serotypes as measured in the sensitive HEK-Blue™ hTLR2 system, the complete analysis of all serotypes and strains is conceptually unmanageable. Consequently, although we fully agree on the relevance of studying differences between the immune activation triggered by different DENV serotypes, as recently showed by Hamlin *et al.*, 2017, we ensured the relevance of our results by combining the *ex-vivo* and *in-vitro* data to understand the mechanisms occurring in course of natural infections. Noteworthy is that our DENV cohort was infected with different DENV serotypes although DENV2 was the most prevalent serotype. Moreover, to address the role of TLR2 on monocytes in initiation of inflammatory responses as measured in human vascular endothelial cells, we used different DENV2 16681 strain preparations that as summarized in Table 2 differed in the ability to activate TLR2.

Hamlin, RE., *et al.* High-dimensional CyTOF analysis of dengue virus-infected human DCs reveals distinct viral signatures. *JCL Insight.* 2(13): e92424 (2017)

11. PBMCs treated with anti TLR2 antibody and infected with virus should be washed after the challenges and prior to transfer of their supernatants to HUVECs to exclude carry over of both of the molecular entities, antibodies and virus to the secondary cell culture. Again, control antibody, such as isotype control or related specificity one application appear as demandable.

Response

We agree with the reviewer that by removing the inoculum and washing the cells we would minimize the carryover of the virus and pre-treatment antibodies. We would like to point out

however, that multiple washing steps especially shortly after one treatment are stressful to primary cells such as PBMCs, and by compromising their viability mask the treatment-specific immune responses. Therefore, instead, to ensure that the carryover of the virus was not the confounding factor in activation of the HUVEC, we assessed the responses of HUVEC at 6hpi. At this time point, the dose of the virus equivalent to that found in the supernatants of PBMCs did not activate HUVEC. We have emphasized this in the text and corresponding supporting figure (Fig. S9, page 16) in the manuscript. Furthermore, in order to minimize the effect of the blocking antibodies on the TLR2 in HUVEC we made sure that their concentrations following treatments in PBMCs and addition of supernatants to HUVEC did not exert inhibition of NF- κ B activity in the HEK-Blue™ hTLR2 systems. Specifically, prior treatment, the PBMCs are blocked with 5 μ g/mL of the respective antibody, concentration that is further diluted by 2-fold following infection. Moreover, the concentration of the carryover blocking antibodies is reduced again by half at the moment of addition to the HUVEC.

To prove specificity in our HUVEC assays we tested the isotype control of the TLR2 blocking antibody as suggested by the reviewer. The figure below (panel C Fig. S10) shows that only the supernatants from DENV2 infected PBMCs induced the upregulation of VCAM-1 and ICAM-1. Moreover, the production of the inflammatory mediators by the PBMCs capable to active HUVEC was abrogated by treatment with anti-TLR2 but not by its isotype-matched antibody. All these data suggest that the production of inflammatory mediators by DENV on human PBMCs capable of induce the expression of the adhesion molecules on HUVEC is TLR2 specific.

Panel C Fig. S10. Production of vasoactive mediators by DENV2 infected PBMCs is TLR2 specific. HUVEC were incubated for 6 hours with cell-free supernatants from mock or DENV2-infected PBMCs in the presence or absence of α TLR2 or isotype control block, n=2-4. Fold-changes in surface expression of E-selectin, VCAM-1 and ICAM-1 compared to the respective mock. P values were obtained by unpaired t test. (*P<0.05, **P<0.001)

The figure has been included as a panel C in the Fig. S10 in page 19 and discussed in pages 16-17.

Results: To ensure that the activation of endothelial cells was due to soluble inflammatory mediators excreted by infected PBMCs rather than the presence of the virus itself, HUVEC were incubated with an equal number of GEC as those present in the supernatants of infected PBMCs. Incubation for six hours did not lead to the upregulation of adhesion molecules when compared to LPS (positive control) (Fig. S9). Thus, endothelial cell activation observed with the supernatants of infected PBMC was due to soluble mediators excreted upon infection PBMCs (Fig. 4, B and C) and not due to the carryover of the virus.

Additionally, isotype control antibody block did not have an effect on the vascular responses of PBMCs infected with DENV2 (Fig. S10C). Moreover, the TLR2 block had no effect on the vascular responses of PBMCs treated with the TLR4/CD14 agonist LPS, indicating that the effect observed for DENV infection is indeed TLR2-specific (Fig. S10D).

12. Figure s8 lacks not merely non-anti TLR2 treated controls, but also plausibility since TLR2 activation towards TNF production with lipopeptide failed to be blockable. The authors should address both aspects by respective data supplementation

Response.

We thank the reviewer for the observation. We have further titrated PAM3CSK4 on PBMCs to make sure we did not over-stimulated the cells at the initially used concentration, which was based on the pertinent literature (Ultaigh et al., 2011). Indeed, PAM3CSK4 induced the intracellular accumulation of IL-1 β and TNF- α in the concentration and TLR2-dependent manner (Fig. S11A and B). Interestingly, at the concentration tested only the intracellular accumulation of IL-1 β was significantly reduced by prior blocking of TLR2 blockade (Fig. S11A), suggesting differential TLR2- mediated pathways triggering the production of these cytokines. Importantly, we did not detect TNF- α accumulation following exposure to DENV2 (Fig. S11C) while IL-1 β accumulation was evident in monocytes fraction of three out of four tested donors and depending on the donor was induced by exposure to either infectious (DENV) or non-infectious virus UV-DENV (Fig. S11D). We have added these data and modified the text accordingly.

Ultaigh, SNA., *et al.* Blockade of Toll-like receptor 2 prevents spontaneous cytokine release from rheumatoid arthritis *ex vivo* synovial explant cultures. *Arthritis Research & Therapy*. 13: R33 (2011)

Figure S11. Intracellular accumulation of IL-1 β and TNF- α in PBMCs after exposure to TLR2 agonist and DENV. PBMCs from healthy donors were (mock)-treated with α TLR2 (5 μ g/mL) for 2h prior treatment with PAM3CSK4 (600 ng/mL), DENV2 (MOI 10) or UV-DENV2 (MOG1000) (**A**) Percentage of monocytes (in PBMCs) with intracellular expression of TNF- α and IL-1 β was measured by flow cytometry at 6h and 18 h post treatment respectively. (**B**) TNF- α intracellular accumulation after treatment with DENV2 and UV-DENV2 (**D**) TNF- α and IL-1 β intracellular accumulation after treatment with several concentrations of PAM3CSK4. (**A** and **B**) Bars represent mean \pm SEM. P value was obtained by paired one-tailed t test. (* $P < 0.05$).

The Fig. S11 has been updated with the above depicted panels in page 20 and discussed in page 17

Results: As expected, positive control PAM3CSK4 induced the intracellular accumulation of IL-1 β and TNF- α in the concentration and TLR2-dependent manner (Fig. S11A and B). Interestingly, at the concentration tested only the intracellular accumulation of IL-1 β was significantly reduced by prior blocking of TLR2 (Fig. S11A), suggesting differential TLR2-mediated pathways triggering the production of these cytokines. Importantly, we did not detect TNF- α accumulation following exposure to DENV2 (Fig. S11C) while IL-1 β

accumulation was evident in monocytes fraction of three out of four tested donors and depending on the donor was induced by exposure to either infectious (DENV) or non-infectious virus UV-DENV (Fig. S11D). Importantly, the observed variations were donor-dependent rather than virus preparation-dependent, highlighting the inherent differences between the HEK-Blue™ hTLR2 and monocytes in TLR2-mediated sensing of DENV. None of the treatments induced intracellular accumulation of both cytokines in the lymphocytic fraction of the PBMCs (Fig. S11E and F).

13. Whether TLR2 non-activating dengue strains also increase TLR2 expression levels and whether their pathogenicity differs from that of activating counterparts both qualitatively and quantitatively should be indicated more rigorously by the manuscript.

Response

The reviewer raises an interesting point. In fact, since we observed differential activation of HEK-Blue™ hTLR2 reporter cells following exposure to different preparations of DENV-2 16681 strain (Table 2). Therefore, to gain further insights into the possible differences in TLR2-dependency in PBMCs, we used all of these preparations for subsequent analyses. Interestingly, any variations in TLR2-mediated inflammatory responses induced by the replicative or non-replicative of these viruses were donor-dependent rather preparation dependent, highlighting the inherent differences between the HEK-Blue™ hTLR2 and primary monocytes in TLR2-mediated sensing of DENV. We have included this information into the results section of the manuscript (pages 13/17).

Results:

To further substantiate the role of TLR2 as a regulator of inflammatory responses, we isolated PBMCs from healthy, DENV-seronegative, donors and infected them under TLR2 axis blocking and non-blocking conditions with DENV2 16681 strain at multiplicities of infection (MOI) of 10, as described previously (39). To gain further insights into the possible repercussions of TLR2-engagement on PBMCs, we used virus preparations that had a differential capacity to activate HEK-Blue™ hTLR2 reporter cells (Table 2). (Page 13)

Importantly, the observed variations were donor-dependent rather than virus preparation-dependent, highlighting the inherent differences between the HEK-Blue™ hTLR2 and monocytes in TLR2-mediated sensing of DENV. (Page 17)

Reviewer #3

Overall summary

1.1 Aguilar-Briseño et al. reported Toll-like receptor 2 senses dengue virions and mediates the innate inflammatory response underlying disease pathogenesis. Their results suggested that the role of TLR2 plays a significant role in mediating the immunopathology, and proposed that drugs targeting on TLR2 may have benefit the patients in severe conditions. How does dengue virus induce severe disease in certain subjects remain an enigma despite many decade intensive research efforts? There are many factors derived from innate immune cells been linked to or correlated plasma leakage in severe patients. The current report attempted to provide in vitro evidence with PBMC isolated from acute dengue patients to demonstrate that TLR2 in innate phagocytes cells play significant role in severe dengue.

Response

We thank the reviewer for the positive and constructive feedback provided. We have addressed all the comments point by point.

1.2 The major issue with the current study is that the virus utilized was from in vitro and was derived from C6/36 cells. Does NS1 in supernatant in dengue virus infected C3/36 cells? If yes, the authors should check the genetic materials of C3/36 cells to ensure that there is no contamination of mammalian Vero cells. Reality is that the virus circulating in acute patients is from infected human beings and the host cells. Interestingly, it has been demonstrated that the viral morphology and property obtained from acute dengue patients are significantly different from that of in vitro virus 2-5. Since the authors had recruited acute dengue patients, the dengue virus should be easily obtained from these specimens and repeat the experiments.

Response

We appreciate the point raised by the reviewer. We have also tested dengue virus produced on monocyte-derived dendritic cells and monocyte-derived macrophages. Please refer to our response to comment 1.2 of the Reviewer 1 for the details of these analyses.

We agree with the reviewer that performing the experiments with patient derived viruses could be an interesting addition to the manuscript. Indeed, recent studies have shown that passaging of patient derived viruses in C6/36 cells significantly alters the viral protein structure especially in the prM-E protein region which is critical for viral entry into host cells and neutralization or enhancement by antibodies (Chaichana *et al.*, 2014). Furthermore, passaging of patient-derived viruses in mosquito cells alters the glycosylation patterns of DENV E proteins which can regulate the engagement of E protein with cell surface receptors

thus altering the virulence of DENV in mammalian cells (Yap *et al.*, 2017). However, viral yield directly obtained from patient isolates is low, especially taking into account these are pediatric patients. Hence, it is not possible to obtain high titers of patient derived viruses required for these experiments without passing them at least three times in mosquito C6/36 cells (WHO 2009). Importantly, DENV2 strain used in our study was produced directly from infectious clone and a maximum of two passages in c6/36 to increase the virus yield.

World Health Organization. (2009). Dengue guidelines for diagnosis, treatment, prevention and control: new edition. Geneva: World Health Organization. <http://www.who.int/iris/handle/10665/44188>

Chaichana, P., *et al.* Levels of Antibody-Dependent Enhancement in Vitro Using Viruses and Plasma from Dengue Patients. Plos One. 9: e92173 (2014)

Yap, S., *et al.* Dengue Virus Glycosylation: What Do We Know? Frontiers in microbiology, 8, 1415. doi:10.3389/fmicb.2017.01415 (2017)

1 In Table 1: The individual virus titres in these subjects, presented as PFU/ml of blood, and the day of onset fever should be shown. Parameters collected from dengue patients are easily affected by fever days. In addition, will the viral titers correlate with the claim that TLR2 expressed more in CM and IM, based upon the day of fever?

Response

We have revised Table 1 accordingly (page 4), including the day of onset of fever. The majority of the patients were included at day 3 (n=17) or day 4 (n=26) of fever while few individuals were included at day 2 (n=6) and day 5 (n=4). In addition, there was no difference in mean day of fever at admittance between DF and DHF/DSS patients. There was no correlation between viral titer and TLR2 expression on any given day in the different subsets of monocytes.

Table 1. Demographic data and clinical parameters of the studied population. Patients were characterized according to the WHO 1997 criteria. DENV serotype and viral load were determined by qRT-PCR. Primary or secondary infection was determined by an HI test on acute and convalescent samples. N/A, not available; IQR, interquartile range; DF, dengue fever; DHF, dengue hemorrhagic fever; DSS, dengue shock syndrome; NS1, non-structural protein 1.

	Total patients	DF	DHF/DSS	Healthy donors
	54	32	22 (DHF -12; DSS -10)	15
Age	8.14 ± 4.01	8.29 ± 4.29	7.29 ± 3.67	10.08 ± 4.06
M/F ratio	0.8	0.9	0.7	1.5
Weight (kg)	22.7	23.7	21.2	29.3
Height (cm)	121.1	122.6	119.3	127.5
Temperature (°C)	37.6	38.0	37.1	NA
Hematocrit (%)	42.8	38.0	44.0	
Platelets (x 10⁹/L)	93.7	116.3	65.0	
Day of fever (Mean)	3.6	3.3	3.9	
DENV1	8	5	3	
DENV2	37	21	16	
DENV3	0	0	0	
DENV4	3	3	0	
NA	6	3	3	
NS1+	32	21	11	
PCR+	48	30	18	
Viral load (copies/ml) (Median, IQR)	2.2x10 ⁴ (7.4x10 ³ - 3.4x10 ⁵)	1.2x10 ⁵ (1.4x10 ⁴ - 1.2x10 ⁶)	7.6x10 ³ (6.3x10 ³ - 3.5x10 ⁴)	
Secondary infection	74%	65%	90%	

2 Figure 1C, it is insufficient to claim that NS3+ is a surrogate marker for viral replication or infection. The figure should be shown according to day of fever. It is likely that these NS3+ cells are a result of phagocytic effect 6. To claim the infection, the authors should sort out these cells and perform the co-culture to recover the infectious dengue virus.

Response

We thank the reviewer for the suggestion to sort out the cells and perform co-culture experiments. However, in the current experimental setup, it will not be possible to perform the aforementioned experiment since the intracellular staining for the detection of NS3 involves fixation and permeabilization of the cells. Furthermore, we would like to clarify that NS3 is not present in the virus particle and is expressed only in actively infected cells. Please refer to our response to **comment 5 of the reviewer 2** for the detailed clarification of this issue.

3. Time-dependent is critical parameter in dengue, especially for those subjects with plasma leakage. All figures associated with dengue patients should be shown kinetically.

Response

We agree with the comment that day of sample collection can affect the clinical parameters of the patients. Sample collection from patients was done only at one point within 96 hours of onset of fever at hospital admittance and not at multiple time points. Therefore, day-wise data of the expression of TLR2 on monocyte is not available for these patients. The majority of the patients were included at day 3 (n=17) or day 4 (n=26) of fever while few individuals were included at day 2 (n=6) and day 5 (n=4). In addition, there was no difference in mean day of fever at admittance between DF and DHF/DSS patients (revised Table 1). Moreover, there was no significant difference in TLR2 expression on monocytes in patients grouped according to day of fever at time of sample collection (Fig. S2).

Fig. S2. TLR2 expression on monocyte subsets of DENV positive patients. DENV positive patients were classified according to day post-fever on the day of sample classification. % of TLR2 positive was determined by flow cytometry. Bars and lines indicate median and IQR

We have included Fig. S2 in the manuscript (page 6) and discussed in pages 3- 4, lines 112-116.

Results: The expression of TLR2 on monocytes during the acute phase of DENV infection may vary according to the days post-fever. Therefore, we classified the DENV-positive patients

according to the number of days post-fever (day 2-5) and found that the TLR2 expression on CM, IM and NM was not dependent on the number of days post fever (Fig. S2).

Reviewers' Comments:

Reviewer #1:

Remarks to the Author:

The authors have addressed the main points raised in the previous review, and now have modified the manuscript accordingly. It is interesting to see that the binding to TLR2 by DENV is dependent on host and cell type factors, including differences in glycosylation patterns in those cells.

Reviewer #2:

Remarks to the Author:

The impression of formal prematurity of the primary manuscript version is largely absent from the revision. Moreover, numerous points such as omissions and discomposure have been addressed, for instance by updating of table 1 documenting patient characteristics and specification of statements such as “.. as dengue or dengue fever”. Yet despite of addition also of new data such as on healthy control – which express TLR2 at increased levels as compared to dengue infected patients - analyses demanded, documentation of bacteria absence from cell cultures performed is being provided in respect to mycobacterial infection, and specific specificity controls have been applied, persuasiveness of the revised as compared to the primary manuscript has not been increased as regards content sufficiently since respective central reviewer points were left open largely.

On the background of consideration of the concept “protein expression upregulation implicates the respective protein functionally, here in pattern recognition” the revised manuscript “still” fails to attempt to explain how general TLR2 upregulation correlates with TLR2 activation by some dengue strains but not by others wherein others than “2” have not been analyzed in detailed fashion because “Dengue virus serotype 2 is widely used in the field to study DENV pathogenesis due to the fact that it is usually associated with severe disease as well as in vivo murine models”. The persuasiveness of this argument appears as tightly limited such as against the background of general TLR2 upregulation yet TLR2 activation by infection with merely some strains. Whether these TLR2 activating ones differ in their pathologic potential - as could be expected – has seemingly not been put forward. Glycosylation difference is considered yet not addressed experimentally and not even exemplified “citationally”. Also, the argument according to which dengue “NS1 signals through TLR4 and not TLR2, as suggested by Chen et al in 2015, our data sheds new light on the contribution of TLR2 to DENV pathogenesis.” appears as somewhat implausible in that the author’s data merely “cut off” TLR4 involvement and a statement on these actual “author’s” TLR2 ligand identity is not being made

at least explicitly. It seems to rather obscure a potential knowledge profit towards the previous study cited.

In respect to TLR1 and TLR6 expression, the authors merely refer to information the vendor from which they purchased the HEK293 cells they applied provided while not providing originary evidence. However, TLR1 and TLR6 overexpression has failed to confer increased TLR2 ligand sensitivity in these cells previously putting into question functioning of these molecules in HEK293 cells. Rather, Ba/F3 cells have been used for studies in which dose kinetic TLR overexpression of these TLRs positively affected ligand sensitiveness. Application of such experimental system or – as urged on before already - blockade of TLRs and CD14 on primary cells such as macrophages would most certainly have provided more informative data as compared to endogenous TLR blockade on HEK293 cells. TLR1 and TLR6 besides TLR2 knockout (or at least k.-down) would have been an ultima ratio in this regard (a point brought up previously already).

Point 1: In respect to point "1", the authors bring up ".. or progress to a potentially fatal disease (severe dengue) also referred to as dengue hemorrhagic fever (DHF) and dengue shock syndrome (DSS)". Potential revision should clarify whether ".. or progress to a potentially fatal disease (severe dengue) also referred to as dengue hemorrhagic fever (DHF) or dengue shock syndrome (DSS)" or ".. or progress to a potentially fatal disease (severe dengue) encompassing dengue hemorrhagic fever (DHF) and dengue shock syndrome (DSS)" better denotes the point to be made.

2: Without naming it, the authors seem to counter point "2" by "Discussion: The seemingly contrasting role of TLR2 on CM and NM are likely to be linked to their overall distinct functions." which appears as limitedly informative. Whether the sentence is (or is not) obsolete should be considered.

4: a statement such as "Results: Chloroquine, chlorpromazine and dynasore were excluded from the experiments due to their high level of cytotoxicity (Fig. S4)." raises the question whether down-titration - rather than using "canonical" amounts possibly instrumental here – would not have been the solution for application of the agents also for HEK293 cells. Given the surfacial overexpression of TLR2 in them, endosome blockade should fail to affect its activation at least to a large degree (against the background of the citations on endosomal TLR functioning brought up in the rebuttal), which would contrast to the antibody driven blockade in a coherent manner.

5: in the response, the authors indicate "We would like clarify that nonstructural protein 3 (NS3) is one of the seven nonstructural proteins and as such is not expressed in the flavivirus virions." Is NS1 considered as TLR2 and/or TLR4 ligand

by others previously also not present in the mature virus particle? Considerations like this consideration might support narrowing down a DV virus component as TLR2 ligand candidate by these current authors. Also, to not apply murine cells because human – mouse species differences exist and therefore merely apply human cells falls too short since demonstration of TLR2 engagement as such by a biological entity even if it fails to infect mice or recruits different signal transducers will “still” be possible with murine cells or systems such as of respective TLR k.o. mice.

6: Rather than stating “..the coincidental association are very unlikely”, the authors should specify the comorbidities and infections instrumental for their patient population to enable estimation of a potential to activate TLR2.

12: “Interestingly, at the concentration tested only the intracellular accumulation of IL-1 β was significantly reduced by prior blocking of TLR2 blockade (Fig. S11A), suggesting differential TLR2- mediated pathways triggering the production of these cytokines.” should be specified.

13: The response such as in respect to “.. induced by the replicative or non-replicative of these viruses were donor-dependent rather preparation dependent” still appears as rather not explaining the TLR2 (and pathology?) specific differences on a ligand molecular level.

Reviewer #3:

Remarks to the Author:

The critical issue is to make the science right. In the field of the dengue, if there is evidence to support a statement that CD14+ classical monocytes are the main target of dengue virus infection in vivo, the pathogenic cause of plasma leakage in affected patients should be achieved already.

As such, the major issue in statement of the revised manuscript is " we found that on the highly permissive to DENV infection CD14+ classical monocytes, ---" in the abstract. If highly permissive, it should be very easy to isolate the CD14+ classical monocytes from acute dengue patients, and recovery the dengue virus in these sorted CD14+ monocytes handily.

It is important to demonstrate the statement in in vivo setting to confirm the CD14+ classical monocytes are "permissive" to dengue virus infection. Infection a cell does not result in production of infectious dengue virus. If a cell is permissive, indicating that this cell is not only infectable by dengue virus but also can produce

infectious dengue virus. As of today, there is no supporting evidence to demonstrate the statement. If the results in the current study is based upon the word "permissive", then the a much more study is needed in order to make the statement in the abstract.

As for the response to in vivo virus, if the current study is not feasible, a discussion with current status of in vivo finding of dengue virus should be made in order to make a fair presentation for scientific community.

Reviewers' comments:

Reviewer #1 (Remarks to the Author):

The authors have addressed the main points raised in the previous review, and now have modified the manuscript accordingly. It is interesting to see that the binding to TLR2 by DENV is dependent on host and cell type factors, including differences in glycosylation patterns in those cells.

Reviewer #2 (Remarks to the Author):

The impression of formal prematurity of the primary manuscript version is largely absent from the revision. Moreover, numerous points such as omissions and discomposure have been addressed, for instance by updating of table 1 documenting patient characteristics and specification of statements such as “.. as dengue or dengue fever”. Yet despite of addition also of new data such as on healthy control – which express TLR2 at increased levels as compared to dengue infected patients - analyses demanded, documentation of bacteria absence from cell cultures performed is being provided in respect to mycobacterial infection, and specific specificity controls have been applied, persuasiveness of the revised as compared to the primary manuscript has not been increased as regards content sufficiently since respective central reviewer points were left open largely.

On the background of consideration of the concept “protein expression upregulation implicates the respective protein functionally, here in pattern recognition” the revised manuscript “still” fails to attempt to explain how general TLR2 upregulation correlates with TLR2 activation by some dengue strains but not by others wherein others than “2” have not been analyzed in detailed fashion because “Dengue virus serotype 2 is widely used in the field to study DENV pathogenesis due to the fact that it is usually associated with severe disease as well as in vivo murine models”. The persuasiveness of this argument appears as tightly limited such as against the background of general TLR2 upregulation yet TLR2 activation by infection with merely some strains. Whether these TLR2 activating ones differ in their pathologic potential - as could be expected – has seemingly not been put forward. Glycosylation difference is considered yet not addressed experimentally and not even exemplified “citationally”. Also, the argument according to which dengue “NS1 signals through TLR4 and not TLR2, as suggested by Chen et al in 2015, our data sheds new light on the contribution of TLR2 to DENV pathogenesis.” appears as somewhat implausible in that the author’s data merely “cut off” TLR4 involvement and a statement on these actual “author’s” TLR2 ligand identity is not being made at least explicitly. It seems to rather obscure a potential knowledge profit towards the previous study cited.

In respect to TLR1 and TLR6 expression, the authors merely refer to information the vendor from which they purchased the HEK293 cells they applied provided while not providing ordinary evidence. However, TLR1 and TLR6 overexpression has failed to confer increased TLR2 ligand sensitivity in these cells previously putting into question functioning of these molecules in HEK293 cells. Rather, Ba/F3 cells have been used for studies in which dose kinetic TLR overexpression of these TLRs positively affected ligand sensitiveness. Application of such experimental system or – as urged on before already - blockade of TLRs and CD14 on primary cells such as macrophages would most certainly have provided more informative data as compared to endogenous TLR blockade on HEK293 cells. TLR1 and TLR6 besides TLR2 knockout (or at least k.-down) would have been an ultima ratio in this regard (a point brought up previously already).

Point 1: In respect to point “1”, the authors bring up “.. or progress to a potentially fatal disease (severe dengue) also referred to as dengue hemorrhagic fever (DHF) and dengue shock syndrome (DSS)”. Potential revision should clarify whether “.. or progress to a potentially fatal disease (severe dengue) also referred to as dengue hemorrhagic fever (DHF) or dengue shock syndrome (DSS)” or “.. or progress to a potentially fatal disease (severe dengue) encompassing dengue hemorrhagic fever (DHF) and dengue shock syndrome (DSS)” better denotes the point to be made.

2: Without naming it, the authors seem to counter point “2” by “Discussion: The seemingly contrasting role of TLR2 on CM and NM are likely to be linked to their overall distinct functionS.” which appears as limitedly informative. Whether the sentence is (or is not) obsolete should be considered.

4: a statement such as “Results: Chloroquine, chlorpromazine and dynasore were excluded from the experiments due to their high level of cytotoxicity (Fig. S4).” raises the question whether down-titration - rather than using “canonical” amounts possibly instrumental here – would not have been the solution for application of the agents also for HEK293 cells. Given the surficial overexpression of TLR2 in them, endosome blockade should fail to affect its activation at least to a large degree (against the background of the citations on endosomal TLR functioning brought up in the rebuttal), which would contrast to the antibody driven blockade in a coherent manner.

5: in the response, the authors indicate “We would like clarify that nonstructural protein 3 (NS3) is one of the seven nonstructural proteins and as such is not expressed in the flavivirus virions.” Is NS1 considered as TLR2 and/or TLR4 ligand by others previously also not present in the mature virus particle? Considerations like this consideration might support narrowing down a DV virus component as TLR2 ligand candidate by these current authors. Also, to not apply murine cells because human – mouse species differences exist and therefore merely apply human cells falls too short since demonstration of TLR2 engagement as such by a biological entity even if it fails to infect mice or recruits different signal transducers will “still” be possible with murine cells or systems such as of respective TLR k.o. mice.

6: Rather than stating “..the coincidental association are very unlikely”, the authors should specify the comorbidities and infections instrumental for their patient population to enable estimation of a potential to activate TLR2.

12: “Interestingly, at the concentration tested only the intracellular accumulation of IL-1 β was significantly reduced by prior blocking of TLR2 blockade (Fig. S11A), suggesting differential TLR2-mediated pathways triggering the production of these cytokines.” should be specified.

13: The response such as in respect to “.. induced by the replicative or non-replicative of these viruses were donor-dependent rather preparation dependent” still appears as rather not explaining the TLR2 (and pathology?) specific differences on a ligand molecular level.

Reviewer #3 (Remarks to the Author):

The critical issue is to make the science right. In the field of the dengue, if there is evidence to support a statement that CD14+ classical monocytes are the main target of dengue virus infection in vivo, the pathogenic cause of plasma leakage in affected patients should be achieved already.

As such, the major issue in statement of the revised manuscript is " we found that on the highly permissive to DENV infection CD14+ classical monocytes, ---" in the abstract. If highly permissive, it

should be very easy to isolate the CD14+ classical monocytes from acute dengue patients, and recover the dengue virus in these sorted CD14+ monocytes handily.

It is important to demonstrate the statement in in vivo setting to confirm the CD14+ classical monocytes are "permissive" to dengue virus infection. Infection a cell does not result in production of infectious dengue virus. If a cell is permissive, indicating that this cell is not only infectable by dengue virus but also can produce infectious dengue virus. As of today, there is no supporting evidence to demonstrate the statement. If the results in the current study is based upon the word "permissive", then the a much more study is needed in order to make the statement in the abstract.

As for the response to in vivo virus, if the current study is not feasible, a discussion with current status of in vivo finding of dengue virus should be made in order to make a fair presentation for scientific community.

Reviewer #1

Overall summary

The authors have addressed the main points raised in the previous review, and now have modified the manuscript accordingly. It is interesting to see that the binding to TLR2 by DENV is dependent on host and cell type factors, including differences in glycosylation patterns in those cells.

Response

We thank the reviewer for their comments and final positive assessment.

Reviewer #2

Overall summary

The impression of formal prematurity of the primary manuscript version is largely absent from the revision. Moreover, numerous points such as omissions and discomposure have been addressed, for instance by updating of table 1 documenting patient characteristics and specification of statements such as “.. as dengue or dengue fever”. Yet despite of addition also of new data such as on healthy control – which express TLR2 at increased levels as compared to dengue infected patients - analyses demanded, documentation of bacteria absence from cell cultures performed is being provided in respect to mycobacterial infection, and specific specificity controls have been applied, persuasiveness of the revised as compared to the primary manuscript has not been increased as regards content sufficiently since respective central reviewer points were left open largely.

On the background of consideration of the concept “protein expression upregulation implicates the respective protein functionally, here in pattern recognition” the revised manuscript “still” fails to attempt to explain how general TLR2 upregulation correlates with TLR2 activation by some dengue strains but not by others wherein others than “2” have not been analyzed in detailed fashion because “Dengue virus serotype 2 is wildly used in the field to study DENV pathogenesis due to the fact that it is usually associated with severe disease as well as in vivo murine models”. The persuasiveness of this argument appears as tightly limited such as against the background of general TLR2 upregulation yet TLR2 activation by infection with merely some strains. Whether these TLR2 activating ones differ in their pathologic potential - as could be expected - has seemingly not been put forward. Glycosylation difference is considered yet not addressed experimentally and not even exemplified “citationally”. Also, the argument according to which dengue “NS1 signals through TLR4 and not TLR2, as suggested by Chen et al in 2015, our data sheds new light on the contribution of TLR2 to DENV pathogenesis.” appears as somewhat implausible in that the author’s data merely “cut off” TLR4 involvement and a statement on these actual “author’s” TLR2 ligand identity is not being made at least explicitly. It seems to rather obscure a potential knowledge profit towards the previous study cited.

Response:

We thank the reviewer for his overall enthusiastic response to the revised manuscript. We are glad the reviewer acknowledges the addition of healthy donor controls and specificity controls. We agree that the analysis of other DENV serotypes in dengue pathogenesis is relevant. We focused on DENV2 as this serotype is the most prevalent etiologic agent in our and in many other cohorts (Vaughn *et al.*, 2000, Thomas *et al.*, 2008, Hamlin *et al.*, 2017). Importantly however, to address the comment of the reviewer we stratified TLR2 expression in patients based on infecting serotype. Unfortunately, the low numbers of patients for

DENV3 (n=3) and no patients positive for DENV4 in our cohort, hampered these analyses for these serotypes. Notably however, we found a similar association of TLR2 expression on classical monocytes in patients who developed severe disease following DENV1 and DENV2 while no differences were observed for intermediate and non-classical monocytes (new supplementary figure 3). These data support our premise that sustained high levels of TLR2 play a role in disease development during the acute phase of DENV infection.

We have included the bellow figure as a supplementary in page 8 and accordingly addressed in the results in page 4.

New supplementary Figure S3. Increased expression of TLR2 in CM from DENV1 and DENV2 positive patients correlates with DENV disease severity. (A and B) PBMCs were isolated from 45 patients undergoing acute DENV serotype 1 infection who developed relatively mild (DF, n=26) or severe (DHF/DSS, n=19) disease. Patients were classified based of infecting serotype, (A) DENV1 and (B) DENV2. Percentages of cells expressing TLR2 were determined for each monocyte subsets stratified by disease severity. Bars represent median with IQR. P values were obtained by Mann-Whitney test (*P< 0.05, **P<0.01).

Results:

Importantly, similar results were yielded when TLR2 expression was stratified based on the infecting serotype; CM from patients who developed DHF/DSS following DENV1 and DENV2 infections showed significantly higher expression of TLR2 when compared to those who developed DF (P<0.05) (Fig. S3 A and B, respectively) while no differences were observed for IM and NM (Fig. S3). Unfortunately, there were not enough patients to evaluate the correlation between TLR2 expression and disease severity following DENV serotypes 3 and 4.

In terms of changes in glycosylation patterns the reviewer might have overlooked our experiments with DCs- and macrophage-derived DENV2, which have been added to the manuscript following the first comments of reviewer 1 (panel B, Fig. S4) and, to which we referred the reviewer 2 as their comments overlapped. As mentioned in the previous response to the reviewer 1 “we fully agree that the difference in glycosylation of viral glycoproteins between the virus produced in mammalian or mosquito’s cells can have an impact on the activation of toll-like receptor 2 by these viruses. Therefore, to address this, we analyzed the ability of DENV2 16681 produced in monocyte (Mo)-derived dendritic cells (DC’s) and Mo-derived macrophages (M ϕ) to activate NF- κ B in the HEK-Blue hTLR2 cells and compared it to the standard virus preparation produced in mosquito C6/36 cells (panel B, Fig. S4). The viruses were obtained by infecting monocyte-derived DC’s and monocyte-derived M ϕ with C6/36-derived virus for 2 hours, after which the cells were washed to remove the surplus of inoculating virus and incubation was continued until 48hpi. Following titrations of DC’s-and M ϕ -derived DENV2 preparations, HEK-Blue™ hTLR2 cells were exposed to increasing numbers of virus particles [based on multiplicity of genomes (MOGs)] to ensure fair comparison. Interestingly, as shown in the figure below, DC’s-derived DENV2 induced NF- κ B activation on hTLR2 cells albeit approximately 3x lower than the C6/36-derived virus at a similar MOG, suggesting that differences in glycosylation pattern between human and mosquito-derived viruses indeed modulate the TLR2-dependency. Notably however, M ϕ -produced DENV2 did not induce NF- κ B activation at any of MOG tested, which implies that virus recognition by TLR2 is not solely modulated by differences between mosquito and mammalian cells but also intra human host, cell-type/tissue-specific factors leading to differences in virus maturation (Dejnirattisai et al., 2010; Parameswaran et al., 2017). It is important to mention however that due to the overall lower yield of DENV released by macrophages, MOG of 4000 could not be reached and thus more in-depth studies are needed to test the premise of intra-host cell type-specific factors influencing TLR2 activation by DENV. Noteworthy in that context are also the data summarized in Table 2 of the manuscript, showing the different capacity of various strains of DENV1-4 to activate TLR2 axis despite being produced in the mosquito C6/36 cells. All these data suggest that the activation of TLR2 by DENV particles is influenced by a strain and a cell type rather than solely by differences in glycosylation’s patterns between the mosquito and human cells”.

Panel B Fig. S4. Activation of HEK-Blue hTLR2 exposed to DENV2 derived from mosquito and human cells. The cells were exposed to increasing multiplicities of genome equivalent copies (MOG). C6/36 cells (MOG100, n=1; MOG500, n=1; MOG1000, n=3; MOG 3500, n=3), monocyte-derived dendritic cells (MOG250, n=1; MOG4000, n=2) and monocyte-derived macrophages for 24h (MOG250, n=1; MOG4000, n=2). NF-κB stimulation was assessed by QUANTI-Blue™, OD values show the induction of NF-κB. Data represent the mean ± SD. P values were obtained by paired one-tailed t test (*P<0.05, **** P<0.001)

We have included the above figure as panel B in the Fig. S4 (page 12) and the corresponding text (here below in blue) in the results section on page 9.

Results:

Considering the differential glycosylation patterns of DENV surface proteins during replication in primary human and mosquito cells (34), we next compared the ability of DENV-2 16681 produced in mosquito C6/36 cells to activate NF-κB in the HEK-Blue hTLR2 cells with that produced on monocyte-derived dendritic cells (DC's) and monocyte-derived macrophages (Mφ). The viruses were obtained by infecting monocyte-derived DC's and monocyte-derived Mφ with C6/36-derived virus for 2 hours, after which the surplus of inoculating virus and incubation was continued until 48hpi. Following titrations of DC-and Mφ -derived DENV2 preparations, HEK-Blue™ hTLR2 cells were exposed to increasing numbers of virus particles (MOGs) to ensure fair comparison. Interestingly, DC-derived DENV2 induced NF-κB activation on hTLR2 cells albeit approximately 3x lower than the C6/36-derived virus at a similar MOG (Fig. S4B), suggesting that differential glycosylation of the virus envelope proteins in human and mosquito cells modulates the TLR2 recognition. Notably however, Mφ-produced DENV2 did not induce NF-κB activation at any of MOG tested, implying that TLR2 engagement may be sensitive to changes in virion characteristics other than differences in glycosylation patterns between mosquito and human cells. It is important to mention however that due to the overall lower yield of DENV released by macrophages, MOG of 4000 could not be reached and thus more in-depth studies are needed

to test the premise of intra-host cell type-specific factors influencing TLR2 activation by DENV. Altogether, our data revealed that TLR2 has the capacity to sense DENV virions.

Herewith we showed that not only mosquito-derived but also human primary cell-derived virions activated TLR2 axis although the latter two to the lower levels. Thus, as discussed with reviewer 1, glycosylation as well as host-cell dependent changes in virions structure may influence TLR2-sensing.

[Redacted]

References

- Vaughn, DW., et al. Dengue viremia titer, antibody response pattern, and virus serotype correlate with disease severity. *J Infect Dis.* 181, 2-9 (2000).
- Zybert, IA., et al. Functional importance of dengue virus maturation: infectious properties of immature virions. *J Gen Virol*, 89: 3047-3051 (2008).
- Rodenhuis-Zybert, et al. Immature dengue virus: a veiled pathogen? *PLoS Pathog*, 6: e1000718 (2010).
- Dejnirattisai, W., et al. Cross-Reacting Antibodies Enhance Dengue Virus Infection in Humans. *Science*, 328 (5979): 745-748 (2010).
- Parameswaran, P., et al. Intrahost Selection Pressures Drive Rapid Dengue Virus Microevolution in Acute Human Infections. *Cell Host & Microbe*, 22: 400-410 (2017).
- Hamlin, Re., et al. High-dimensional CyTOF analysis of dengue virus-infected human DCs reveals distinct viral signatures. *JCI insight*. 2(13): e92424 (2017).
- Thomas, L., et al. Influence of the dengue serotype, previous dengue infection, and plasma viral load on clinical presentation and outcome during a dengue-2 and dengue-4 co-epidemic. *Am J Trop Med Hyg.* 78(6):990-998 (2018).

In respect to TLR1 and TLR6 expression, the authors merely refer to information the vendor from which they purchased the HEK293 cells they applied provided while not providing originary evidence. However, TLR1 and TLR6 overexpression has failed to confer increased TLR2 ligand sensitivity in these cells previously putting into question functioning of these molecules in HEK293 cells.

Rather, Ba/F3 cells have been used for studies in which dose kinetic TLR overexpression of these TLRs positively affected ligand sensitiveness. Application of such experimental system or – as urged on before already - blockade of TLRs and CD14 on primary cells such as macrophages would most certainly have provided more informative data as compared to endogenous TLR blockade on HEK293 cells. TLR1 and TLR6 besides TLR2 knockout (or at least k-down) would have been an ultima ratio in this regard (a point brought up previously already).

Response:

We are somewhat confused by this comment of the reviewer especially as it seems to be based on the uncited and hence unknown to us data from HEK293 and Ba/F3 cells. To our knowledge, several reports have used Ba/F3 to assess the potency of TLR4 and TLR2 ligands (Kawasaki *et al.*, 2000; Akashi *et al.*, 2001; Duesberg *et al.*, 2002; Youn *et al.*, 2006, Fujimoto *et al.*, 2009, Tawaratsumida *et al.*, 2009). Interestingly, in these studies HEK293 cells (Akashi *et al.*, 2001; Duesberg *et al.*, 2002; Youn *et al.*, 2006) or human PBMC's (Fujimoto *et al.*, 2009; Tawaratsumida *et al.*, 2009) have been also used in parallel to validate and support their

findings. Importantly, there have been different HEK293 reporter cell lines engineered by different groups (Cooper *et al.*, 2005; Gaudreault *et al.*, 2007; Shamsul *et al.*, 2010; Mistry *et al.*, 2015), which we cannot take responsibility of. Consequently, we fail to understand their relevance for our InvivoGen-derived cell line, which clearly shows TLR1 and TLR6 dependency to TLR2 ligands PAM3 and PAM2, respectively despite no ectopic expression of these co-receptors (panels B and C, Figure 2 in the manuscript).

Having said that, we certainly agree with the reviewer that the mechanism found using the overexpression system HEK-Blue™ hTLR2/CD14 cells may not represent the factual mechanism of TLR2 function in human monocytes of dengue patients, which has been the ultimate aim of our study. Therefore, we would like to stress the fact that we used HEK-Blue™ hTLR2/CD14 to study the proof of principle of TLR2 function in dengue sensing and subsequently verified it in monocytes in the context of PBMCs. As shown in Figure 3, panels C and D, TLR2 and TLR6 control the DENV-mediated upregulation of CD16 in human primary monocytes and TLR2 is in the control of inflammatory responses triggered following infection in PBMCs (new Figure 5). Importantly, we also validated the function of TLR2 axis in primary mononuclear cells by analyzing the production of inflammatory mediators in the supernatants of PBMCs exposed to DENV2 and its UV-inactivated DENV2 equivalent (new Figure 5 and new supplementary figures S14 and S15 showing the respective isotype control). Consequently, we identified TLR2-axis dependent cytokines and chemokines induced in the course of infection, including TNF- α , IL-6, IL-10, IP-10, IFN- α 2, IFN- β , IFN- γ and IFN- λ 1 (new supplementary figure S14). Interestingly, UV-DENV moderately induced the production of TNF- α , IL-1 β , IL-6 and IFN- λ 2/3 when compared to the mock-treated PBMCs. Notably, the results obtained in PBMC did not entirely mirror the results obtained in HEK-Blue™ hTLR2 cells. The differences in cytokine levels produced between active DENV infection and UV-DENV in PBMCs may thus reflect the differential expression as well as cross talk of different PRR's in sensing replicative and non-replicative virions, which do not occur in HEKs. Moreover, the level of NF- κ B/AP1 activation measured in the reporter cell lines does not reflect variety of different immunomodulators that can be released down-stream of TLR2 in primary immune cells. Indeed, the blocking of TLR2 and CD14 significantly decreased the production of IFN- β and IFN- λ 1, and reduced production of TNF- α (new Figure 5) while, TLR1/6 or the isotype controls had no such effect (Fig. S15). Interestingly, the TLR2 axis block did not have an effect on the IL-1 β , IL-6, IL-10 and IFN- γ concentrations induced by DENV infection (new supplementary figure S16) suggesting that other pathways are involved in their production (Re *et al.*, 2004, Hochdorfer *et al.*, 2011, Lin *et al.*, 2017). Importantly, these additional data are in line with the DENV replication- dependent and TLR2-mediated production of inflammatory mediators capable to activate HUVEC *in vitro* (Figure 4 in the manuscript). In addition, following the recommendation of the reviewer, by increasing the time of infection to 48h in human PBMC's, we were able to detect DENV positive monocytes (new panel F, Figure 2) and thus verify the role of the TLR2 axis in DENV infection. Indeed, in line with our experiments in HEK-Blue™ hTLR2 cells, blocking TLR2 and CD14 significantly decreased the frequency of infected monocytes (panel F, updated Figure 2).

Together with the additional acquired results in primary cells, our data collectively indicate that the TLR2 axis plays a key role in the immune responses induced by DENV in human primary cells. The aforementioned new and/or updated figures have been included in the manuscript (pages 11, 13, 23, 25-27) and discussed accordingly (pages 9-10, 20 and 30).

We also agree with the reviewer that it will be important to evaluate the effect of DENV infection on TLR2 expression and its activity in other relevant target cells. In fact, as mentioned in the introduction on page 2, line 69, Torres *et al*, 2013 have observed an increased TLR2 expression on plasmacytoid dendritic cells but not myeloid dendritic cells of dengue virus infected patients. Importantly however, in our pediatric cohort, we did not observe a significant expression of TLR2 on cells other than monocytes and thus we focused on this cell type. Considering the inherent differences (age, ethnicity) between the cohorts analyzed in our study and that of Torres *et al*, we propose that new studies should address their relevance in a side-by-side comparison of different cohorts. Furthermore, we are currently characterizing the effect of DENV infection on the upregulation/activation of TLR2 in human monocyte-derived M ϕ , and murine (TLR2/6 KO) bone-marrow derived macrophages. Importantly however, considering profound differences in the expression level of the TLR2 axis found in these cells when compared to human monocytes, the additional analyses needed to form relevant conclusions are far beyond the scope of this manuscript and thus constitute a separate study.

Panel F, Figure 2. DENV infection in human primary monocytes is TLR2/CD14 dependent. PBMC's from healthy donors (n= 2) were (mock) - treated with αTLR2, αTLR1, αTLR6 (5 μg/mL) and αCD14 (3 μg/mL) for 2h prior to infection with DENV2 at MOI of 20 (n=6) for 48h. Percentages of DENV-(E) - positive cells were determined by flow cytometry. Data represent the mean ± SEM. P values were obtained by one-way ANOVA, Dunnett post hoc test (**P<0.001, ***P<0.0001).

New supplementary Figure S6. DENV2 but not UV-inactivated DENV2 infects human primary monocytes. (A and B) PBMC's from healthy donors (n= 2) were (mock) - treated with α TLR2 and α TLR1/6 Isotype controls (5 μ g/mL) for 2h prior to infection with DENV2 at MOI of 20 (n=6) or its UV- inactivated equivalent (UV-DENV2) (n= 6) for 48h. Percentages of DENV-(E) - positive cells were determined by flow cytometry. Data represent the mean \pm SEM. P values were obtained by one-tailed paired t test (***)P<0.001).

New Figure 5. TLR2/CD14-dependent cytokines induced by DENV2 infection. PBMCs from healthy donors (n= 2) were (mock) - treated with α TLR2, α TLR1, α TLR6 (5 μ g/mL) and α CD14 (3 μ g/mL) for 2h prior to infection with DENV2 at MOI of 20 (n=2-3) for 48h. Cytokine production was measured by flow cytometry using LegendPlex. Each graph shows the production in pictograms per milliliter (pg/mL) of the respective cytokine. P values were obtained by one-way ANOVA, Dunnett post hoc test (**P<0.01).

New supplementary Figure S14. Differential cytokine production by DENV2 and UV-inactivated DENV2. PBMCs from healthy donors (n= 2) were (mock) - infected with DENV2 at MOI of 20 (n= 3) or its UV- inactivated equivalent (UV-DENV2) (n= 3) for 48h. Cytokine production was measured by flow cytometry using LegendPlex. Each graph shows the production in pictograms per milliliter (pg/mL) of the respective cytokine. P values were obtained by one-tailed paired t test (*P<0.05, **P<0.01).

New supplementary Figure S15. α TLR2 isotype do not impair the DENV2 infection-induced production of cytokines. PBMCs from healthy donors (n= 2) were (mock) - treated with α TLR2 isotype control (5 μ g/mL) for 2h prior to infection with DENV2 at MOI of 20 (n=2-3) for 48h. Cytokine production was measured by flow cytometry using LegendPlex. Each graph shows the production in pictograms per milliliter (pg/mL) of the respective cytokine.

New supplementary figure S16. Active DENV2 infection-induced cytokines not in control of TLR2/CD14. PBMCs from healthy donors ($n=2$) were (mock) - treated with α TLR2, α TLR1, α TLR6, α TLR2 isotype control ($5 \mu\text{g/mL}$) and α CD14 ($3 \mu\text{g/mL}$) for 2h prior to infection with DENV2 at MOI of 20 ($n=2-3$) for 48h. Cytokine production was measured by flow cytometry using LegendPlex. Each graph shows the production in pictograms per milliliter (pg/mL) of the respective cytokine.

Results:

As shown in Fig. 2E, blockage of TLR2, CD14 ($P<0.0001$) and to a lesser extent that of TLR6 ($P<0.05$), reduced the number of infected HEK-Blue™ hTLR2 cells. Importantly, consistent with these results, in PBMCs specific blocking of TLR2 ($P<0.0001$) and CD14 ($P<0.001$) but not that of TLR1/6 significantly decreased the frequency of infected monocytes (Fig. 2F and Fig. S6A for UV-inactivated DENV2, and Fig. S6B for isotype controls).

Therefore, we analyzed inflammatory mediators released from the cells throughout 48h of infection. Interestingly, the concentrations of IP-10 ($P<0.01$), IFN- α 2 ($P<0.05$), IFN- β ($P<0.05$) and IFN- λ 1 ($P<0.01$) were significantly higher in the supernatants from DENV2-infected PBMCs in comparison from those of UV-DENV2-stimulated PBMCs (Fig. S14), while only a slight increase was observed for TNF- α , IL-6, IL-10 and IFN- γ (Fig. S14). Additionally, UV-DENV2 also induced production of TNF- α , IL-1 β , IL-6 and IFN- λ 2/3 (Fig. S14), however

not as potently as the replicative virus. Remarkably, blocking TLR2 and CD14 but not TLR1/6 or the use of control antibody (Fig. 5 and Fig. S15), significantly decreased the production of IFN- β ($P < 0.01$) and IFN- $\lambda 1$ ($P < 0.01$) induced by active DENV infection, while the production of TNF- α was marginally reduced in these conditions (Fig. 5). The levels of IFN- $\alpha 2$ were moderately reduced when blocking CD14 but not when blocking other TLR2 (co-) receptors, while only the block of TLR2 was able to reduce the levels of IP-10 (Fig. 5). The TLR2 axis block did not impair the levels of IL-1 β , IL-6, IL-10 and IFN- γ induced by DENV infection (Fig. S16). Altogether, these results support the DENV replication- dependent and TLR2-mediated production of inflammatory mediators are capable to activate endothelial cells in vitro (Fig. 4) and highlight that variety of pathways cross-talking following TLR2 axis-dependent and independent sensing of DENV infection. In addition, it is important to note that the observed variations in cytokine production following (UV-) DENV infections were donor- rather than virus preparation-dependent, further highlighting the inherent differences in TLR2-mediated sensing of DENV infection between the overexpression system HEK-Blue™ hTLR2 and primary monocytes, which are equipped with multiple PRRs. Altogether, our data show that TLR2 sensing of dengue virus infection induces production of inflammatory mediators, which in turn can activate endothelial cells.

Discussion:

Sensing of DENV infection by TLR2 did not rely on DENV infection as UV-irradiated virus was also sensed by the TLR2 expressed on HEK-Blue™ hTLR2 cells. Notably, the results obtained in PBMCs did not entirely mirror the results obtained in HEK-Blue™ TLR2 cells in which both virus preparation triggered equal NF- κ B activation. The level of NF- κ B/AP1 activation measured in the reporter cell lines is unlikely to reflect the variety of different immunomodulators that can be released down-stream of TLR2 in primary immune cells. The differences in cytokine levels produced between active DENV infection and UV-DENV in PBMCs may reflect the differential expression, as well as the crosstalk of different PRR's in sensing replicative and non-replicative virions, which do not occur, or are not detectable in the NF- κ B/AP-1 reported system (46, 60). Indeed, by UV-irradiating the virus, we likely impeded the endosomal sensing of viral ssRNA by TLR7/8, which ultimately might have affected the potentiating effect of TLR2/7/8 cross talk (61). Moreover, the TLR2/CD14 dependent production of IFN- $\lambda 1$, which has been described to be released by dendritic cells in a TLR3 dependent manner (62), may suggest the possibility of TLR2/TLR3 cross talk (63). Indeed, in human DC's the stimulation of TLR2 blocked the induction of cytokines that are controlled by TLR3 (45). Further studies are required to elucidate the cross talk of different PRR's in the course of DENV infection.

Material and Methods:

Cytokine and chemokine determination

Protein levels of IL-1 β , TNF- α , IL-6, IL-8, IL-10, IL-12p70, IP-10, GM-CSF, IFN- $\alpha 2$, IFN- β , IFN- γ , IFN- $\lambda 1$ and IFN- $\lambda 2/3$ were determined in cell-free supernatants using the human anti-virus

response panel (13-plex, LEGENDplex™, BioLegend). Data were collected using a FACSVerser flow cytometer (BD Biosciences) and analyzed using LEGENDplex™ v8.0 (BioLegend).

References

Kawasaki, K., *et al.* Mouse toll-like receptor 4. MD-2 complex mediates lipopolysaccharide-mimetic signal transduction by taxol. *J Biol Chem*, 275 (4): 2251-2254 (2000).

Akashi, S., *et al.* Human MD-2 confers on mouse Toll-like receptor 4 species-specific lipopolysaccharide recognition. *Int Immunol*, 13 (12): 1595-1599 (2001).

Duesberg, Uta., *et al.* Cell activation by synthetic lipopeptides of the hepatitis C virus (HCV)—core protein is mediated by toll like receptors (TLRs) 2 and 4. *Immunol Lett*, 84(2): 89-95 (2002).

Re, F., *et al.* IL-10 Released by concomitant TLR2 stimulation blocks the induction of a subset of Th1 cytokines that are specifically induced by TLR4 or TLR3 in human dendritic cells. *J Immunol*, 173, 7548–7555 (2004).

Cooper, A., *et al.* Cytokine induction by the hepatitis B virus capsid in macrophages is facilitated by membrane heparan sulfate and involves TLR2. *J Immunol*, 175: 3165-3176 (2005).

Gaudreault, E., *et al.* Epstein-Barr virus induces MCP-1 secretion by human monocytes via TLR2. *J Virol*, 81(15): 8016-8024 (2007)

Youn, H S., *et al.* Inhibition of homodimerization of Toll-like receptor 4 by curcumin. *Biochem pharmacol*, 72:62-69 (2006).

Fujimoto, Y., *et al.* Lipopeptides from Staphylococcus aureus as Tlr2 Ligands: Prediction with mRNA Expression, Chemical Synthesis, and Immunostimulatory Activities. *Chem Bio Chem*, 10(14): 2311-2315 (2009).

Tawaratsumida, K., *et al.* Characterization of N-terminal structure of TLR2-activating lipoprotein in Staphylococcus aureus*. *J Biol Chem*, 284(14): 9147-9152 (2009).

Shamsul, H Q., *et al.* The Toll-like receptor 2 (TLR2) ligand FSL-1 is internalized via the clathrin-dependent endocytic pathway triggered by CD14 and CD36 but not by TLR2. *Immunology*, 130(2): 262-272 (2010).

Hochdorfer, T., *et al.* Activation of the PI3K pathway increases TLR-induced TNF- α and IL-6 but reduces IL-1 β production in mast cells. *Cell Signal*, 23(5): 866-875 (2011).

Torres, S., *et al.* Differential expression of toll-like receptors in dendritic cells of patients with dengue during early and late acute phases of the disease. *PLoS Pathog.* 11, e1005053 (2013).

Mistry, P., *et al.* Inhibition of TLR2 signaling by small molecule inhibitors targeting a pocket within the TLR2 TIR domain. *PNAS*, 112(17): 5455-5460 (2015).

Lin, B., *et al.* Systematic Investigation of Multi-TLR Sensing Identifies Regulators of Sustained Gene Activation in Macrophages. *Cell Syst.* 5, 25-37 (2017).

Specific comments

1. In respect to point “1”, the authors bring up “.. or progress to a potentially fatal disease (severe dengue) also referred to as dengue hemorrhagic fever (DHF) and dengue shock syndrome (DSS)”. Potential revision should clarify whether “.. or progress to a potentially fatal disease (severe dengue) also referred to as dengue hemorrhagic fever (DHF) or dengue shock syndrome (DSS)” or “.. or progress to a potentially fatal disease (severe dengue) encompassing dengue hemorrhagic fever (DHF) and dengue shock syndrome (DSS)” better denotes the point to be made.

Response:

We have adjusted the text in order to clarify that severe disease refers to both DHF and DSS.

Introduction:

Clinical outcomes of DENV infection vary considerably and can be either limited to an acute febrile illness referred to as dengue fever (DF), or progress to a potentially fatal disease (severe dengue) encompassing dengue hemorrhagic fever (DHF) and dengue shock syndrome (DSS).

2. Without naming it, the authors seem to counter point “2” by “Discussion: The seemingly contrasting role of TLR2 on CM and NM are likely to be linked to their overall distinct functionS.” which appears as limitedly informative. Whether the sentence is (or is not) obsolete should be considered.

Response:

To improve the readability, we have briefly recapped the function of the different monocytes subsets, which was elaborated on in the discussion (page 28).

Discussion:

The seemingly contrasting role of TLR2 on CM and NM are likely to be linked to their overall distinct function. CM are equipped with various PRRs and scavenger receptors, recognizing PAMPS thereby removing microorganisms, lipids, and dying cells via phagocytosis and thus involved in sensing and inflammatory response to stress-inducing factors. NM or patrolling monocytes on the other hand play a unique role in protecting endothelial integrity (49–51) and once activated, they differentiate into anti-inflammatory macrophages to repair damaged tissues (52).

4. a statement such as “Results: Chloroquine, chlorpromazine and dynasore were excluded from the experiments due to their high level of cytotoxicity (Fig. S4).” raises the question whether down-titration - rather than using “canonical” amounts possibly instrumental here - would not have been the solution for application of the agents also for HEK293 cells. Given the surficial overexpression of TLR2 in them, endosome blockade should fail to affect its activation at least to a large degree (against the background of the citations on endosomal TLR functioning brought up in the rebuttal), which would contrast to the antibody driven blockade in a coherent manner.

Response:

We apologize for not being clear enough. For chloroquine, chlorpromazine and dynasore we could not find concentrations that would be relatively non-toxic yet still inhibitory. Importantly, we do show that a more specific inhibitor of clathrin-pit formation, pit-stop (PS), as well as NH₄Cl that neutralizes the pH of intracellular compartments significantly reduced but did not abrogate NF-κB activation mediated by PAM3 (p<0.05) and DENV (p<0.05), whereas no significant effect was seen for PAM2 (panel D, Figure 2). Thus, in line with our current understanding of TLR2/CD14 signaling, clathrin-mediated endocytosis does potentiate NF-κB activation (Shamsul, *et al.* 2010).

Reference

Shamsul, H Q., *et al.* The Toll-like receptor 2 (TLR2) ligand FSL-1 is internalized via the clathrin-dependent endocytic pathway triggered by CD14 and CD36 but not by TLR2. *Immunology*, 130(2): 262-272 (2010).

5. in the response, the authors indicate “We would like clarify that nonstructural protein 3 (NS3) is one of the seven nonstructural proteins and as such is not expressed in the flavivirus virions.” Is NS1 considered as TLR2 and/or TLR4 ligand by others previously also not present in the mature virus particle? Considerations like this consideration might support narrowing down a DV virus component as TLR2 ligand candidate by these current authors. Also, to not apply murine cells because human - mouse species differences exist and therefore merely apply human cells falls too short since demonstration of TLR2 engagement as such by a biological entity even if it fails to infect mice or recruits different signal transducers will “still” be possible with murine cells or systems such as of respective TLR k.o. mice

Response:

We would like to note that NS1 is a non-structural protein and thus is not expressed on the virus particle. Therefore, a ligand of TLR2 in the virions are likely to be one or both proteins expressed on the virus envelope such as E and/or pr(M). We have clarified it further on the introduction (page 2) and discussion (page 29). We agree with the reviewer that using the murine system it will still be possible to demonstrate TLR2 engagement by DENV. However, considering the inherent differences in TLR2 axis expression in human monocyte-derived

M ϕ , and murine (TLR2/6 KO) bone-marrow derived macrophages found in these cells when compared to human monocytes, the additional analyses needed to form relevant conclusions are far beyond the scope of this manuscript and thus constitute a separate and an ongoing study. We would like to refer the reviewer to page 12 of this document where we discussed this point in detail. Importantly, our additional experiments with human PBMC's showing that in monocytes DENV infection relied on the engagement of TLR2 and CD14 corroborated the significance of TLR2 axis binding in the course of infection.

Introduction:

Chen *et al*, attributed TLR2-activation following DENV infection to one of the viral proteins, non-structural protein 1 (NS1) that is released from cells replicating the virus (24).

Discussion:

The engagement of TLR2 was independent of the ability of DENV to replicate and relied on the sensing of the viral particle suggesting the involvement of structural proteins expressed on the virus surface [E, (pr)M] in the engagement of this PRR rather than any of the five non-structural proteins (NS1- NS5).

6. Rather than stating “..the coincidental association are very unlikely”, the authors should specify the comorbidities and infections instrumental for their patient population to enable estimation of a potential to activate TLR2.

Response

We agree with the reviewer, however as most co-morbidities including bacterial co-infections are observed in the elderly, the role of co-morbidities in a pediatric cohort remains speculative. We have added this information to the discussion (page 31).

Discussion:

Lastly, considering the role of TLR2 and TLR4 in detecting bacterial PAMPS, it will be important to address the role of common co-morbidities including bacterial and parasitic co-infections or microbial translocation (6, 7, 68, 69), during DENV infection in patients.

12. “Interestingly, at the concentration tested only the intracellular accumulation of IL-1 β was significantly reduced by prior blocking of TLR2 blockade (Fig. S11A), suggesting differential TLR2- mediated pathways triggering the production of these cytokines.” should be specified.

Response:

We have included the possible mechanisms underlying this findings (Re *et al.*, Hochdorfer *et al.*, 2011, Lin *et al.*, 2017).

References

Re, F., *et al.* IL-10 Released by concomitant TLR2 stimulation blocks the induction of a subset of Th1 cytokines that are specifically induced by TLR4 or TLR3 in human dendritic cells. *J Immunol*, 173, 7548–7555 (2004).

Hochdorfer, T., *et al.* Activation of the PI3K pathway increases TLR-induced TNF- α and IL-6 but reduces IL-1 β production in mast cells. *Cell Signal*, 23(5): 866-875 (2011).

Lin, B., *et al.* Systematic Investigation of Multi-TLR Sensing Identifies Regulators of Sustained Gene Activation in Macrophages. *Cell Syst*. 5, 25-37 (2017).

Results:

Interestingly, at the concentration tested only the intracellular accumulation of IL-1 β was significantly reduced by prior TLR2 blockade (Fig. S13A), suggesting that differential pathways down-stream of TLR2- trigger the production of these cytokines (44). Indeed, TLR2 may also lead to the activation of the inflammasome pathway that is likely to contribute to IL-1 β production. Alternatively, signal integration of different PRR can lead to different and non-additive immune responses (45, 46).

13. The response such as in respect to “.. induced by the replicative or non-replicative of these viruses were donor-dependent rather preparation dependent” still appears as rather not explaining the TLR2 (and pathology?) specific differences on a ligand molecular level.

Response:

As suggested by the reviewer we have elaborated on the differences between the responses to the replicative and non-replicative virus. We would like to refer to page 11 where we in detail addressed the differences in the levels of cytokines induced by active DENV infection and UV-DENV. Briefly, we postulate that such differences may reflect the crosstalk between TLR2 and other different PRR's including endosomal sensors like TLR3 or TLR7/8 involved in sensing active DENV infection or virions with native ssRNA PAMPs. In contrast, the non-replicative UV-DENV, in which the viral RNA structure has been compromised, will not be detected by these receptors (Wang *et al.*, 2006; Sun *et al.*, 2009). We have addressed this point in a new paragraph of the discussion (page 30).

References

Wang, JP., *et al.* Flavivirus activation of plasmacytoid dendritic cells delineates key elements of TLR7 signaling beyond endosomal recognition. *J Immunol*, 177(10): 7114-7121 (2006).

Sun, P., *et al.* Functional characterization of ex vivo blood myeloid and plasmacytoid dendritic cells after infection with dengue virus. *Virology*, 383: 207-215 (2009).

Discussion:

Sensing of DENV infection by TLR2 did not rely on DENV infection as UV-irradiated virus was also sensed by the TLR2 expressed on HEK-Blue™ hTLR2 cells. Notably, the results obtained in PBMCs did not entirely mirror the results obtained in HEK-Blue™ TLR2 cells in which both virus preparation triggered equal NF-κB activation. The level of NF-κB/AP1 activation measured in the reporter cell lines is unlikely to reflect the variety of different immunomodulators that can be released down-stream of TLR2 in primary immune cells. The differences in cytokine levels produced between active DENV infection and UV-DENV in PBMCs may reflect the differential expression, as well as the crosstalk of different PRR's in sensing replicative and non-replicative virions, which do not occur, or are not detectable in the NF-κB/AP-1 reported system (46, 60). Indeed, by UV-irradiating the virus, we likely impeded the endosomal sensing of viral ssRNA by TLR7/8, which ultimately might have affected the potentiating effect of TLR2/7/8 cross talk (61). Moreover, the TLR2/CD14 dependent production of IFN-λ1, which has been described to be released by dendritic cells in a TLR3 dependent manner (62), may suggest the possibility of TLR2/TLR3 cross talk (63). Indeed, in human DC's the stimulation of TLR2 blocked the induction of cytokines that are controlled by TLR3 (45). Further studies are required to elucidate the cross-talk of different PRR's in the course of DENV infection.

Reviewer #3

Overall summary

The critical issue is to make the science right. In the field of the dengue, if there is evidence to support a statement that CD14+ classical monocytes are the main target of dengue virus infection *in vivo*, the pathogenic cause of plasma leakage in affected patients should be achieved already.

Response:

We completely agree with the reviewer that the most important is to make the science right. We would like to clarify the statement on page 10 (lines 301-303): "TLR2+CD14++ monocytes represent the main target cells during acute DENV infection and why CD14 expression correlates with DENV infection *in-vivo* (Fig. 2F)"; we meant the main targets out of the whole monocyte pool. We have corrected the sentence to remove this ambiguity.

Results:

Remarkably however, at the same time, TLR2/CD14-mediated CME facilitates DENV infection, which may explain why, among the three monocyte subsets, the TLR2-expressing CD14++ monocytes (Fig. 1C) represent the main target cells during acute DENV infection and why CD14 expression correlates with DENV infection *in vivo* (Fig. 2G).

As such, the major issue in statement of the revised manuscript is " we found that on the highly permissive to DENV infection CD14+ classical monocytes, ---" in the abstract. If highly permissive, it should be very easy to isolate the CD14+ classical monocytes from acute dengue patients, and recovery the dengue virus in these sorted CD14+ monocytes handily. It is important to demonstrate the statement in *in vivo* setting to confirm the CD14+ classical monocytes are "permissive" to dengue virus infection. Infection a cell does not result in production of infectious dengue virus. If a cell is permissive, indicating that this cell is not only infectable by dengue virus but also can produce infectious dengue virus. As of today, there is no supporting evidence to demonstrate the statement. If the results in the current study is based upon the word "permissive", then the a much more study is needed in order to make the statement in the abstract.

Response:

We agree with the point raised by the reviewer. We have added the experiments showing the contribution of TLR2 and CD14 to DENV infection in primary human monocytes (Panel F, updated figure 2 and new supplementary figure 6) (please refer to pages 12-13 of this document). This together, with other studies documenting the ability of monocytes to support virus replication and/or active production of virus progeny (Chen et al., 2002; Chao et al., 2008; Michlmayr et al., 2017) suggested to us that monocytes are permissive to DENV

infection. However, as pointed out by the reviewer, our analyses *ex-vivo* and detection of NS3 protein imply an ongoing replication rather than productive infection per se. Based on this and other data (Halstead et al, 1977), we have adjusted the word “permissive” to “susceptible” and addressed this matter in the discussion (page 29).

References

Halstead, SB., et al. Dengue viruses and mononuclear phagocytes. I. Infection Enhancement by non-neutralizing antibody. J Exp Med, 146: 201-217 (1977).

Chen, Y-C., et al. Activation of terminally differentiated human monocytes/macrophages by dengue virus: productive infection, hierarchical production of innate cytokines and chemokines, and the synergistic effect of lipopolysaccharide. J Virol, 76(19): 9877-9887 (2002).

Chao, Y-C., et al. Higher infection of dengue virus serotype 2 in human monocytes of patients with G6PD deficiency. PLoS One, 3(2): e1557 (2008).

Michlmayr, D., et al. CD14+CD16+ monocytes are the main target of Zika virus infection in peripheral blood mononuclear cells in a pediatric study in Nicaragua. Nat Microbiol, 2: 1462-1470 (2017).

Abstract:

By analyzing TLR2 expression on mononuclear blood cells isolated in an acute phase from DENV-infected pediatric patients, we found that on the susceptible to DENV infection CD14++ classical monocytes, TLR2 expression correlated with severe disease development.

Discussion:

Surprisingly, however, blocking TLR2 or CD14 had no appreciable effect on viral production, suggesting that the few cells that were infected produced more virions per cell, or that other cell populations present in a very low frequency, and thus undetected by flow cytometry, are responsible for virus production (58). Interestingly, in our patient’s cohort, high viral titers were present in children with DF compared to those that subsequently developed DHF. The underlying mechanism and relevance of these findings requires further investigation; however, it is likely to be due to the reduced or altered immune activation including antiviral responses following the TLR2 blockage or in case of natural infection, severely reduced TLR2 surface expression in DF patients as compared to DHF/DSS.

As for the response to in vivo virus, if the current study is not feasible, a discussion with current status of in vivo finding of dengue virus should be made in order to make a fair presentation for scientific community.

Response:

We have included the recommended update of the recent *in vivo* findings into the discussion (page 30).

Discussion:

Unfortunately, due to the limited amount of sample obtained from our pediatric cohort, we could not assess the effect of the human circulating virus in our *in vitro* systems. Interestingly, Raut *et al* (59), showed that DENVs serotype 1 circulating in humans are much more infectious when compared to the DENVs produced in laboratory cell lines. Moreover, the human circulating DENV1 virions showed a higher degree of maturity than those cultured *in vitro*. Considering the reduced ability of human primary cell derived DENV2 to engage TLR2 described in our studies, it will be important to address how maturation levels of the virions influence the course of DENV infections. We are currently assessing the role of immature dengue particles in our *in vitro* models.

Reviewers' Comments:

Reviewer #2:

Remarks to the Author:

A coincidence of protein expression upregulation is a rather weak argument towards function and involvement of the upregulated molecule in the particular pattern recognition weakened further by the finding that even strains not involving TLR2 in their recognition "still" upregulate the expression of it. Still there is a very substantial fuzziness in the various statements on different glycosylation in virus producing and other cells against the continuative openness of the question for the identity of the glycosylation that drives the respective PRR activation or at least listing of literature borne candidate molecules. Global enzymatic deglycosylation of for instance homogenates of TLR2 activating virus strains to render them as non stimulatory as the TLR2 nonstimulatory counterparts would add substantial impact to the statement made by the authors. Also, implication of CD14, TLR1 or/and TLR6 involvement would be way more convincing by k.o. or at least individual (rather than merely anti TLR2) isotype control application. "The seemingly contrasting role of TLR2 on CM and NM are likely to be linked to their overall distinct function." might be revisable such as towards "The seemingly contrasting role of TLR2 on CM and NM is likely due to the latter's distinct functions.". Furthermore, the title "...senses dengue virions and mediates the innate 2 inflammatory response underlying disease pathogenesis" appears as somewhat tautological on the one and over-interpretative on the other hand. Specifically, "senses" already encompasses "mediates inflammatory response" and "underlying disease pathogenesis" underrates (a potential) clearance mediation in clinically unremarkably infected (→ self cleared) individuals.

Reviewer #3:

Remarks to the Author:

The authors did not address the concerns raised previously by the reviewer. The fundamental issue is that the natural biological function of monocytes is phagocytosis of unwanted cells or cell debris. The infected monocytes defined in the current manuscript by the presence of dengue viral antigens are insufficient to address the infection of these cells by the dengue virus. These could be a consequence of phagocytosis of infected cells by these monocytes. According to Fig. 1, the NS3+ cells in monocyte populations were between 60 and 100 percent. It should be very easy to clarify whether these monocytes were infected directly by dengue virus or due to the activity of phagocytosis of dengue virus infected by these monocytes. The authors should sort out these cells and perform co-culture to recover the infectious dengue virus to clarify the observed infected phenomenon;

was it due to phagocytose other infected cells or due to direct infection by the dengue virus??

In addition, time is sensitive issue in dengue pathogenesis. The attached file demonstrates that the time lapse from the bite of mosquito carrying infectious dengue virus to the time of rash develop, on average is 7 to 10 days. Kinetically, the peak of dengue viremia is during the febrile stage. As addressed in the current manuscript, the subjects enrolled to the study were on average 3.3 to 3.9 days after onset of the fever, indicating that the viral roads were downward bound. Hence, it is imperative to clarify whether the current findings are the consequences of infection of dengue virus in these monocytes or as claimed to be the instigator for the disease severity. It is important to know that correlation does not constitute to the cause of disease development.

Minor points:

1. In Table 1. The data were collected in dengue endemic country. It is important to include the immune status of those healthy donors in the table.
2. How specific of the home-made NS3 antibody for FACS analysis in these monocytes?

Reviewer #2

Overall summary

A coincidence of protein expression upregulation is a rather weak argument towards function and involvement of the upregulated molecule in the particular pattern recognition weakened further by the finding that even strains not involving TLR2 in their recognition “still” upregulate the expression of it. Still there is a very substantial fuzziness in the various statements on different glycosylation in virus producing and other cells against the continuative openness of the question for the identity of the glycosylation that drives the respective PRR activation or at least listing of literature borne candidate molecules. Global enzymatic deglycosylation of for instance homogenates of TLR2 activating virus strains to render them as non stimulatory as the TLR2 nonstimulatory counterparts would add substantial impact to the statement made by the authors. Also, implication of CD14, TLR1 or/and TLR6 involvement would be way more convincing by k.o. or at least individual (rather than merely anti TLR2) isotype control application. “The seemingly contrasting role of TLR2 on CM and NM are likely to be linked to their overall distinct function.” might be revisable such as towards “The seemingly contrasting role of TLR2 on CM and NM is likely due to the latter’s distinct functions.”. Furthermore, the title “..senses dengue virions and mediates the innate 2 inflammatory response underlying disease pathogenesis” appears as somewhat tautological on the one and over-interpretative on the other hand. Specifically, “senses” already encompasses “mediates inflammatory response” and “underlying disease pathogenesis” underrates (a potential) clearance mediation in clinically unremarkably infected (→ self cleared) individuals.

Response:

The matter of “coincidence of protein expression upregulation is a rather weak argument towards function and involvement of the upregulated molecule” was also raised by reviewer 2 in the first round of revision (specific comment #13): “*Whether TLR2 non-activating dengue strains also increase TLR2 expression levels and whether their pathogenicity differs from that of activating counterparts both qualitatively and quantitatively should be indicated more rigorously by the manuscript*”. Based on the reviewer’s initial comment, we revised the result section of the manuscript accordingly as suggested (Previous Manuscript NCOMMS-18-2649203A, pages 13 and 17).

Results:

(Page 13) To further substantiate the role of TLR2 as a regulator of inflammatory responses, we isolated PBMCs from healthy, DENV-seronegative, donors and infected them under TLR2 axis blocking and non-blocking conditions with DENV2 16681 strain at multiplicities of infection (MOI) of 10, as described previously (39). To gain further insights into the possible repercussions of TLR2-engagement on PBMCs, we used virus preparations that had a differential capacity to activate HEK-Blue™ hTLR2 reporter cells (Table 2).

(Page 17) Importantly, the observed variations were donor-dependent rather than virus preparation-dependent, highlighting the inherent differences between the HEK-Blue™ hTLR2 and monocytes in TLR2-mediated sensing of DENV.

Unfortunately, in the second revision, the reviewer did not acknowledge and/or endorse on our response, and rephrased the question with regard to the implications of other DENV serotypes: *“protein expression upregulation implicates the respective protein functionally, here in pattern recognition” the revised manuscript “still” fails to attempt to explain how general TLR2 upregulation correlates with TLR2 activation by some dengue strains but not by others wherein others than “2” have not been analyzed in detailed fashion because “Dengue virus serotype 2 is widely used in the field to study DENV pathogenesis due to the fact that it is usually associated with severe disease as well as in vivo murine models”.* We thus further revised the text and specifically stated throughout the manuscript results section (manuscript pages 15, 19 and 20) as well as in the discussion (page 30), that the discrepancies observed in terms of the ability of some strains to activate HEK-Blue™ hTLR2 or primary monocytes are likely a consequence of different expression levels of the TLR2 axis components and pathways downstream thereof. We also point out that the HEK-Blue™ hTLR2 cells are used to monitor only NF-κB and AP-I-dependent activation, while other transcription factors are also likely to be involved in human primary monocytes. Therefore, in our opinion, the fuzziness feeling of reviewer 2 is completely unsubstantiated. Moreover, we addressed reviewer’s 2 concerns by stratifying TLR2 expression in patients based on the infecting DENV serotype and found a similar association of TLR2 expression on classical monocytes in patients that developed severe disease following DENV1 and DENV2 while no differences were observed for intermediate and non-classical monocytes (Figure S3, here below). These data support our premise that sustained high levels of TLR2 play a role in disease development during the acute phase of DENV infection. Consequently, we do not understand why reviewer 2 repeatedly addresses this matter without acknowledging our explanation and additionally included data.

Results:

(Page15) To further substantiate the role of TLR2 as a regulator of inflammatory responses, we isolated PBMCs from healthy, DENV-seronegative, donors and infected them under TLR2 axis blocking and non-blocking conditions with DENV2 16681 strain at multiplicities of infection (MOI) of 10, as described previously (40). To gain further insights into the possible repercussions of TLR2-engagement on PBMCs, we used virus preparations that had a differential capacity to activate HEK-Blue™ hTLR2 reporter cells (Table 2).

(Page 19) Supernatants from UV-DENV treated PBMCs led to a relatively mild activation of HUVEC when compared to infectious DENV (Fig. S12 A and B), despite potent activation of NF-κB in HEK-Blue™ hTLR2 by UV-DENV (Fig. 2A).

(Page 20) Altogether, these results support the DENV replication- dependent and TLR2-mediated production of inflammatory mediators are capable to activate endothelial cells in vitro (Fig. 4) and highlight that variety of pathways cross-talking following TLR2 axis-dependent and independent sensing of DENV infection. In addition, it is important to note that the observed variations in cytokine production following (UV-) DENV infections were donor- rather than virus preparation-dependent, further highlighting the inherent

differences in TLR2-mediated sensing of DENV infection between the overexpression system HEK-Blue™ hTLR2 and primary monocytes, which are equipped with multiple PRRs.

Discussion:

(Page 30) Sensing of DENV infection by TLR2 did not rely on DENV infection as UV-irradiated virus was also sensed by the TLR2 expressed on HEK-Blue™ hTLR2 cells. Notably, the results obtained in PBMCs did not entirely mirror the results obtained in HEK-Blue™ TLR2 cells in which both virus preparation triggered equal NF-κB activation. The level of NF-κB/AP1 activation measured in the reporter cell lines is unlikely to reflect the variety of different immunomodulators that can be released down-stream of TLR2 in primary immune cells. The differences in cytokine levels produced between active DENV infection and UV-DENV in PBMCs may reflect the differential expression, as well as the crosstalk of different PRR's in sensing replicative and non-replicative virions, which do not occur, or are not detectable in the NF-κB/AP-1 reported system (46, 60). Indeed, by UV-irradiating the virus, we likely impeded the endosomal sensing of viral ssRNA by TLR7/8, which ultimately might have affected the potentiating effect of TLR2/7/8 cross talk (61). Moreover, the TLR2/CD14 dependent production of IFN-λ1, which has been described to be released by dendritic cells in a TLR3 dependent manner (62), may suggest the possibility of TLR2/TLR3 cross talk (63). Indeed, in human DC's the stimulation of TLR2 blocked the induction of cytokines that are controlled by TLR3 (45). Further studies are required to elucidate the cross talk of different PRR's in the course of DENV infection.

Figure S3. Increased expression of TLR2 in CM from DENV1 and DENV2 positive patients correlates with DENV disease severity. (A and B) PBMCs were isolated from 45 patients undergoing acute DENV serotype 1 infection who developed relatively mild (DF, n=26) or severe (DHF/DSS, n=19) disease. Patients were classified based of infecting serotype, (A) DENV1 and (B) DENV2. Percentages of cells expressing TLR2 were determined for each monocyte subsets stratified by disease severity. Bars represent median with IQR. P values were obtained by Mann-Whitney test (*P< 0.05, **P<0.01).

Moreover, the fact that in patients, TLR2 expression on classical monocytes differentiates between the acute phase of mild or severe disease suggests that it could be used as a prognostic marker for disease pathogenesis and could play a role in dengue infection. We completely agree with the reviewer that correlation does not imply causation and functional analysis has to be performed to ground the causation. That is exactly why we addressed the role of TLR2 in (1) virus replicative cycle (manuscript pages 8-10, Fig. 2E and 2F; Figs. S6-S7) and (2) virus sensing in PBMCs (manuscript pages 15-16 and 19-20, Fig. 3; Fig. 5; Fig. S9, Figs. S13-S16). Collectively our data suggest that high TLR2 expression on monocytes, as observed in healthy pediatric individuals can (1) facilitate DENV infection in these cells and (2) during the course of infection underlies the developing inflammatory responses. Thus, we strongly disagree with the statement that our conclusions are based on “coincidence of expression”.

Notably, reviewer 2 mentioned the matter of glycosylation only in the second round of revision (the matter was brought up in the first round by reviewer 1) and seems to miss the point that by deglycosylating a virus, we are affecting its ability to attach to cells, a first step in the virus infectious cycle that precludes the specific receptor binding and internalization. As such, deglycosylating “virus homogenate” would therefore fail to answer the original question of reviewer 1, which was, whether different glycosylation patterns in mosquito or mammalian cells influence TLR2 binding. The fact that reviewer 1 was satisfied by our answer together with the additional experiments that we performed with primary cell-derived virus (Fig. S4B, manuscript page 12), further validates this point. In addition, in the second revision round, and to address reviewer’s 2 comment concerning the matter of glycosylation (which he did not comment on in the final revision), we referred to confidential data from another of our ongoing studies in which we have found

[[Redacted]]

We have very specifically explained why TLR2 KO experiments using murine cells did not fit the scope of our current manuscript. Macrophages among other cells are considered target cells of dengue virus however they express a different set of specific receptors, which confer their susceptibility and/or permissiveness to the virus (Miller et al., 2008; Lo et al., 2016). Most importantly, we and others (Matsuguchi et al., 2000; Juarez et al., 2010) found that the expression of TLR2 is extremely low on murine (and human) macrophages and significantly lower compared to human monocytes. Experiments using murine TLR2 KO macrophages would thus not provide any useful additional insights into the function of TLR2 on human monocytes- which was the primary aim of our study. Instead, as suggested, we performed additional experiments in human PBMCs to further attest the role of the TLR2 axis in the induction of inflammatory responses as well as in DENV infection. Specifically, we identified TLR2/CD14 dependent cytokines and chemokines induced in the course of infection (Fig. 5 and Fig. S15) and, following the recommendation of the reviewer, by increasing the time of infection to 48h, we were able to detect DENV positive monocytes in a TLR2/CD14 dependent manner (Fig. 2F and Fig. S6).

We agree with the reviewer that the CD14 isotype control is missing in our experiments (Fig. S6 and Fig. S15)- that is our oversight. Importantly however, the specificity of the CD14 block has been confirmed by the inability to no longer detect expression of CD14 while preserving detection of CD16 and TLR2 monocytes in flow cytometry, following pretreatment with the blocking antibody (Reviewer Figure 2, here below). Should the isotype control be needed, we will gladly add it. With respect to TLR6 and TLR1 antibodies, we would like to stress that they are of the same isotype and sensing of the infection was TLR2/TLR6 but not TLR2/1 dependent (Fig. 2E and Fig. 3C and 3D). Therefore, in all following experiments, the TLR1 antibody served as an internal control of specificity. Moreover, TLR1/TLR6 Ab isotype controls did not show any effect on monocyte infection (Fig. S6).

Reviewer Figure 2. TLR2 expression and monocyte subsets distribution after 2h blockage with α CD14. PBMC's from healthy donors (n=1) were (mock)-treated with α CD14 (5 μ g/mL) for 2h. **(A)** Monocyte subsets distribution was determined by flow cytometry. **(B)** TLR2 surface expression in monocytes.

With regards to “The seemingly contrasting role of TLR2 on CM and NM are likely to be linked to their overall distinct function.” This point was brought up in the second revision by reviewer 2: “which appears as limitedly informative”. We revised the discussion accordingly in the last response to the reviewers by recapping the function of both CN and NM (manuscript page 28, here below). This quote is thus clearly taken out of the context.

Discussion

(Page 28) The seemingly contrasting role of TLR2 on CM and NM are likely to be linked to their overall distinct function. CM are equipped with various PRRs and scavenger receptors, recognizing PAMPs thereby removing microorganisms, lipids, and dying cells via phagocytosis and thus involved in sensing and inflammatory response to stress-inducing factors. NM or patrolling monocytes on the other hand play a unique role in protecting endothelial integrity (49–51) and once activated, they differentiate into anti-inflammatory macrophages to repair damaged tissues (52). The functions of TLR2 on these monocyte subsets may also be dictated by the differential expression of CD14, CD16 and/or other receptors capable of modulating DENV infection and DENV-infection-mediated responses (53, 54).

Matsuguchi, T., et al. Gene expressions of toll-like receptor 2, but not toll-like receptor 4, is induced by LPS and inflammatory cytokines in mouse macrophages. *J Immunol.* 165: 5767-5772(2000).

Zybert, IA., et al. Functional importance of dengue virus maturation: infectious properties of immature virions. *J Gen Virol*, 89: 3047-3051 (2008).

Miller, JL., et al. The mannose receptor mediates dengue virus infection of macrophages. *PLoS Pathog*, 4(2):e17 (2008).

Rodenhuis-Zybert, IA., et al. Immature dengue virus: a veiled pathogen? *PLoS Pathog*, 6: e1000718 (2010).

Juarez, E., et al. Differential expression of Toll-like receptors on human alveolar macrophages and autologous peripheral monocytes. *Respir Res.* 11 (2): 1-3 (2010).

Dejnirattisai, W., et al. Cross-Reacting Antibodies Enhance Dengue Virus Infection in Humans. *Science*, 328 (5979): 745-748 (2010).

Lo, YL., et al. Dengue virus infection is through a cooperative interaction between a mannose receptor and CLEC5A on macrophages as a multivalent hetero-complex. *PLoS One*, 11(11): e0166474 (2016).

The last and newly introduced comment of the reviewer regarding the title of the manuscript seems somewhat delayed considering that the title has not changed since the submission of the article in November 9, 2018. However, we have no objection to revise the title to “TLR2 facilitates DENV infection in human monocytes and triggers inflammatory responses that can contribute to disease pathogenesis”.

Reviewer #3

Overall summary

The authors did not address the concerns raised previously by the reviewer. The fundamental issue is that the natural biological function of monocytes is phagocytosis of unwanted cells or cell debris. The infected monocytes defined in the current manuscript by the presence of dengue viral antigens are insufficient to address the infection of these cells by the dengue virus. These could be a consequence of phagocytosis of infected cells by these monocytes. According to Fig. 1, the NS3+ cells in monocyte populations were between 60 and 100 percent. It should be very easy to clarify whether these monocytes were infected directly by dengue virus or due to the activity of phagocytosis of dengue virus infected by these monocytes. The authors should sort out these cells and perform co-culture to recover the infectious dengue virus to clarify the observed infected phenomenon; was it due to phagocytose other infected cells or due to direct infection by the dengue virus??

In addition, time is sensitive issue in dengue pathogenesis. The attached file demonstrates that the time lapse from the bite of mosquito carrying infectious dengue virus to the time of rash develop, on average is 7 to 10 days. Kinetically, the peak of dengue viremia is during the febrile stage. As addressed in the current manuscript, the subjects enrolled to the study were on average 3.3 to 3.9 days after onset of the fever, indicating that the viral roads were downward bound. Hence, it is imperative to clarify whether the current findings are the consequences of infection of dengue virus in these monocytes or as claimed to be the instigator for the disease severity. It is important to know that correlation does not constitute to the cause of disease development.

Response

We are utterly confused by the comments of the reviewer 3, since in the initial review round, the reviewer acknowledged our response to his suggestion regarding the *in vivo* permissiveness of monocytes and patient virus in which we state that the volume of pediatric serum sample is a major limiting factor in the number of experiments that can be performed and therefore impedes the sorting experiment for which a large number of cells is required: *“As for the response to in vivo virus, if the current study is not feasible, a discussion with current state of in vivo finding of dengue virus should be made in order to make fair presentation for the scientific community”*. Accordingly, and as suggested by reviewer 3, we added a paragraph in the discussion section of the manuscript addressing the current views on *in vivo* infection. We specifically, as recommended by reviewer 3, discussed the matter of different maturation status of human viruses (here below discussion paragraph in blue).

Discussion:

(Page 30) Unfortunately, due to the limited amount of sample obtained from our pediatric cohort, we could not assess the effect of the human circulating virus in our *in vitro* systems. Interestingly, Raut *et al* (59), showed that DENVs serotype 1 circulating in

humans are much more infectious when compared to the DENVs produced in laboratory cell lines. Moreover, the human circulating DENV1 virions showed a higher degree of maturity than those cultured *in vitro*. Considering the reduced ability of human primary cell derived DENV2 to engage TLR2 described in our studies, it will be important to address how maturation levels of the virions influence the course of DENV infections. We are currently assessing the role of immature dengue particles in our *in vitro* models.

Regarding the issue of detecting viral infection, we feel the reviewer forgot or discarded the arguments we provided in the previous revision round. There, we acknowledged an array of reports published in the last decades demonstrating *ex vivo* and *in vitro* DENV infection in monocytes as well as our new data validating the ability of the virus to infect primary monocytes (Chen *et al.*, 2002; Chao *et al.*, 2008; Michlmayr *et al.*, 2017), including a very recent publication from Prof. Eva Harris lab, in Nature Microbiology (Michlmayr *et al.*, 2017), in which DENV infection of monocytes is underscored by assessing the expression of a nonstructural (not contained in a virus particle and expressed only in cells replicating the virus) protein 3 (NS3) of the virus. As such, this is a well-established method to differentiate between detection of incoming virus (as implied by the reviewer) and active replication, and is used also for other viruses including in a recent publication from Dr. Raphael Gaudin lab, in Nature Communications (Ayala-Nunez *et al.*, 2019) wherein Zika virus infection is underscored by detection of a nonstructural protein NS2B of the virus.

Furthermore, to substantiate the role of TLR2 in dengue virus replication in monocytes, we performed *in vitro* infection experiments in PBMCs using UV-irradiated DENV that is capable of entering cells but does not replicate (Fig. 2F and Fig. S6, see below). Importantly, our data was in line with that of Michlmayr *et al.*, 2017, and showed that also by detection of DENV-E antigen, we can distinguish between the replicative virus and the incoming virus. In addition, the data substantiated that ability of TLR2 to facilitate DENV replication in monocytes and implied that high TLR2 expression on the surface of monocytes from healthy children can increase the susceptibility of these cells to DENV virus infection. Please note, that following the previous suggestion of the reviewer 3, we adjusted the wording from permissive to susceptible since we agreed with the reviewer that while staining NS3 indicates active replication it does not per se infer virus production. Consequently, it is unclear to us why reviewer 3, despite his approval of the arguments in the second round, circles back to the first round of revision and seems to withhold acknowledging the fact that we had already addressed all of their comments.

Panel F, Fig. 2. DENV2 infects human primary monocytes in a TLR2/CD14 dependent manner. (F) PBMC's from healthy donors (n= 2) were (mock) - treated with α TLR2, α TLR1, α TLR6 (5 μ g/mL) and α CD14 (3 μ g/mL) for 2h prior to infection with DENV2 at MOI of 20 (n=6) for 48h. Percentages of DENV-E (4G2)- positive cells were determined by flow cytometry. Data represent the mean \pm SEM. P values were obtained by one-tailed paired t test (***) $P < 0.001$, **** $P > 0.0001$).

Figure S6. DENV2 but not UV-inactivated DENV2 infects human primary monocytes. (A and B) PBMC's from healthy donors (n= 2) were (mock) - treated with α TLR2 and α TLR1/6 Isotype controls (5 μ g/mL) for 2h prior to infection with DENV2 at MOI of 20 (n=6) or its UV- inactivated equivalent (UV-DENV2) (n= 6) for 48h. Percentages of DENV-E (4G2)- positive cells were determined by flow cytometry. Data represent the mean \pm SEM. P values were obtained by one-tailed paired t test (***) $P < 0.001$).

Reviewer's 3 last statement regarding the viremia's kinetics and timing of the role of TLR2 in DENV infection and pathogenesis is ungrounded and confuses the aim and message of our study. In fact, the PDF file that reviewer 3 referred to, shows a figure from a paper published in 1952 (Sabin, 1952) with, what we can only assume based on the its color and smooth character, an additional manipulated arbitrary line drawn on it, which does not in any way provide the evidence the reviewer is referring to. In addition, we could not find that figure in Sabin's paper, however a subsequent revision of Sabin's research by Snow et al in 2014 showed an example of such a figure, which is differed from the figure the reviewer referred to (added here below). It is thus very disappointing to read that based on this figure, and only in the final revision, the reviewer manages to challenge the setup and message of the study. Collectively, our data clearly suggest that

high TLR2 expression on monocytes, as observed in healthy pediatric individuals can (1) facilitate DENV infection in these cells and (2) during the course of infection underlies the developing inflammatory responses. The fact that in patients, TLR2 expression on classical monocytes differentiates between the acute phase of mild or severe disease (even if it is in the beginning of downward bound viremia phase) suggests that it could be used as a prognostic marker for disease pathogenesis.

[Redacted]

Sabin, AB., et al. Research on dengue during world war II. *Am J Trop Med Hyg*, 1(1): 30-50 (1952).

Snow, GE., et al. Research on dengue during world war II revisited. *Am J Trop Med Hyg*, 91(6): 1203-1217 (2014).

Michlmayr, D., et al. CD14+CD16+ monocytes are the main target of Zika virus infection in peripheral blood mononuclear cells in a pediatric study in Nicaragua. *Nat Microbiol*, 2: 1462-1470 (2017).

Ayala-Nunez, V., et al. Zika virus enhances monocyte adhesion and transmigration favoring viral dissemination to neural cells. *Nat Commun*, 10: 4430 (2019).

Minor points

1. In Table 1. The data were collected in dengue endemic country. It is important to include the immune status of those healthy donors in the table.

Response:

We understand the interest of the reviewer with the regard to the immune status of the healthy children in our cohort. Unfortunately, we do not have details with regard to the immune status of these children -apart from the fact that they were considered healthy and were dengue PCR-negative. As stated in the previous review round, obtaining blood samples from healthy children is very challenging. Therefore, we used samples that were collected during DENV season in 2018 for the purpose of screening for DENV+ asymptomatic cases. Due to the enormous workload (+100 samples/day for 6 months), the samples of the DENV PCR negative individuals were only stored in the biobank. Importantly, the high TLR2 surface expression on the monocytes of our healthy control individuals corroborates findings from other studies (Chang et al., 2007; Tadema et al., 2011; Karananou et al., 2016), suggesting that these data are representative irrespective of their immune status.

Chang, JH., et al. Changes in Toll-like Receptor (TLR)-2 and TLR4 Expression and Function but Not Polymorphisms Are Associated with Acute Anterior Uveitis. *Invest Ophthalmol Vis Sci*, 48: 1711-1717 (2007).

Tadema, H., et al. Increased expression of toll-like receptors by monocytes and natural killer cells in ANCA-associated vasculitis. *PLoS One*, 6 (9): e24315 (2011).

Karananou, P., et al. Altered Expression of TLR2 and TLR4 on Peripheral CD14+ Blood Monocytes in Children with Urinary Tract Infection. *Biomed Res Int*, 2016;6052891 (2016).

2. How specific of the home-made NS3 antibody for FACS analysis in these monocytes?

Response:

We do not understand the reviewer's comment here. The NS3 antibody we used in our study was not home-made as implied by reviewer 3 but commercially purchased (GeneText, cat GTX124252) as clearly stated in the material and methods section (Manuscript page 33). A figure showing its specificity had been included following the first revision round and included as Fig. S1B in our current manuscript (See figure below).

Fig. S1B. Validation of the anti-NS3 antibody in patient's PBMCs. PBMCs were isolated from a patient undergoing acute DENV infection. To confirm the presence of DENV, cells were stained with a rabbit anti-NS3 antibody followed by goat anti-rabbit IgG conjugated with FITC. A non-specific polyclonal rabbit antibody was used as a negative control for staining of DENV infection. The figure above represents a histogram comparing the anti-NS3 antibody (red line) and the isotype control (grey

Chen, Y-C., et al. Activation of terminally differentiated human monocytes/macrophages by dengue virus: productive infection, hierarchical production of innate cytokines and chemokines, and the synergistic effect of lipopolysaccharide. *J Virol*, 76(19): 9877-9887 (2002).

Chao, Y-C., et al. Higher infection of dengue virus serotype 2 in human monocytes of patients with G6PD deficiency. *PLoS One*, 3(2): e1557 (2008).

Michlmayr, D., et al. CD14+CD16+ monocytes are the main target of Zika virus infection in peripheral blood mononuclear cells in a pediatric study in Nicaragua. *Nat Microbiol*, 2: 1462-1470 (2017).

Reviewers' Comments:

Reviewer #2:

Remarks to the Author:

The interesting manuscript is improved to a notable degree. For instance new figure 5 (legend background yellow as indicator of novelty) is complex yet largely strong in respect to TLR2 and CD14 implication as DENV sensors but fails to support the TLR6 involvement brought up otherwise. Furthermore, a potential TLR2 agonist is being named, yet as such put into relation with the subsequent wording. As triple function assignment appears the implication of TLR2 as infection biomarker and DNV entry receptor in addition to a DENV sensor function. In this context, the statement "the fuzziness feeling of reviewer 2 is completely unsubstantiated. " from the author's perspective surprises this reviewer against the background of a seeming lack of narrowing down (if not identification) of a distinct or at least class of virus product molecule/s beyond NS1 (implicated previously as TLR4 ligand according to the introductory text and repeatedly the discussion) as TLR2 ligand "beyond glycosylation which differs in various strains rendering specific ones TLR2 stimulatory" and evidence for TLR2 coreceptor involvement beyond anti TLR1 and - 6 antibody experiments rather than knock down or knock out application throughout even the current manuscript. Not ".. some strains to activate HEK-BlueTM hTLR2 or primary monocytes are likely a consequence of .." rather than the sheer strain difference in respect to TLR2 activating capacity is the point viewed by this reviewer as requiring further elucidative effort towards the ligand side at least conceptually or correlatively (e.g. 'glycosylation xy is carried by strain a but not b', whether stimulatory capacity correlates negatively or positively with TLR2 activating capacity and pathogenicity appears as being of large interest also). Mouse k.o. application would surely be informative towards collection of further evidence for a -/+TLR1/6/CD14 involvement, yet application of human cells in this sense would be equally appropriate. Affecting global yet moiety (e.g. N- or O-) specific experimental deglycosylation of virus products would support the yet seemingly merely textual implication of TLR2 specific cell activation by (an unspecified) glycosylation. Such analysis would rather not be aiming at analysis of virus attachment to host cells but "As such, deglycosylating "virus homogenate" would therefore * answer the original question of reviewer 1, which was, whether different glycosylation patterns in mosquito or mammalian cells influence TLR2 binding. *[fail to] in original text removed here". If one of the points listed above and considerable as important - if not crucial - would have been brought up by this reviewer in the secondary review round as suggested in the reply, the also mentioned bringing up by another reviewer in the first round also mentioned in the current rebuttal letter needs to be weighed against to keep up with fairness.

Texts such as "CM are equipped with various PRRs and scavenger receptors,

recognizing PAMPS thereby removing microorganisms, lipids, and dying cells via phagocytosis and thus involved in sensing and inflammatory response to stress-inducing factors. NM or patrolling monocytes on the other hand play a unique role in protecting endothelial integrity (49–51) and once activated, they differentiate into anti-inflammatory macrophages to repair damaged tissues (52). The functions of TLR2 on these monocyte subsets may also be dictated by the differential expression of CD14, CD16 and/or other receptors capable of modulating DENV infection and DENV-infection-mediated responders (53, 54).” applied as argument towards this reviewer’s points and cited from the revised discussion paragraph appear as tightly limitedly informative. Further revision towards achievement of increased clarity is demandable. Also, “.. we found that on the susceptible to DENV infection CD14++ classical monocytes, TLR2 expression correlated with severe disease development.” in the abstract appears as somewhat difficult to comprehend both, textually and correspondingly. For instance, whether high expression of sensors would not – by enhancing sensitivity - enhance rather than impair infection clearance towards lowering of disease severity could be considered. Consequently, sensor blockade should be detrimental in infection because it would blind the host immune system towards allocation of “the battle field” to the invader. If, however, the TLR2/6/CD14 complex is the DENV entry receptor, blockade of each complex member or better k.o. or k. down should abrogate virus binding rather than reduce it – dubbed as “paradox” in the abstract for a not necessarily ad hoc comprehensible reason - and “entry receptor” would be worth to be contained in the title. From within the first introduction sentence, “.. cause at least 300 million infections per year, of which nearly 100 manifest clinically (1).” appears as certainly mistakable at least. The beginning of the respective sentence is “The four serotypes of DENV (DENV 1-4) ..” and should introduce the “DENV” abbreviation (if not doing so within the abstract is sufficient). Also, “.. presence of cross-reactive, infection-enhancing antibodies increase the risk of severe disease (5–7).” should be specified in respect to the underlying mode of action such as by explaining the nature of the “cross reactivity”. Moreover, “Importantly, their distribution is influenced by inflammatory conditions (32).” might possibly be better suited as “Importantly, their proportionality is influenced by inflammatory conditions (32).”. Remarkably, it is now clarified: “This indicates that sensing of virions is enough to trigger a shift in the monocyte subpopulations. This effect was, however, independent of TLR2, since TLR2 block did not prevent the increase in CM numbers (Fig. S9E).”, which honestly puts a DENV sensing role of TLR2 into relation. “Interestingly, at the concentration tested only the intracellular accumulation of IL-1 β was significantly reduced by prior TLR2 blockade (Fig. S13A), suggesting that differential pathways down-stream of TLR2- trigger the production of these cytokines (44).” seems to sum up IL-1beta as more than one cytokine? Also the next sentence “Indeed, TLR2 may also lead to the activation of the inflammasome pathway that is likely to contribute to IL-1 β production.” appears a somewhat fuzzy. The legend of figure S13 is immature in

that it fails to precisely inform on each subfigure while that of S14 contains "graph shows production in pictograms per milliliter .." indicating a need for revision. ".. influence the course of several bacteria, fungi and viral infections" might profit from consequent usage of adjectives rather than names. There is further potential for optimization of the wording such as by heavy text compression and clarification throughout the entire manuscript not excluding the newly entered text parts (yellow background). For instance, "Although the role of CD14 in DENV infection was not surprising as previous studies suggested its function as an attachment/entry receptor for DENV (39, 57), this is the first time TLR2 is observed to aid the virus in establishing infection. Surprisingly, however, blocking TLR2 or CD14 had no appreciable effect on viral production, suggesting that the few cells that were infected produced more virions per cell, or that other cell populations present in a very low frequency, and thus undetected by flow cytometry, are responsible for virus production (58)." is difficult to comprehend and should be elaborated further.

The paper comparatively analyzes TLR2 expression on immune cells of DENV infected individuals to a large degree and thus brings up potentially clinically relevant phenomena in respect to expression levels on specific monocyte sub populations at different disease severity grades. The functional implications such as DENV driven TLR2 activation and DENV binding to TLR2 independently of activation, however, still remain as conceptual rather than sufficiently comprehensively demonstrative. The DENV site mediating TLR2 docking should at least be narrowed down such as by bringing up the glycosylation – citation of an example for which seems to still lack - kind it depends on. "TNF- α " in "HEK-Blue™ hTLR4 cells (Fig. S4 C and D), which only responded to LPS and TNF- α treatments." is surprising against the background of the seeming unpopularity of HEK293 cell TNF- α responsiveness (unless receptor is overexpressed) but since this issue has not been brought up earlier it should not count towards the authors now. The forelast paragraph of the discussion starting with "Our study has a few limitations." and naming a number of receptors not yet analyzed for their DENV sensing properties appears as largely obsolete. For the sake of coherence also other text parts such as redundant ones should be revised respectively.

Reviewer #4:

Remarks to the Author:

This manuscript by Aguilar-Briseno reports an interesting investigation demonstrating the involvement of TLR2 in the pro-inflammatory response to DENV infection. The authors combined clinical with in vitro investigations to show that DENV binds TLR2 that activates NF κ B and induces the downstream pro-

inflammatory cytokines that increase endothelial cell permeability. They also suggest that TLR2/CD14 could also play a role in virus entry and hence determine infection tropism in the various monocyte subsets. They concluded that the TLR2 axis could be a therapeutic target to reduce the risk of severe dengue.

This manuscript has undergone revision and is thus well written. The evidence for TLR2 as a contributor to the inflammatory response to DENV is convincing, although I believe this could be strengthened even further as indicated below. I am, however, less convinced that TLR2-dependent clathrin-mediated endocytosis is a portal for DENV infection. While the authors were careful to use anti-NS3 antibodies to show infection in Fig 1D, in vitro experiments shown in Figs 2E and 2F used anti-E antibody to detect DENV. When taken with Fig S7, findings in Figs 2F could be due to DENV uptake but not infection. This claim, to me, proves distracting as the TLR2-induced pro-inflammatory response, although augmented by virus uptake, does not require TLR2 to also serve as a portal of infection. Mixing this claim based on relative weak data in an otherwise cohesive story on TLR2-induced pro-inflammatory response in dengue has not been helpful in telling a compelling story.

My other major concerns are as follows:

1. The clinical relevance of the in vitro investigations is unclear. There are three reasons for this: Firstly, would DENVs that are decorated with polyclonal cross-reactive antibodies still bind and activate TLR2? This is important as secondary infection is associated with increased risk of DHF/DSS, as shown in Table 1. Secondly, the viruses tested were all highly passaged, non-contemporaneous laboratory strains. Including viruses isolated in the patient cohort in the in vitro investigations would have been very useful. Finally, TLR2 activation is only pronounced with DENVs cultured in C6/36 cells. Same strains of DENV expanded in DCs and macrophages produced significantly less TLR2-dependent responses. Extrapolating these findings to the clinical scenario would suggest that TLR2 is only important at the point of infection but its role in driving pathogenesis of vascular leakage, which is thought to be correlated with viremia and possibly NS1 antigenemia levels, could be small relative to other pathways. In this regard, the inclusion of contemporary strains of DENVs in the investigation could also prove helpful.

2. Details on how the patients were diagnosed and classified into DHF/DSS vs DF and selected for inclusion in this study needs to be more detailed. Was diagnosis in all patients confirmed by RT-PCR or NS1 or were subjects that were IgM positive but RT-PCR/NS1 negative also included? This study used a case-control design – how many patients were enrolled and, of these, how many DHF/DSS and DF cases were selected for inclusion in this study? How was plasma leakage, which is central

to this study, confirmed in these patients. Lines 913-917 only described collection of a single blood sample whereas rising hematocrit has to be shown by more than one blood sampling.

3. I am puzzled by Fig 1D. Why were there so few samples (n=3) from DHF/DSS patients in this analysis? The number of subjects is very different from those shown in Fig 1C and S8. Moreover, the high infection rates in the small number of samples from DHF/DSS patients are in contrast to the seemingly lower viremia in DHF/DSS compared to DF patients shown in Fig S8. The latter data set also lacks detail on when viremia was measured, which could greatly confound the analysis. Details are needed to clarify how subjects were selected for the analyses shown in these figure panels.

4. The impact of TLR2 expression in NM appears to be very small compared to the difference in TLR2 expression in CM (Fig 1C). This difference did not remain statistically significant when the analysis was stratified by DENV serotype (Fig S3). The discussion in lines 801-810 overinterprets this data and is thus speculative.

Minor comment:

1. Please introduce what MOG stands for.
2. Clathrin-dependent endocytosis should be abbreviated as CDE. If the author wishes to use CME, then it would be more intuitive to use the term clathrin-mediated endocytosis.
3. Fig S6A. Is there an error in the x-axis labeling? The legend and text suggest that this data is based on inoculation with UV-DENV2.

Reviewer #2

Overall summary

The interesting manuscript is improved to a notable degree. For instance new figure 5 (legend background yellow as indicator of novelty) is complex yet largely strong in respect to TLR2 and CD14 implication as DENV sensors but fails to support the TLR6 involvement brought up otherwise. Furthermore, a potential TLR2 agonist is being named, yet as such put into relation with the subsequent wording. As triple function assignment appears the implication of TLR2 as infection biomarker and DNV entry receptor in addition to a DENV sensor function. In this context, the statement "the fuzziness feeling of reviewer 2 is completely unsubstantiated. " from the author's perspective surprises this reviewer against the background of a seeming lack of narrowing down (if not identification) of a distinct or at least class of virus product molecule/s beyond NS1 (implicated previously as TLR4 ligand according to the introductory text and repeatedly the discussion) as TLR2 ligand "beyond glycosylation which differs in various strains rendering specific ones TLR2 stimulatory" and evidence for TLR2 coreceptor involvement beyond anti TLR1 and -6 antibody experiments rather than knock down or knock out application throughout even the current manuscript. Not ".. some strains to activate HEK-Blue™ hTLR2 or primary monocytes are likely a consequence of .." rather than the sheer strain difference in respect to TLR2 activating capacity is the point viewed by this reviewer as requiring further elucidative effort towards the ligand side at least conceptually or correlatively (e.g. 'glycosylation xy is carried by strain a but not b', whether stimulatory capacity correlates negatively or positively with TLR2 activating capacity and pathogenicity appears as being of large interest also). Mouse k.o. application would surely be informative towards collection of further evidence for a +/-TLR1/6/CD14 involvement, yet application of human cells in this sense would be equally appropriate. Affecting global yet moiety (e.g. N- or O-) specific experimental deglycosylation of virus products would support the yet seemingly merely textual implication of TLR2 specific cell activation by (an unspecified) glycosylation. Such analysis would rather not be aiming at analysis of virus attachment to host cells but "As such, deglycosylating "virus homogenate" would therefore * answer the original question of reviewer 1, which was, whether different glycosylation patterns in mosquito or mammalian cells influence TLR2 binding. *[fail to] in original text removed here". If one of the points listed above and considerable as important - if not crucial - would have been brought up by this reviewer in the secondary review round as suggested in the reply, the also mentioned bringing up by another reviewer in the first round also mentioned in the current rebuttal letter needs to be weighed against to keep up with fairness.

The DENV site mediating TLR2 docking should at least be narrowed down such as by bringing up the glycosylation – citation of an example for which seems to still lack - kind it depends on.

Response:

We thank the reviewer for their comments and feedback provided to our manuscript. We have addressed all their concerns accordingly.

We agree with the reviewer that it will be important to dissect the role of glycosylation patterns in the interaction between DENV and TLR2. However, this is far beyond the scope of this manuscript. Thus, we have removed any claims concerning the role of glycosylation in the TLR2-mediated immune responses in the context of DENV infection. Instead, as the reviewer and editor suggested, we now included a paragraph in the discussion (see below) with a reference of how

the glycosylation of the DENV E and prM glycoproteins may influence the responses observed throughout our manuscript.

Concerning the involvement of the TLR1/TLR6/CD14, we are glad that reviewer now agrees that in this sense the use of human cell lines would be equally appropriate as using mice models.

Discussion (Pages 29-30, lines 859-880): DENV produced in mosquito cells triggered stronger TLR2-dependent NF- κ B activation than the DC- and M ϕ - derived virus. This suggests that the virus transmitted during the blood meal is likely to initiate and more significantly contribute to TLR2-mediated inflammation. The reason for the differences in the capacity to activate TLR2 axis depending on the virus origin is yet unclear however is likely attributable to the host- and/or cell-type specific modifications on the surface of the virus particle. For instance, differential glycosylation patterns of 2 potential N-linked glycosylation sites on the DENV E protein produced in mosquito and primary human DCs influence their capacity to interact with DENV receptors, DC-SIGN and L-SIGN, and thus may dictate DENV tropism in vivo (65). Interestingly, E protein is not the only protein on the surface of the virion that can be glycosylated. DENV prM protein contains a single glycosylation site, and although prM is cleaved by furin during viral maturation, a substantial fraction of uncleaved prM is present on some DENV particles. In fact, DENV exists as a number of different viral forms depending on the degree of maturation. PrM -containing fully immature and partially mature DENV virions are particularly abundant in mosquito and mammalian tumor cell lines-produced viral preparations (66, 67). On the other hand, DENV produced in primary cells (68) and human circulating DENV1 virions (69) seem to show a higher degree of maturity than those produced in cell lines. Considering the reduced ability of human primary cell derived DENV2 to engage TLR2 described in our study, it will thus be important to address how both glycosylation and maturation levels of the virions influence TLR2-mediated responses during infection. Unfortunately, due to the limited amount of sample obtained from our pediatric cohort, we could not assess the effect of the human circulating virus in our in vitro systems. However, we are currently assessing the role of immature dengue particles in our in vitro models.

Texts such as “CM are equipped with various PRRs and scavenger receptors, recognizing PAMPS thereby removing microorganisms, lipids, and dying cells via phagocytosis and thus involved in sensing and inflammatory response to stress- inducing factors. NM or patrolling monocytes on the other hand play a unique role in protecting endothelial integrity (49–51) and once activated, they differentiate into anti- inflammatory macrophages to repair damaged tissues (52). The functions of TLR2 on these monocyte subsets may also be dictated by the differential expression of CD14, CD16 and/or other receptors capable of modulating DENV infection and DENV-infection- mediated responders (53, 54).” applied as argument towards this reviewer’s points and cited from the revised discussion paragraph appear as tightly limitedly informative. Further revision towards achievement of increased clarity is demandable.

Response:

We have revised the text accordingly (discussion: pages 28, lines 806-815).

CM are equipped with various PRRs and scavenger receptors that recognize PAMPS, remove microorganisms, lipids, and dying cells via phagocytosis and are thus, involved in sensing and inducing inflammatory responses to stress-inducing factors. In contrast, NM or patrolling monocytes have a unique role in protecting endothelial integrity by removing damaged cells and debris. Moreover, once activated, NM differentiate into anti-inflammatory macrophages to repair

damaged tissues, thereby promoting wound healing and the resolution of inflammation (51–54). The functions of TLR2 on these monocyte subsets may also be dictated by the differential expression of CD14, CD16, TLR10, CD36 (20, 55–58) and/or other receptors capable of modulating DENV infection and DENV-infection-mediated responses (59, 60).

Also, “.. we found that on the susceptible to DENV infection CD14++ classical monocytes, TLR2 expression correlated with severe disease development.” in the abstract appears as somewhat difficult to comprehend both, textually and correspondingly. For instance, whether high expression of sensors would not – by enhancing sensitivity - enhance rather than impair infection clearance towards lowering of disease severity could be considered. Consequently, sensor blockade should be detrimental in infection because it would blind the host immune system towards allocation of “the battle field” to the invader. If, however, the TLR2/6/CD14 complex is the DENV entry receptor, blockade of each complex member or better k.o. or k. down should abrogate virus binding rather than reduce it – dubbed as “paradox” in the abstract for a not necessarily ad hoc comprehensible reason - and “entry receptor” would be worth to be contained in the title. There is further potential for optimization of the wording such as by heavy text compression and clarification throughout the entire manuscript not excluding the newly entered text parts (yellow background). For instance, “Although the role of CD14 in DENV infection was not surprising as previous studies suggested its function as an attachment/entry receptor for DENV (39, 57), this is the first time TLR2 is observed to aid the virus in establishing infection. Surprisingly, however, blocking TLR2 or CD14 had no appreciable effect on viral production, suggesting that the few cells that were infected produced more virions per cell, or that other cell populations present in a very low frequency, and thus undetected by flow cytometry, are responsible for virus production (58).” is difficult to comprehend and should be elaborated further.

Response:

We agree with the reviewer that more studies are required to fully comprehend the mechanism of the TLR2-mediated activation and expression during DENV infection, however, considering the body of the required analysis that work is beyond the scope of this manuscript. We now discussed the possible repercussions of TLR2 downregulation in the acute phase of infection in the discussion (page 29, lines 829-849).

Differential surface expression of TLR2 on CM of patients with relatively mild disease symptoms (DF) and those who progressed to severe disease (DHF/DSS) suggests a distinct regulation of TLR2 expression in these patients. Based on our data and that of others’(34) TLR2-axis mediated NF-κB activation is partially dependent on CME. Thus, the reduction of TLR2 expression on the surface of CM during viremia observed in patients with DF might indicate that activation of the DENV infection-mediated TLR2 axis led to TLR2/ligand internalization, and ultimately desensitization of the monocytes to TLR2-engaging PAMPs and DAMPS (61). In addition, TLR2 complex internalization might also result in more balanced inflammatory and antiviral immune responses, since internalization is required to induce IFN type I producing signaling cascades (18). At the same time, and considering the ability of TLR2 blockade to limit the number of DENV-Ag positive monocytes, it’s internalization would also impede the ability of the virus to infect these cells. Conversely, a sustained relatively high TLR2 surface expression following DENV infection, as seen on CM of patients that developed severe disease, could be suggestive of a reduced internalization of the TLR2 complex. In this scenario, prolonged sensing of TLR2 – engaging PAMPs and DAMPS would lead to mainly pro-inflammatory responses and a relatively higher number of infected monocytes. The exact mechanisms governing TLR2 expression

following DENV sensing and/or infection remain to be elucidated and will be the focus of our future studies. Importantly, reduced TLR2 surface expression following infection may also mirror the release of soluble TLR2, the levels of which were shown to be elevated in some infection and inflammatory conditions (62–64). Moreover, plasma levels of soluble TLR2 were found to be associated with a resolution of inflammation (64).

We have removed a claim concerning the role of the TLR2 axis as a putative receptor for DENV from the discussion. Instead, we conclude the role of TLR2/CD14 as a possible attachment factors for DENV and stress their limited role in virus production. We have revised accordingly in the abstract (page 1, lines 29-31) and the results (page 9-10, lines 286-307).

Abstract:

We found that on CD14++ classical monocytes, TLR2 expression correlated with severe disease development.

Results:

Interestingly, many viruses, including DENV, hijack CME to gain access to internal compartments of host cells (36, 37). Moreover, CD14 and its unknown co-receptor has been previously proposed to act as an attachment receptor for DENV (38). We therefore hypothesized that TLR2/CD14-dependent CME facilitates DENV entry to establish infection. Accordingly, we tested if blockade of TLR2 or its co-receptors had any impact on the percentage of infected cells and/or virus production. Since, due to technical issues, we could not exploit the same NS3 antibody as we used in our patient cohort, we tested the accumulation of E protein and used UV-inactivated DENV to ensure we do not detect incoming virus in our assay (Fig S6A). As shown in Fig. 2E, blockage of TLR2, CD14 (P<0.0001) and to a lesser extent that of TLR6 (P<0.05), reduced the number of DENV-Ag positive HEK-Blue™ hTLR2 cells. Consistent with these results, in PBMCs, specific blocking of TLR2 (P<0.0001) and CD14 (P<0.0001) but not that of TLR1/6 significantly decreased the frequency of DENV Ag-positive monocytes (Fig. 2F, and Fig. S6B for isotype controls). These data might explain why among the three monocyte subsets, the TLR2-positive CD14++ monocytes (Fig. 1C) expressed the highest level of DENV NS3 during acute DENV infection and why CD14 expression correlates with DENV replication in vivo (Fig. 2G). Counterintuitively, however, despite the significant decrease in the frequency of DENV-Ag positive cells, TLR2 and CD14 blockage had little to no effect on viral production in both cell models (Fig. S7). There was also no clear correlation between the expression of TLR2 (percentage of positive cells and MFIs) and the viral load in our patient’s cohort (Fig. S8 A and B), suggesting that TLR2/CD14-dependent CME does not significantly contribute to progeny virus release and/or other cells such as DCs (39) are the main source of the virus. Altogether, these data indicate that host cells utilize TLR2/6/CD14 mediated NF-κB activation as a quick innate mechanism to sense DENV infection.

From within the first introduction sentence, “.. cause at least 300 million infections per year, of which nearly 100 manifest clinically (1).” appears as certainly mistakable at least. The beginning of the respective sentence is “The four serotypes of DENV (DENV 1-4) ..” and should introduce the “DENV” abbreviation (if not doing so within the abstract is sufficient).

Response:

The numbers are correct but now are more exactly stated as cited by the WHO (<https://www.who.int/en/news-room/fact-sheets/detail/dengue-and-severe-dengue>). The DENV abbreviation had been introduced within the abstract (page 1, line 25) however we reintroduce it also in the first sentence from the introduction as requested (Page 2, lines 38-39).

The four serotypes of dengue virus (DENV1-4) are estimated to cause 390 million infections per year, of which 96 million manifest clinically (1).

Also, “.. presence of cross-reactive, infection-enhancing antibodies increase the risk of severe disease (5–7).” should be specified in respect to the underlying mode of action such as by explaining the nature of the “cross reactivity”. Moreover, “Importantly, their distribution is influenced by inflammatory conditions (32).” might possibly be better suited as “Importantly, their proportionality is influenced by inflammatory conditions (32).”.

Response:

We have revised the text accordingly (page 2 and 3, lines 45-47 and 89, respectively):

Factors exacerbating inflammation such as high viral titers and presence of cross-reactive, infection-enhancing antibodies raised from previous infection with another serotype, increase the risk of severe disease (5–7).

Importantly, their frequencies are influenced by inflammatory conditions (32).

Remarkably, it is now clarified: “This indicates that sensing of virions is enough to trigger a shift in the monocyte subpopulations. This effect was, however, independent of TLR2, since TLR2 block did not prevent the increase in CM numbers (Fig. S9E).”, which honestly puts a DENV sensing role of TLR2 into relation.

Response:

We appreciate the reviewer’s comment. We are not trying to claim that TLR2 is an only sensor of DENV infection. Indeed, not all tested cytokines were under control of TLR2 suggesting that other plasma membrane -expressed PRRs such as C-type lectin receptors such as MR, DC-SIGN and/or CLEC5A may play a role in the process (Wang *et al.*, 2011, Wu *et al.*, 2013, Arboleda-Alzate *et al.*, 2017). Importantly however, sole blockage of TLR2 reduced inflammatory responses capable of activating endothelial cells (manuscript Fig 4, page 21).

Wang *et al.*, DC-SIGN (CD209) promoter -336 A/G polymorphism is associated with dengue hemorrhagic fever and correlated to DC-SIGN expression and immune augmentation. 2011 *PLoS Negl Trop Dis.* 5(1): e934

Wu *et al.*, CLEC5A is critical for dengue virus-induced inflammasome activation in human macrophages. 2013. *Blood.* 121 (1): 95-106

Arboleda-Alzate *et al.*, Human macrophages differentiated in the presence of vitamin D3 restrict dengue virus infection and innate responses by downregulating mannose receptor expression. 2017, *PLoS Negl Trop Dis.* 11(10); 1-18

“Interestingly, at the concentration tested only the intracellular accumulation of IL-1 β was significantly reduced by prior TLR2 blockade (Fig. S13A), suggesting that differential pathways down-stream of TLR2- trigger the production of these cytokines (44).” seems to sum up IL-1beta as more than one cytokine? Also the next sentence “Indeed, TLR2 may also lead to the activation of the inflammasome pathway that is likely to contribute to IL-1 β production.” appears a somewhat fuzzy.

Response:

We would like to clarify that we were referring to both, TNF- α and IL-1 β , as mentioned in the results section (page 20, lines 579-584):

“As expected, positive control PAM3CSK4 induced the intracellular accumulation of IL-1 β and TNF- α in the concentration and TLR2-dependent manner (Fig. S13 A and B). Interestingly, at the concentration tested only the intracellular accumulation of IL-1 β was significantly reduced by prior TLR2 blockade (Fig. S13A), suggesting that differential pathways down-stream of TLR2-trigger the production of these cytokines (44).”

Concerning “Indeed, TLR2 may also lead to the activation of the inflammasome pathway that is likely to contribute to IL-1 β production”. We have revised accordingly (page 20, lines 584-585).

Indeed, TLR2 signaling may also lead to the activation of the inflammasome pathway that contributes to the production of IL-1 β (45, 46).

The legend of figure S13 is immature in that it fails to precisely inform on each subfigure while that of S14 contains “graph shows production in pictograms per milliliter ..” indicating a need for revision. “.. influence the course of several bacteria, fungi and viral infections” might profit from consequent usage of adjectives rather than names.

Response:

We have revised the legend of figure S13 accordingly.

Fig. S13. Intracellular accumulation of IL-1 β and TNF- α in PBMCs after exposure with DENV. PBMCs from healthy donors were (mock)-treated with α TLR2 (5 μ g/mL) for 2h prior treatment with PAM3CSK4 ((A) 600 ng/mL, 300 ng/mL, 150 ng/mL, 50 ng/mL), DENV2 (MOI 10) or UV-DENV2 (MOG 1000). (A-D) Percentage of monocytes (in PBMCs) with intracellular expression of TNF- α and IL-1 β was measured by flow cytometry at 6h and 18h post treatment respectively. (E and F) Percentage of lymphocytes with intracellular expression of TNF- α and IL-1 β , respectively. (A, C, E and F) Bars represent mean \pm SEM. P value was obtained by paired one-tailed t test. (*P<0.05). Source data are provided as a source data file.

We thank the reviewer for noticing the mistake in the legend of figure S14. We have revised the figure legends of the figures with LegendPlex data accordingly. Concerning “. influence the course of several bacteria, fungi and viral infections”, we have revised accordingly (discussion page 28, lines, 800-802).

For instance, single-nucleotide polymorphisms occurring in the TLR2/1/6 axis have been reported to influence the course of chlamydiosis, leprosy and hepatitis B and C virus infections (50).

The forelast paragraph of the discussion starting with “Our study has a few limitations.” and naming a number of receptors not yet analyzed for their DENV sensing properties appears as largely obsolete. For the sake of coherence also other text parts such as redundant ones should be revised respectively.

Response:

We have revised the penultimate paragraph of the discussion accordingly. (Page 30-31, lines 896-908).

Our study provides fundamental insights into the function of TLR2 in the course of DENV infection, however some limitations should also be considered. Nearly 74% of our cohort included patients undergoing secondary infections implying that they had pre-existing cross-reactive antibodies circulating in the blood at the time of sampling. Our in vitro infection model, however, did not address the infection enhancing or neutralizing effects of dengue-specific antibodies on TLR2-mediated immune responses (60). For instance, the phenomenon of antibody-dependent enhancement (ADE) of infection, postulating that sub neutralizing concentrations of DENV-specific antibodies, can facilitate an additional mode of entry, thereby enhancing DENV infection and the aberrant inflammatory responses seen in DHF/DSS patients. Additionally, the presence of DENV-Ab immunocomplexes is likely to influence TLR2/CD14-mediated responses (60). Lastly, considering the role of TLR2 and TLR4 in detecting bacterial PAMPS, it will be important to address the effect of common co-morbidities including bacterial and parasitic co-infections or microbial translocation (6, 7, 74, 75), on DENV pathogenesis and prognosis.

Reviewer #4

Overall summary

This manuscript by Aguilar-Briseno reports an interesting investigation demonstrating the involvement of TLR2 in the pro-inflammatory response to DENV infection. The authors combined clinical with in vitro investigations to show that DENV binds TLR2 that activates NFkB and induces the downstream pro-inflammatory cytokines that increase endothelial cell permeability. They also suggest that TLR2/CD14 could also play a role in virus entry and hence determine infection tropism in the various monocyte subsets. They concluded that the TLR2 axis could be a therapeutic target to reduce the risk of severe dengue.

This manuscript has undergone revision and is thus well written. The evidence for TLR2 as a contributor to the inflammatory response to DENV is convincing, although I believe this could be strengthened even further as indicated below. I am, however, less convinced that TLR2-dependent clathrin-mediated endocytosis is a portal for DENV infection. While the authors were careful to use anti-NS3 antibodies to show infection in Fig 1D, in vitro experiments shown in Figs 2E and 2F used anti-E antibody to detect DENV. When taken with Fig S7, findings in Figs 2F could be due to DENV uptake but not infection. This claim, to me, proves distracting as the TLR2-induced pro-inflammatory response, although augmented by virus uptake, does not require TLR2 to also serve as a portal of infection. Mixing this claim based on relative weak data in an otherwise cohesive story on TLR2-induced pro-inflammatory response in dengue has not been helpful in telling a compelling story.

Response:

We thank the reviewer for the overall positive and constructive feedback towards our manuscript. We have addressed all their comments and concerns point by point.

We understand the point of the reviewer with regard of the role of TLR2 as a portal for DENV infection. This premise is supported by the fact that we did not detect DENV E protein in the PBMCs exposed for 48h to the replication-incompetent UV-DENV (Fig. S6). In addition, based on our data indicating an important role of CME in TLR2/CD14 responses combined with multiple other studies (Van der Schaar *et al.*, 2008, Acosta *et al.*, 2008, Carro *et al.*, 2018) showing the importance of CME in DENV infection we now conclude that TLR2 and/or CD14 serve at least as attachment factors for DENV in monocytes. Importantly, other populations (dendritic cells) present in low numbers are likely to contribute to the production of virions and this could explain why the virus production was not affected by the TLR2 blockage. Since the relevance of TLR2 in facilitating DENV infection in monocytes requires thorough investigation, we have revised the text accordingly and removed the claim of its role as a putative receptor.

Van der Schaar *et al.*, Dissecting the cell entry pathway of dengue virus by single-particle tracking in living cells. 2008. *PLoS Pathog.* 4.

Acosta *et al.*, Functional entry of dengue virus into *Aedes albopictus* mosquito cells is dependent on clathrin-mediated endocytosis. 2008. *J Gen Virol.* 89: 474-484

Carro *et al.*, Blockade of dengue virus entry into myeloid cells by endocytic inhibitors in the presence or absence of antibodies. 2018. *PLoS Negl Trop Dis.* 12(8): e0006685.

(Results, pages 9-10, lines 286-307): Interestingly, many viruses, including DENV, hijack CME to gain access to internal compartments of host cells (36, 37). Moreover, CD14 and its unknown co-receptor has been previously proposed to act as an attachment receptor for DENV (38). We

therefore hypothesized that TLR2/CD14-dependent CME facilitates DENV entry to establish infection. Accordingly, we tested if blockade of TLR2 or its co-receptors had any impact on the percentage of infected cells and/or virus production. Since, due to technical issues, we could not exploit the same NS3 antibody as we used in our patient cohort, we tested the accumulation of E protein and used UV-inactivated DENV to ensure we do not detect incoming virus in our assay (Fig S6A). As shown in Fig. 2E, blockage of TLR2, CD14 ($P<0.0001$) and to a lesser extent that of TLR6 ($P<0.05$), reduced the number of DENV-Ag positive HEK-Blue™ hTLR2 cells. Consistent with these results, in PBMCs, specific blocking of TLR2 ($P<0.0001$) and CD14 ($P<0.0001$) but not that of TLR1/6 significantly decreased the frequency of DENV Ag-positive monocytes (Fig. 2F, and Fig. S6B for isotype controls). These data might explain why among the three monocyte subsets, the TLR2-positive CD14++ monocytes (Fig. 1C) expressed the highest level of DENV NS3 during acute DENV infection and why CD14 expression correlates with DENV replication in vivo (Fig. 2G). Counterintuitively, however, despite the significant decrease in the frequency of DENV-Ag positive cells, TLR2 and CD14 blockage had little to no effect on viral production in both cell models (Fig. S7). There was also no clear correlation between the expression of TLR2 (percentage of positive cells and MFIs) and the viral load in our patient's cohort (Fig. S8 A and B), suggesting that TLR2/CD14-dependent CME does not significantly contribute to progeny virus release and/or other cells such as DCs (39) are the main source of the virus. Altogether, these data indicate that host cells utilize TLR2/6/CD14 mediated NF- κ B activation as a quick innate mechanism to sense DENV infection.

Major concerns

1. The clinical relevance of the in vitro investigations is unclear. There are three reasons for this: Firstly, would DENVs that are decorated with polyclonal cross-reactive antibodies still bind and activate TLR2? This is important as secondary infection is associated with increased risk of DHF/DSS, as shown in Table 1. Secondly, the viruses tested were all highly passaged, non-contemporaneous laboratory strains. Including viruses isolated in the patient cohort in the in vitro investigations would have been very useful. Finally, TLR2 activation is only pronounced with DENVs cultured in C6/36 cells. Same strains of DENV expanded in DCs and macrophages produced significantly less TLR2-dependent responses. Extrapolating these findings to the clinical scenario would suggest that TLR2 is only important at the point of infection but its role in driving pathogenesis of vascular leakage, which is thought to be correlated with viremia and possibly NS1 antigenemia levels, could be small relative to other pathways. In this regard, the inclusion of contemporary strains of DENVs in the investigation could also prove helpful.

Response:

We have addressed each of the reviewer's concerns accordingly:

1.- The reviewer raised a very interesting point concerning the ability of DENVs particles decorated with polyclonal cross-reactive antibodies to activate TLR2. We agree with the importance of this matter for the clinical relevance of our data and now address this point in the discussion (page 30-31, lines 896-908).

Our study provides fundamental insights into the function of TLR2 in the course of DENV infection, however some limitations should also be considered. Nearly 74% of our cohort included patients undergoing secondary infections implying that they had pre-existing cross-reactive antibodies circulating in the blood at the time of sampling. Our in vitro infection model, however, did not address the infection enhancing or neutralizing effects of dengue-specific antibodies on TLR2-mediated immune responses (60). For instance, the phenomenon of

antibody-dependent enhancement (ADE) of infection, postulating that sub neutralizing concentrations of DENV-specific antibodies, can facilitate an additional mode of entry, thereby enhancing DENV infection and the aberrant inflammatory responses seen in DHF/DSS patients. Additionally, the presence of DENV-Ab immunocomplexes is likely to influence TLR2/CD14-mediated responses (60). Lastly, considering the role of TLR2 and TLR4 in detecting bacterial PAMPS, it will be important to address the effect of common co-morbidities including bacterial and parasitic co-infections or microbial translocation (6, 7, 74, 75), on DENV pathogenesis and prognosis.

2.- We agree with the reviewer that performing the experiments with patient derived viruses would be useful. However, viral yield directly obtained from patient isolates is low, especially taking into account that these are pediatric patients. Hence, it is not possible to obtain high titers of patient derived viruses required for these experiments without passaging them at least three times in mosquito C6/36 cells (WHO 2009). Importantly, DENV2 strain used in our study was produced directly from infectious clone and a maximum of two passages in C6/36 to increase the virus yield. Consequently, the used virus is virtually identical to the original 16681 strain isolated from DHF patient in Thailand (Kinney *et al.*, 1997).

Kinney *et al.* Construction of infectious cDNA clones for dengue 2 virus: strain 16681 and its attenuated vaccine derivative, strain PDK-53. 1997. *Virology*. 230(2): 300-308

World Health Organization. (2009). Dengue guidelines for diagnosis, treatment, prevention and control: new edition. Geneva: World Health Organization. <http://www.who.int/iris/handle/10665/44188>

3.- Although NS1 has been implicated in disease pathogenesis of vascular leakage. A recent report published in Bioarchive (Lee *et al.*, 2020), has put into relation the role of NS1 and severe dengue pathogenesis. Interestingly, by doing virus chimerization approaches the authors found that the prM-E structural region, but not NS1, promotes the production of inflammatory mediators and exerts an important role in DENV pathogenesis *in vivo*. Interestingly, we recently found that fully immature virions containing prM/E on their surface of (prM DENV) and which are commonly released from DENV infected cells potently activate our HEK-TLR2 cells. These immature virions were produced in human cells lacking the enzyme responsible for the maturation of the newly assembled virions progeny. Our preliminary data suggest that the extent of DENV maturation seems to be the key modulator of the TLR2-mediated sensing. We already addressed this matter in the previous rounds of revision and would like to keep this data confidential, as they are part of a separate and ongoing study.

Lee *et al.* Relative contribution of non-structural protein 1 in dengue pathogenesis. 2020. *bioRxiv*.

2. Details on how the patients were diagnosed and classified into DHF/DSS vs DF and selected for inclusion in this study needs to be more detailed. Was diagnosis in all patients confirmed by RT-PCR or NS1 or were subjects that were IgM positive but RT-PCR/NS1 negative also included? This study used a case-control design – how many patients were enrolled and, of these, how many DHF/DSS and DF cases were selected for inclusion in this study? How was plasma leakage, which is central to this study, confirmed in these patients. Lines 913-917 only described collection of a single blood sample whereas rising hematocrit has to be shown by more than one blood sampling.

Response:

We thank the reviewer for this question and have now addressed these issues in the paper.

The design of the case-control study was as following: suspected DENV patients were consecutively enrolled and a blood sample was drawn for DENV diagnosis (see below) at hospital admittance and at hospital discharge. In total, 110 subjects were enrolled. Patients were diagnosed as acute DENV-infected as following: a positive qRT-PCR or NS1 positive by rapid test at hospital admission, or seroconversion from DENV-IgM negative to IgM positive during the hospital stay (admittance and discharge sample). Patients who were DENV-IgM positive and RT-PCR/NS1 negative at admission were excluded from analysis. Of the 110 patients enrolled, 21 were negative for DENV diagnostics, 12 were positive for DENV IgM, and negative for qRT-PCR and/or NS1 at admission and hence excluded (see above), and 3 subjects were excluded due to lack of follow up sample at discharge. Of 20 subjects, samples were not available for analysis. This resulted in a cohort of 54 confirmed acute DENV-infected cases. Patients were followed up during their hospital stay and WHO 1997 classification was done after hospital discharge. 32 subjects were classified as DF, 22 as DHF and 10 as DSS patients. We have added this to the material and methods section of the paper (page 32, lines 925-934 and 942-944, respectively).

A second blood sample was collected upon discharge from the hospital. Patients were classified according to the WHO 1997 criteria upon hospital discharge (2). Plasma leakage was confirmed by at least one of the following manifestations: 1/ a rise in the hematocrit equal to or greater than 20% above average for age, sex and population in the admission sample (reference percentages: <http://www.hematocell.fr/index.php/les-cellules-du-sang/15-les-cellules-du-sang-et-de-la-moelle-osseuse/valeurs-normales-de-lhemogramme-selon-lage/129-hemogramme-selon-lage>) or 2/ A drop in the hematocrit following volume-replacement treatment equal to or greater than 20% of baseline and follow up visit or discharge (between 1-3 days after initial sample) or 3/ Signs of plasma leakage such as pleural effusion and ascites by ultrasound.

Patients were diagnosed as acute DENV-infected as following: a positive qRT-PCR or NS1 at hospital admission, or seroconversion from anti-DENV IgM negative to anti-DENV IgM positive during the hospital stay.

3. I am puzzled by Fig 1D. Why were there so few samples (n=3) from DHF/DSS patients in this analysis? The number of subjects is very different from those shown in Fig 1C and S8. Moreover, the high infection rates in the small number of samples from DHF/DSS patients are in contrast to the seemingly lower viremia in DHF/DSS compared to DF patients shown in Fig S8. The latter data set also lacks detail on when viremia was measured, which could greatly confound the analysis. Details are needed to clarify how subjects were selected for the analyses shown in these figure panels.

Response:

The anti-NS3 staining panel was only performed on a subset of patients as more PBMC are required for a good quality intracellular staining. Sufficient cell yield after PBMC thawing was used as selection criteria to perform the staining. Indeed, % of NS3 positive cells does not correlate with the viremia measured in the blood in all DENV patients. This is probably due to production of viral particles in other PBMC subsets, such as conventional DC, and at other locations in the body, such as lymph nodes.

As described in the materials and methods section, viral load was measured at time of hospital admittance and enrollment. This is within 96h after onset of fever. We have added this to the figure legend of Fig S8.

Fig. S8. TLR2 expression does not correlate with viral load. Correlation analysis of (A) percentages of TLR2 positive cells and (B) mean fluorescence intensity (MFI) of TLR2 expression on monocytes subsets and DENV viral load (determined by qPCR) in our patient's cohort. Green dots represent patients with dengue fever; red dots represent the patients that subsequently developed DHF/DSS. Viral load was measured at hospital admittance within 96h of onset of symptoms. Association was tested by Spearman correlation. Source data are provided as a source data file.

4. The impact of TLR2 expression in NM appears to be very small compared to the difference in TLR2 expression in CM (Fig 1C). This difference did not remain statistically significant when the analysis was stratified by DENV serotype (Fig S3). The discussion in lines 801-810 overinterprets this data and is thus speculative.

Response:

We agreed with the point of the reviewer. We have revised the text accordingly (Discussion, page 28, lines 803-815).

Although high frequencies of TLR2-expressing CM correlated with severe disease, the opposite trend, albeit less strong, was noted for non-classical monocytes. The seemingly contrasting roles of TLR2 on CM and NM are likely to be linked to their distinct functions. CM are equipped with various PRRs and scavenger receptors that recognize PAMPS, remove microorganisms, lipids, and dying cells via phagocytosis and are thus, involved in sensing and inducing inflammatory responses to stress-inducing factors. In contrast, NM or patrolling monocytes have a unique role in protecting endothelial integrity by removing damaged cells and debris. Moreover, once activated, NM differentiate into anti-inflammatory macrophages to repair damaged tissues, thereby promoting wound healing and the resolution of inflammation (51–54). The functions of TLR2 on these monocyte subsets may also be dictated by the differential expression of CD14, CD16, TLR10, CD36 (20, 55–58) and/or other receptors capable of modulating DENV infection and DENV-infection-mediated responses (59, 60).

Minor comments

1. Please introduce what MOG stands for.

Response:

We have revised accordingly. (Page 9, lines 261-263).

HEK-Blue™ hTLR2 cells were exposed to increasing numbers of virus particles i.e. multiplicity of genomes (MOGs) to ensure fair comparison.

2. Clathrin-dependent endocytosis should be abbreviated as CDE. If the author wishes to use CME, then it would be more intuitive to use the term clathrin-mediated endocytosis.

Response:

We have revised the text accordingly. (Page 9, lines 273-275).

While expressed at the plasma membrane, TLR2-induced NF- κ B activation is controlled by clathrin-mediated endocytosis (CME), in which CD14 serves as an important upstream regulator (34).

3. Fig S6A. Is there an error in the x-axis labeling? The legend and text suggest that this data is based on inoculation with UV-DENV2

Response:

We would like to clarify that there is no error in the x-axis of this figure. To note, the replication incompetent UV-DENV2 was used as a negative control for this experiment. Moreover, the lack of DENV antigen in the monocytes (within PBMCs) exposed to UV-DENV2 validates the infection observed on the PBMCs exposed to DENV2.